

# 1  Distribution and characteristics of supraglacial channels on

# 2  mountain glaciers in Valais, Switzerland

Holly Wytiahlowsky[1], Chris R. Stokes[1], Rebecca A. Hodge[1], Caroline C. Clason[1], Stewart
S.R. Jamieson[1]
[1]Department of Geography, Durham University, Durham, DH1 3LE, United Kingdom
*Correspondence to*: Holly Wytiahlowsky (holly.e.wytiahlowsky@durham.ac.uk)
**Abstract.** Supraglacial channels form a key component of glacier hydrology, transporting surface meltwater to
englacial and proglacial positions, which impacts ice flow dynamics, surface mass balance and the hydrochemistry
of glacial runoff. The presence of supraglacial channels is well-documented on ice sheets using satellite imagery,
but little is known about their distribution and characteristics on smaller mountain glaciers because most channels
fall below the resolution of freely-available satellite imagery. Here we use high-resolution (0.15 m) orthophotos
to delineate <2000 supraglacial channels (> 0.5 m wide) across a sample of 285 glaciers, 85 of which contain
channels, in Valais Canton, Switzerland, and investigate their distribution and characteristics. We find that glacier
hypsometry, size and slope are good predictors of drainage density, with glaciers characterised by lower relief
slopes (with fewer crevasses) and larger ablation areas (high meltwater supply) exhibiting higher drainage
densities. Drainage density is higher when glaciers terminate at mid-range elevations (2600 – 3100 m.a.s.l),
likely due to less surface lowering than at lower elevations, which allows channels to persist. On average, 80 % of high
order channels run-off supraglacially, with 20 % terminating englacially. However, there is marked inter-glacier
variability in where channels terminate, with 40 % of glaciers containing no englacially-terminating channels,
versus 3.5 % where all channels are terminate englacially. Most channels are slightly sinuous, with higher
sinuosities associated with large, high-order channels that are heavily incised and more likely to reactivate
annually. In comparison to ice sheets, the majority of channels reach the terminus supraglacially and little
meltwater is stored in lakes.



## 1    Introduction


Glaciers and ice caps are rapidly losing mass (Wouters et al., 2019; Hugonnet et al., 2021; Tepes et al., 2021) and
are anticipated to contribute to sea level rise throughout the 21st century and beyond (Bamber et al., 2019;
Edwards et al., 2021; Rounce et al., 2023). Glaciers in the lower latitudes (e.g., the European Alps, Caucasus,
New Zealand, the USA) are particularly vulnerable to atmospheric warming and may experience complete
deglaciation by 2100 under a strong warming scenario (e.g., RCP8.5) (Zekollari et al., 2019; Rounce et al., 2023).
In populated mountain regions, these changes will have profound impacts, as glaciers and snowpacks act as vital
water towers, supplying 1.9 billion people worldwide who live in or downstream of glacial catchments with crucial
freshwater (Carey et al., 2017; Zemp et al., 2019; Immerzeel et al., 2020; Sommer et al., 2020; Hugonnet et al.,
2021; Clason et al., 2023). Glacier meltwater responsible for feeding proglacial rivers is commonly transported to
the proglacial margin by supraglacial channels, with these channels forming an important component of the glacial
hydrological system. The presence and distribution of supraglacial channels holds implications for a range of
glacio-hydrological processes as they affect how efficiently meltwater is routed over, through and under glaciers,
along with potential to impact on suspended sediment concentrations and hydrochemistry of proglacial rivers.
Higher suspended sediment concentrations, for example, pose harm to downstream ecosystems and proglacial
reservoirs, with concentrations generally higher if meltwater is routed to the bed (Swift et al., 2002), rather than
transported supraglacially. Supraglacial channels also affect glacier surface mass balance because channels can
efficiently transport mass from the glacier as runoff. In contrast, in the absence of channels, surface melt may
percolate into snow or firn and refreeze. Despite the importance of meltwater routing, the controls and patterns of
meltwater transport on mountain glaciers remain relatively understudied. Notably, it is not fully understood why
the channelised flow of meltwater occurs on some glaciers but not others, nor what effect this has on glacier
systems (Pitcher & Smith, 2019).
The term 'supraglacial stream' was first coined in the 1970s and 1980s from observations of channels in
Scandinavia and the European Alps (e.g., Knighton, 1972, 1981, 1985; Ferguson, 1973; Hambrey, 1977; Seaberg,
1988), with their morphology often compared to terrestrial streams. However, these early studies only provided
small-scale, local observations of channels on individual glaciers. By comparison, a recent revival in supraglacial
channel research has primarily focused on large-scale remote-sensing observations of the Greenland Ice Sheet
(GrIS) (e.g., Smith et al., 2015; Gleason et al., 2021; Karlstrom & Yang, 2016; Yang et al., 2015, 2016, 2018,
2019, 2020, 2021, 2022), and to a lesser extent, Antarctica (e.g., Kingslake et al., 2017, Bell et al., 2017; Chen et
al., in press). Recent remote sensing techniques for channel detection on the GrIS (e.g., Yang and Smith, 2013;
King et al., 2016) have not been applied to mountain environments because the majority of channels are likely to
be below the resolution of the highest resolution satellite platforms (e.g., Sentinel-2 (~10 m), WorldView 3 (~3.7
m)). As a result, it is not known whether the principles that govern channel formation on larger ice sheet
environments also apply to mountain glaciers. The latter are characterised by typically steeper and more complex
topography and tend to have a larger coverage of debris cover in comparison to ice sheet surfaces.
Much remains unknown about channel distribution in mountainous environments, but previous research has
helped to establish some fundaments about supraglacial channels (e.g., Knighton, 1972, 1981; Ferguson, 1973;
Yang et al., 2016). Specifically, the formation of meltwater channels is thought to occur when channel incision
via thermal erosion exceeds the rate of surface lowering (Marston, 1983). Channel formation is also influenced



by the rate of meltwater production and surface topography, with channels tending to form parallel to the steepest ice flow direction (Irvine-Fynn et al., 2011; Mantelli et al., 2015). Surface topography may re-enforce itself, as once an incised channel forms, the higher incision rates may result in an increasingly topographically constrained channel that reactivates annually. On a smaller scale, high concentrations of channels have been suggested to be correlated with high surface roughness (Irvine-Fynn, 2011), while the present of micro to macro scale surface structures also influence meltwater routing (e.g., Rippin et al., 2015). Where channels occur, they are often reactivated annually depending on their incised depth, which is strongly controlled by discharge rates and slope (St Germain & Moorman, 2019). The most deeply incised channels are suggested to be a product of high discharge or increased slope, and most commonly exhibit asymmetric cross-profiles due to the dominant direction of solar radiation (St Germain & Moorman, 2019). Additionally, discharge rates are a strong control on channel morphology, in particular sinuosity, with channels observed to increase in sinuosity throughout the melt season (e.g., Dozier, 1976; Hambrey, 1977; St Germain & Moorman, 2019). Similar to terrestrial river networks, supraglacial channels generally follow Horton's laws, meaning that higher-order channels (i.e., where the highest-order is the main channel) are longer, have lower slopes, and are comprised of a lower number of channel segments (Horton, 1945; Yang et al., 2016). However, much of what we know about supraglacial channels was established from cold to polythermal systems or from observations of a small number of individual glaciers (e.g., Knighton, 1972, 1981, 1985; Gleason et al., 2016; St Germain & Moorman, 2019).

In this paper we investigate a range of potential controls on channel distribution and properties for a large sample of glaciers (n = 285) in a region characterised by high melt rates. We use high-resolution (~0.15 m) orthophoto imagery from 2020 to produce the first comprehensive inventory of almost 2,000 supraglacial channels in a mountain glacier environment, with a focus on Valais Canton, Switzerland. Our aim is to characterise the morphometry of supraglacial channels on mountain glaciers, providing insight into where and why they form. Using GIS software, we extract channel metrics (length, sinuosity, slope, elevation, terminus type, proximity to debris) and glacier characteristics (aspect, size, drainage density, elevation, crevassed extent) which are supplemented by qualitative observations. Using our data, we explore the relationship between glacier and channel characteristics using statistical measures and infer whether glacier surface characteristics can explain the presence or absence of channels.

## 2    Study location

When compared to many glacierised regions, Switzerland has the largest repository of high spatial and temporal resolution national LiDAR and orthophoto surveys, providing excellent coverage for the mapping of supraglacial channels. We focus on Valais Canton in southern Switzerland which contains 303 glaciers over 0.1 km$^2$, covering a total area of 545 km$^2$ in 2015 (Fig. 1; Linsbauer et al., 2021). It is the most glacierised Swiss Canton and in 2015 glaciers in Valais had a mean area of 1.8 km$^2$, a median of 0.43 km$^2$ and a maximum area of 77.3 km$^2$ (Aletsch Glacier) (Linsbauer et al., 2021). We identified Valais Canton as a suitable study site as its glacier size distribution is comparable with Switzerland as a whole, and the glaciers range from shallow to steep gradients, have differing hypsometries (ice area-elevation distributions) and vary in crevasse densities. Thus, this study site captures a wide range of potential influences on channel distributions and characteristics. Valais is comprised of the Bernese Alps



to the north and the Pennine Alps in the south, separated by the Rhône Valley (Fig. 1). Glaciers in the Bernese
Alps are the largest in the canton and most exhibit a south-to-southeast aspect (mean: 163°). In contrast, the largest
glaciers in the Pennine Alps have a north and west aspect (mean: 347°). Glaciers in Valais have an average
maximum elevation of 3450 m.a.s.l (min: 2356, max: 4599) and an overall mean elevation of 3091 m.a.s.l (min:
2267, max: 4025).

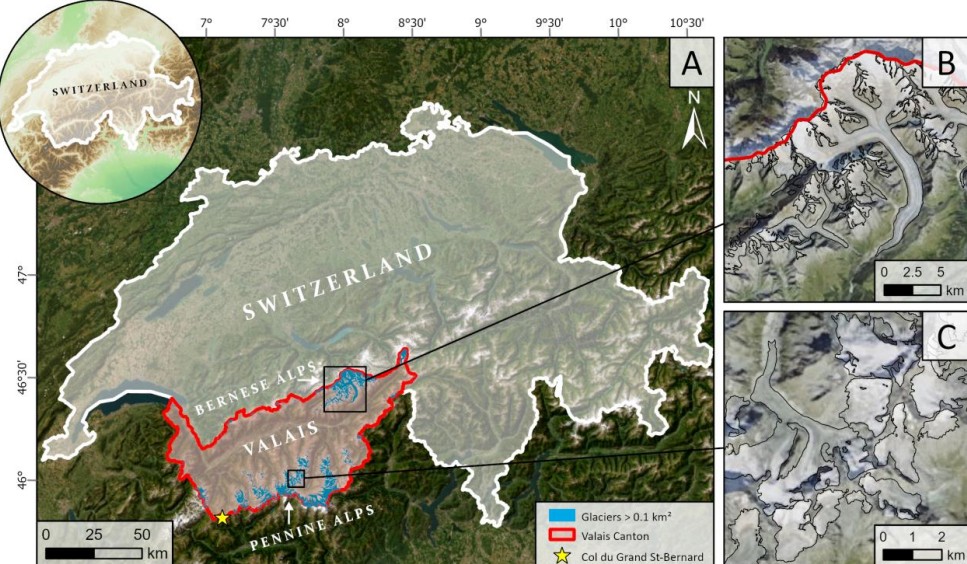


**Figure 1: The study site area, which contains 303 glaciers >0.1 km². (A) The location of Valais Canton (red)**

**shown within Switzerland, with glaciers shown in blue; (B) the larger glaciers (e.g., Aletsch Glacier, centre**
**right) in the north of Valais Canton; and (C) the smaller glaciers in the south of Valais Canton, with varying**
**levels of debris cover. The location of Col du Grand St-Bernard meteorological weather station is indicated**
**by a yellow star. Glacier outlines used are from the Swiss Glacier Inventory (SGI2016), with glacier extent**
**shown as of 2015-16. The outlines are overlayed on basemap imagery sourced from Esri (2024).**


Within Valais, the only meteorological station with a similar elevation to many glacier termini is Col du Grand
St-Bernard (2472 m.a.s.l) in the Pennine Alps, which (between 1991 and 2020) recorded mean July air
temperatures (2 m) of 8.4°C, mean January temperatures of -6.9°C, and mean annual temperatures of -0.1°C. At
Col du Grand St-Bernard, July averages 140 mm of precipitation (1991-2020), with 12.7 days a month
experiencing > 1 mm of precipitation, compared to a January average of 242 mm across an average of 12.9 days.
The coldest temperatures are consistently recorded at Jungfraujoch (3571 m.a.s.l) in the Bernese Alps, which
(between 1991 and 2020) recorded mean July air temperatures (2 m) of 0.4°C, mean January temperatures of -
12.5°C, and mean annual temperatures of -7.3°C. However, Switzerland's climate is changing and mean air
temperatures between 2013 and 2022 were 2.5°C warmer than pre-industrial temperatures (MeteoSwiss, 2024),
which has greatly impacted the mass balance of Swiss glaciers in recent decades (Fischer et al., 2015; Davaze et
al., 2020). The most recent Swiss glacier inventory (2016) records a historical glacier surface area change rate of



–0.6 % a$^{-1}$ (1973 – 2016, -350 km$^2$) and reveals a statistically significant acceleration of glacier area loss in the
Alps, with annual losses between 2010 to 2016 (-0.8 %) having increased since 1973 to 2010 (-0.6 %) (Linsbauer
et al., 2021). Volumetric losses in the Swiss Alps between 1980 and 2010 were 22.51 ± 1.76 km$^3$, which were
largely concentrated between 2700 and 2800 m.a.s.l, with no elevation band experiencing positive elevation
changes (Fischer et al., 2015).

### 3    Methods

#### 3.1    Imagery acquisition and channel delineation

The application of recently deployed automated methods of channel detection in ice sheet settings (e.g., Yang et
al., 2019) were tested at our study site and found to be insufficient for detecting the generally much smaller
channels. We initially applied a modified NDWI$_{ice}$ method developed by Yang and Smith (2013) to delineate water
bodies from WorldView-2 imagery (1.84 m) on the Greenland Ice Sheet. However, our NDWI$_{ice}$ output
predominantly detected water-filled crevasses, and only detected larger channels, missing many channels that
were visible, but contained very small amounts of water or incised channels where the water surface was not
visible. As a result, we undertook manual mapping, which in some instances has been found to be seven-fold more
accurate in ascertaining channel density compared to multispectral methods (King et al., 2016). We obtained high-
resolution cloud-free orthophoto imagery (0.15 m resolution) from SwissTopo (swisstopo.admin.ch), with
acquisition dates during mid-July 2020. Imagery was not available for later in the melt season; hence 6% of all
glacier termini were still snow-covered and were omitted from further analyses as the presence or absence of
channels could not be confirmed.
We first removed all glaciers in Valais Canton smaller than 0.1 km$^2$ from our study glaciers (582 reduced to 303
glaciers). This is because they are likely too small to produce large enough channels to be detected by our imagery,
and because many small glaciers in the Swiss Glacier Inventory (SGI2016) are unlikely to meet the criteria to be
identifiable as glaciers (Leigh et al., 2019). Once glaciers with snow-covered termini were also omitted, a total of
285 glaciers remain in our study site. Each glacier was then systematically surveyed for supraglacial channels in
*QGIS* (e.g., Fig. 2). Only channels confidently visible at a 1:1,000 scale were delineated for the purpose of
consistency, meaning the minimum channel width we delineated is ~0.5 m wide. We solely focus on these larger
(>0.5 m) channels because of difficulties in delineating small channels objectively, which include problems with
differentiating complex rill networks from structural features (e.g., fractures) and the need for more subjective
judgments where channel width may periodically decrease beyond the pixel width. Individual channels were
mapped from their downstream end until they were no longer clearly visible or when channels could not be
confidently and objectively mapped. When channels have tributaries above the mapping resolution, the main
channels were mapped as one segment, continuing up the largest channel at each confluence. Each tributary
channel was then subsequently mapped as a new individual segment. Once mapped, each entire channel segment
was assigned codes based on its attributes, i.e., where it terminated (e.g., terminus, moulin, crevasse, lake,
periphery, adjoins another channel, disappears beyond the resolution), and whether it was on bare ice or a debris-
covered part of the glacier.



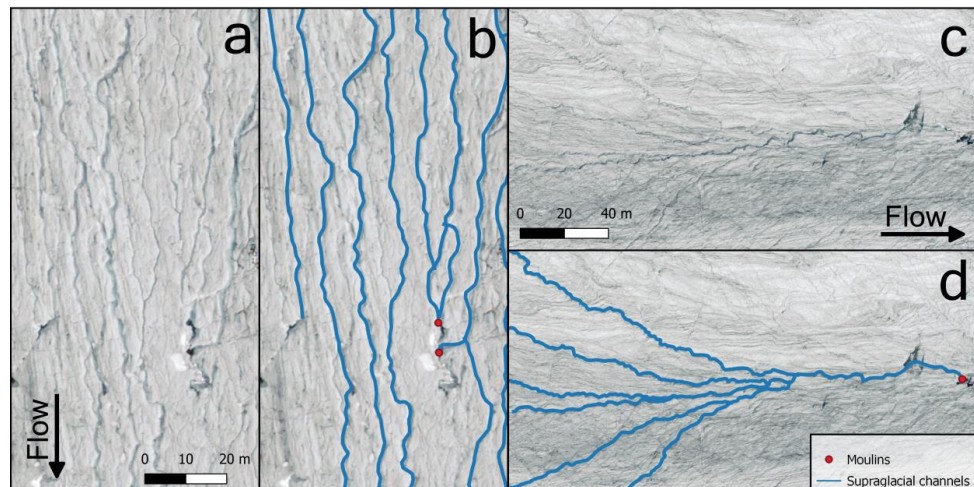

**Figure 2: Examples of the mapped output and corresponding orthophoto. Channels are shown in blue, and moulins are represented as red circles when a mapped channel is moulin terminating. (a - b) Supraglacial channels on the Rhone Glacier and (c-d) Allalin Glacier. Imagery source: Federal Office of Topography swisstopo.**

### 3.2    Metrics

A total of 1890 channel segments (polylines) were mapped across 85 glaciers found to contain channels. We then used the high-resolution (0.5 m) SwissALTI3D DEM (2019) from swisstopo.admin.ch (1 sigma accuracy of ± 0.3 m for each dimension) to extract morphometric characteristics from each channel segment.  The DEM is coarser than the orthophotos used for channel delineation and there is a one-year offset between their acquisition dates. However, as the DEM is used to calculate larger scale metrics such as elevation and slope, the small offset is unlikely to affect overall results. Channel metrics extracted were segment length, straight line distance, elevation (minimum, maximum), elevation difference and channel slope. The start and end point of each segment were then used to derive the sinuosity of each segment (channel length/straight line distance) and drainage density for each glacier (total length of channels/glacier area). We use the glacier snow-free area at the time of mapping to calculate drainage density, which results in a higher value that if the entire glacier area was used.

Glacier characteristics were obtained from the Swiss Glacier Inventory (SGI2016), which provided information on glacier area, aspect, and elevation (minimum, maximum and mean) in 2015 (Linsbauer et al., 2021). This record is used as it is the most up-to-date record of Swiss glacier area, but glaciers have since undergone substantial recession. The values for glacier slope from the SGI2016 cannot be used as they encompass the whole glacier, whereas for our analysis we wanted to measure the slope of the snow-free portion of the ablation area at the time of channel mapping. To calculate slope values, each glacier polygon was clipped to its snow-free area, and then zonal statistics in *QGIS* were used to extract the mean, minimum and maximum slope value from the SwissALTI3D DEM for each polygon. The snow-free slope value is the only glacial slope value used in data



analyses. Upon completion of mapping, we assigned codes to each glacier based on the extent of debris cover and
crevassed area due to their potential controls on channel formation. Glacier debris cover is visually estimated and
allocated to one of five classes (none, <10 %, 10-25 %, 25-50 % and >50 %) and crevassed area is assigned to
one of three classes little to none (less than 10 % of the ablation area), moderate (10-50 % covered), and heavily
crevassed (covers >50 % of the ablation area).

**3.3    Statistical tests**
To determine whether there is a relationship between channel morphometry and glacier characteristics, we
produced a correlation matrix using Spearman's rank correlation ($\rho$) (e.g., St Germain & Moorman, 2019). Each
metric used in this analysis comprises 1890 values, each representing an individual channel segment. The analysis
used the following channel variables: segment length, channel slope, sinuosity, minimum elevation, maximum
elevation and elevation range, and the following glacier variables: drainage density, glacier area, mean slope of
the snow-free area, aspect, glacier minimum elevation, glacier mean elevation and glacier maximum elevation.
For each of the glacier variables, all channel segments on the same glacier are allocated the same value. A singular
ANOVA test was conducted to determine the significance of the relationship between debris cover and sinuosity
as an ANOVA test is best suited to determining if there is a significant difference between different classes of
debris cover. We also conducted a Principal Component Analysis (PCA) to determine the relationship between
variables and to identify drivers of variance amongst the dataset, with data normalised to aid in identifying patterns
within the data.

**4    Results**
**4.1    Glacier observations**
The study area contained 285 glaciers with an area over 0.1 km$^2$ and a snow-free terminus in 2020 (Linsbauer et
al., 2021). Of these, 85 glaciers were found to have supraglacial channels above the mapping resolution (~0.5 m).
Glaciers with channels (n = 85) have a mean area of 5 km$^2$ and glaciers without channels (n = 200) have a mean
area of 0.6 km$^2$. However, glacier area frequency distributions peak in the 0.1 to 1 km$^2$ category for both glaciers
with and without channels (Fig. 3a). All glaciers larger than 5.6 km$^2$ were found to contain channels (Table 1, Fig.
3a). Where channels are present, glaciers have an overall mean slope of 21° and a maximum mean slope of 43°,
with glacier slope being positively skewed towards lower slope values (Table 1, Fig. 3b). By comparison, where
channels are absent, glaciers are characterised by steeper overall slopes (mean: 28°, max: 45°) (Fig. 3b).
Glaciers without channels are more likely to terminate at higher elevations (mean minimum elevation: 2936 m)
compared to glaciers with channels (mean minimum elevation: 2797 m), which are often characterised by longer
valley glacier tongues. Where glaciers support channels, they are more likely to have a higher maximum elevation
(mean max elevation: 3637 m) than glaciers without channels (mean max elevation: 3555 m). Where channels are
present, there is a mean drainage density of 2.4 km/km$^2$ and a maximum of 15.2 km/km$^2$. The latter was found on
the Upper Theodul Glacier, which is situated on a low slope plateau and has the lowest glacier slope angle in the



dataset (13°) (Fig. 3c, Fig. 4a). To summarise, glaciers containing channels are larger, have lower mean slopes,
and have a larger portion of their area at lower elevations compared to glaciers without channels.
**Table 1: A quantitative summary of glacier and channel characteristics.**

| | Channel Length (m) | Channel Slope (°) | Sinuosity | Drainage Density (km/km²) | Mean Glacier Slope (°) | Glacier Area (km²) |
|---|---|---|---|---|---|---|
| **Count** | 1890 | 1890 | 1890 | 85 | 85 | 85 |
| **Minimum** | 5.2 | 0.8 | 1.0 | 0 | 10.4 | 0.1 |
| **Median** | 152.2 | 6.3 | 1.1 | 1.5 | 20.6 | 1.5 |
| **Mean** | 211.7 | 8.0 | 1.1 | 2.4 | 21.0 | 5.0 |
| **Maximum** | 4314.4 | 47.8 | 3.8 | 15.3 | 43.0 | 83.0 |
| **Range** | 4309.3 | 47.0 | 2.8 | 15.2 | 32.6 | 82.9 |
| **Standard Deviation** | 228.3 | 6.3 | 0.1 | 2.6 | 6.5 | 10.7 |
| **Standard Error** | 5.3 | 0.1 | 0.0 | 0.3 | 0.7 | 1.2 |
| **Kurtosis** | 65.8 | 7.4 | 153.0 | 9.0 | 1.1 | 35.3 |
| **Skewness** | 5.5 | 2.3 | 9.5 | 2.6 | 0.8 | 5.4 |


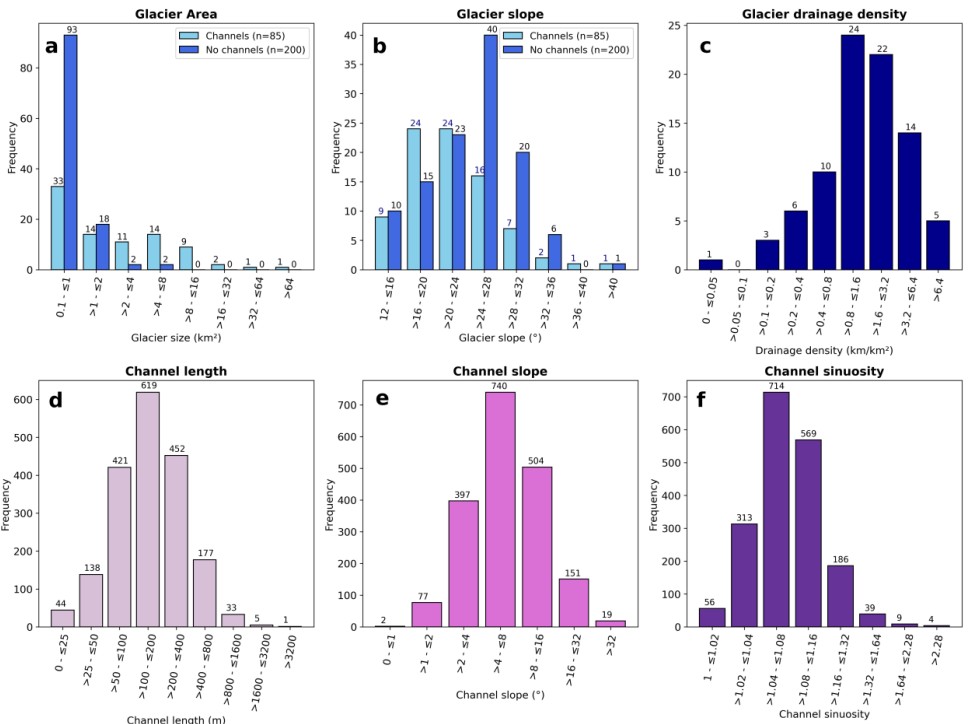

**Figure 3: Histograms of extracted metrics. Note that the x-axis is log a scale (except for Fig. 3b) and the numbers above each bar represent the number of channels/glaciers within each class. (a) Glacier area (km²); (b) glacier slope (°); (c) glacier drainage density (km/km²); (d) channel length (m); (e) channel slope (°); and (f) channel sinuosity.**



Figure 4: Examples of supraglacial channels. (a) Channels on the Upper Theodul Glacier; (b) a channel at the interface between bare ice and debris-covered ice on Glacier du Brenay; (c) channels on the Great Aletsch Glacier - note the straight segments where crevasses have been exploited; (d) channels on the debris-covered terminus of the Upper Aletsch Glacier; and (e) sinuous channels towards the terminus of Gorner Glacier. Flow indicates the glacier flow direction. Imagery source: Federal Office of Topography swisstopo.



### 4.2    Channel characteristics

Individual channel segments have a mean length of 212 m, with a positively skewed leptokurtic distribution (Fig. 3d; Table 1). Few segments exceed 1,600 m as the length of most glacier's ablation area is smaller than this value. The channel segments have a mean slope of 8°, and most exhibit a slope between 4 to 16° (Fig. 3e; Table 1). The maximum channel slope observed is 48°, but the overall distribution is positively skewed towards smaller slope values. The sinuosity index of each channel ranges from 1 (straight line) to a maximum of 3.8, with a mean value of 1.1, which is slightly sinuous, but not high enough to be defined as meandering (Table 1). Sinuosity is the most positively skewed variable, with a highly leptokurtic distribution, as most channels are not very sinuous (Fig. 3f).

Channels terminate in a range of settings, with 47 % joining another channel, 15 % terminating in crevasses, 14 % terminating in moulins, 13 % disappearing below the mapping resolution, 8 % running off at the glacier terminus, 2 % running off the side of the glacier, and 1 % terminating in a supraglacial lake. When only considering terminal segments, 72 % of segments terminate englacially (crevasses or moulins), 25 % run-off (terminus or periphery), and 3 % terminate in a supraglacial lake. However, larger glaciers with higher drainage densities disproportionately impact on these values. For example, 582 out of the 1890 mapped channel segments are on the Aletsch Glacier, where no channels reach the terminus, hence englacially terminating channels may be overrepresented by a singular glacier. Thus, when the percentage of channels terminating in each position are extracted as an average value from each glacier, 80 % of channels reach the terminus supraglacially and 20 % of channels terminate englacially. Overall, 48 % of glaciers have no englacially-terminating channels, with only 3.5 % of glaciers that solely contain englacially terminating channels.

Where channels occur, qualitative observations indicate structural and topographic controls on their distribution and morphology. For example, large channels often occur at the interface between debris-covered and bare ice (e.g., Fig. 4b), particularly adjacent to medial moraines, where they are confined to a topographic depression, commonly occurring at the confluence between two tributaries. The influence of glacier structure on channel morphology is also observed where trace or shallow crevasses are exploited to produce long, straight channel sections (e.g., Fig. 4c). By comparison, the most sinuous channels tend to occur at lower elevations on large glaciers characterised by larger flat areas towards to their terminus (Fig. 4e).

### 4.3    Controls on channel morphology and distribution

To investigate whether there are links between different supraglacial channel characteristics, we plotted our extracted metrics against each other (Fig. 5). Previous studies informed our choice of variables tested for potential relationships, with a focus on how glacier properties (slope, area and elevation) affect glacier drainage density (e.g., Yang et al., 2016) and channel morphology, such as sinuosity and channel length (e.g., St Germain & Moorman, 2019). We found that the most sinuous channels are most likely to occur on lower glacier slopes (0 to 10°), with a clear upper boundary demonstrating that channels on steeper slopes (> 20°) are unlikely to exhibit a sinuosity over 1.2 (Fig. 5a). A relationship between channel segment length and slope is also apparent, with the longest channel segments occurring on the lowest slope angles, which often have a lower density of crevasses (Fig. 5b). This relationship is clearly defined by an upper limit, indicating slope is a clear control on both channel



length and sinuosity (Fig. 5a-b). We find no noticeable difference in sinuosity between channels on bare ice and
those on debris-covered glaciers (Fig. 5c). However, channels proximal to debris (i.e., channels which are likely
to have some sediment content) are more likely to be highly sinuous than channels on bare ice or continuous
debris cover, with differences between the classes statistically significant ($p = <0.05$) in an ANOVA test (Fig. 5c).
Channel segments that terminate in moulins tend to be the longest (mean: 341 m, max: 1999 m), followed by
channels that disappear below the mapping resolution (mean: 259 m, max: 4314 m), and then channels reaching
the glacier terminus (mean: 214 m, max: 1193 m) (Fig. 5d). The very few (1 %) channels that terminate in
supraglacial lakes tend to be short (mean: 109 m, max: 260 m), along with channels that adjoin a higher-order
channels (mean: 169 m, max: 1174 m) (Fig. 5d).

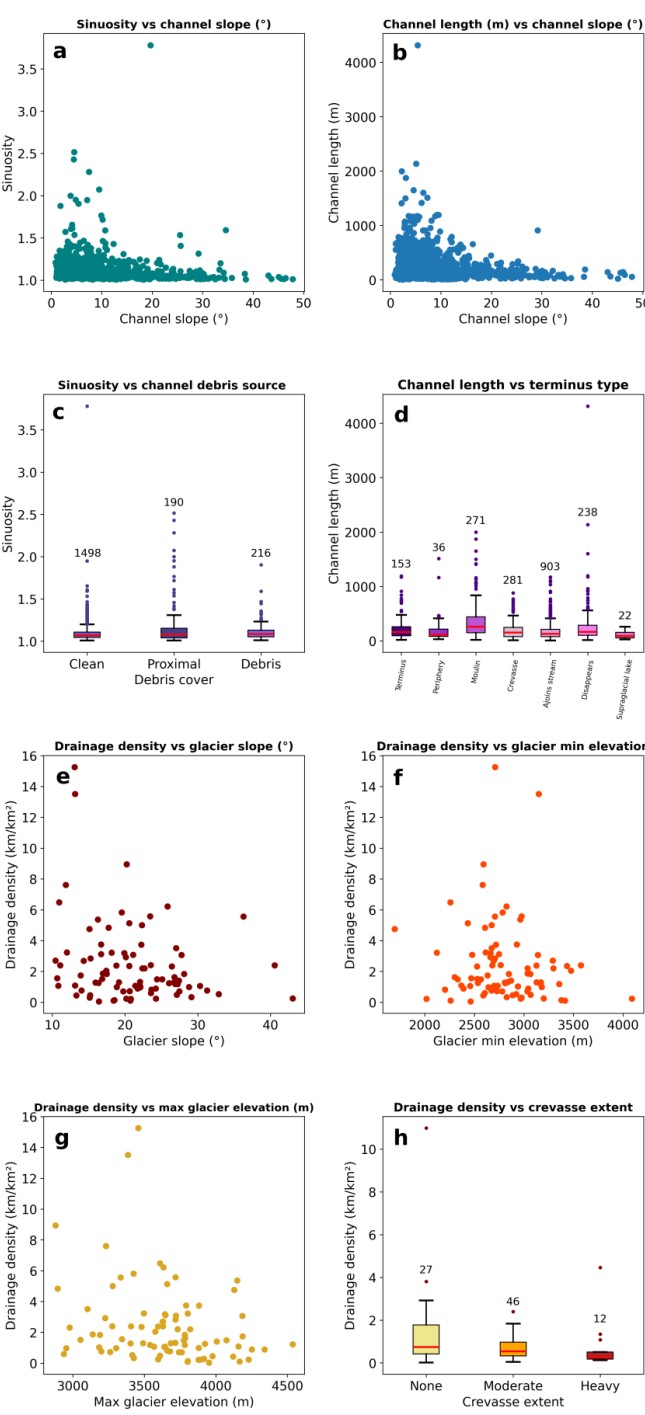


**Figure 5: Relationships between channel characteristics (a-d) and glacier metrics (e-h). Plots a-d contains**

**data from 1890 channels, and plots e-h contain data for the 85 glaciers with channels. (a) Sinuosity vs**





channel slope (°); (b) channel length vs channel slope (°); (c) sinuosity vs channel debris source; (d) channel
length (m) vs terminus type; (e) drainage density (km/km$^2$) vs glacier slope (°); (f) drainage density
(km/km$^2$) vs minimum glacier elevation; (g) drainage density (km/km$^2$) vs maximum glacier elevation (m);
and (h) drainage density (km/km$^2$) vs crevasse extent.

The influence of glacier characteristics on channel characteristics is less evident, perhaps due to the lower number
of data points. However, a relationship between drainage density and glacier slope exists, with the highest drainage
densities occurring on the lowest surface slopes (Fig 5e). Glaciers with high drainage densities tend to have a
minimum elevation range between 2600 and 3100 m.a.s.l (Fig. 5f), whereas a relationship between drainage
density and maximum elevation is less evident (Fig. 5g). However, the very highest drainage densities tend to
occur where glaciers have a lower maximum elevation (Fig. 5g). Additionally, the crevassed extent of a glacier
affects drainage density, as increased channel inception prevents the formation of longer channels (Fig. 5h).

**4.4  Spearman's rank and principal component analysis**
We examined the controls on channel morphology and drainage density by calculating a correlation matrix. We
use Spearman's rank correlation ($\rho$) and significance values ($p$), given that many of our relationships do not appear
linear. Additionally, given the large sample size within our dataset and that most p-values are < 0.05, relationships,
including weak ones, are significant.
The strongest control on glacier drainage density is identified to be glacier mean elevation ($\rho = -0.66$, $p = \leq 0.001$),
with higher drainage densities present when a larger portion of glacier mass exists at lower elevations (Fig. 6).
This is followed by glacier mean slope ($\rho = -0.46$, $p = \leq 0.001$), with higher drainage densities on glaciers
characterised by a lower mean slope. This is consistent with Fig 7e, with the highest drainage densities observed
at glaciers with very low slope angles (e.g., the Upper Theodul Glacier; Fig. 4d).



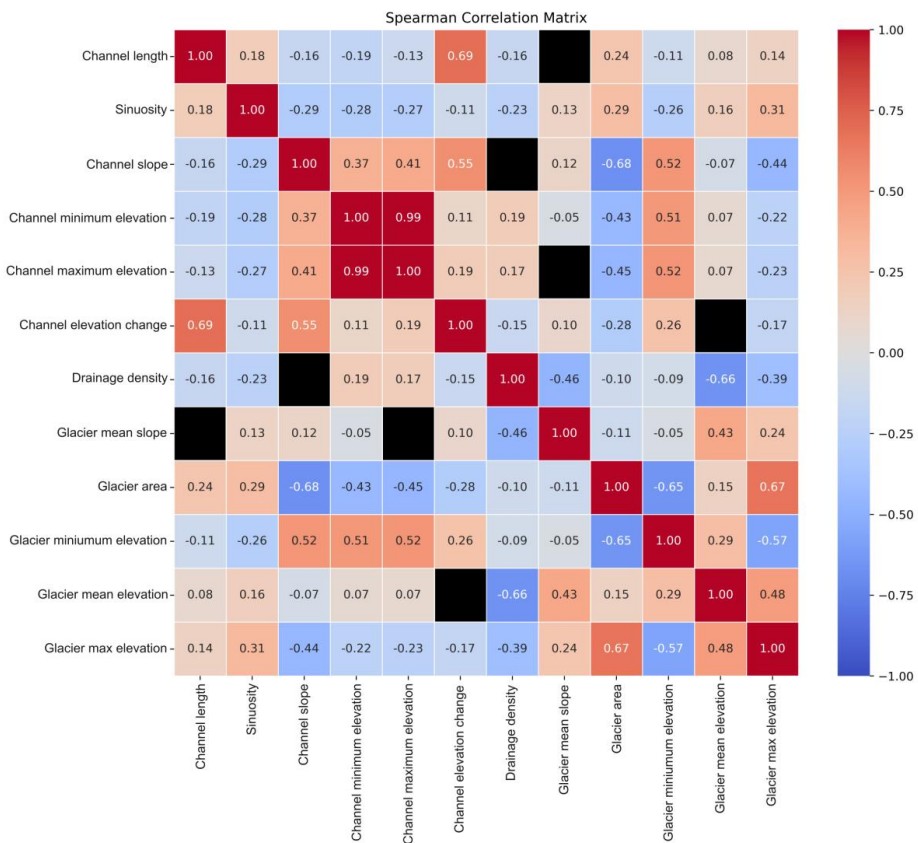


**Figure 6: A heatmap matrix of Spearman's rank correlation to show the relationship between glacier and**
**channel characteristics. Correlation values are scaled along a colour ramp and non-significant relationships**
**(p) >0.05 are coloured black.**


By comparison, channel morphology is characterised by more complex and weaker relationships between
variables. For example, high channel sinuosity can in part be explained by multiple weak correlations: sinuosity
values tend to increase with a decrease in channel slope ($\rho$ = -0.29, $p \leq 0.001$), and the most sinuous channels
occur on larger glaciers ($\rho$ = 0.29, $p \leq 0.001$) with lower minimum elevations ($\rho$ = -0.26, $p \leq 0.001$) (Fig. 6).
Lower relief surfaces are often found when valley glaciers extend down-valley to flatter terrain, supported by
observations of highly sinuous channels on low slope glacier tongues (e.g., Gorner Glacier) (Fig. 4e). The channel
slope is primarily controlled by glacier characteristics, with higher slope channels mostly existing on smaller
glaciers ($\rho$ = -0.68, $p \leq 0.05$), which terminate at higher elevations ($\rho$ = 0.52, $p \leq 0.05$), meaning the steepest
channels are likely to be found at high elevation cirques and hanging glaciers.

To assess the relationship between variables and determine the drivers of variance, we conducted a Principal
Component Analysis (PCA) (see Appendix 1). The PCA loadings show that glacier area has a large negative



loading on principle component 1, closely followed by strong positive loadings from minimum glacier elevation
and channel elevation (maximum and minimum). By comparison, principal component 2 shows a strong positive
loading from drainage density, and large negative loadings from mean glacier slope and glacier mean elevation.
The first two components explain 50 % of the variance within the dataset, with an additional 13 %, 12 % and 9 %
explained by principal components 3, 4 and 5, respectively, which together explain 84 % of the variability. Given
the complexity of the dataset, our cluster analysis reveals no clear clustering of data, but the PCA loadings show
an expected relationship between elevation variables and slope variables, with variables such as drainage density
not closely related to a singular other variable. Overall, our PCA analysis reveals no singular driver of variance,
instead, it is apparent that there is a complex, yet interlinked relationship between all variables that explain the
distribution and appearance of supraglacial channels.

**5    Discussion**
**5.1    Controls on the spatial distribution of channels**
Out of 285 glaciers examined within our study area, we find that 85 contain channels (>0.5 m) using high-
resolution imagery (0.15 m) from mid-July 2020. The presence of channels above our mapping threshold (>0.5 m
wide) is primarily controlled by a combination of sufficient meltwater supply and distance for meltwater to
coalesce and incise. Hence, in mountain glacier environments, larger channels are infrequently detected on cirque
glaciers due to their smaller ablation area, steeper and often crevassed slopes, and limited distance for meltwater
to coalesce. By comparison, all glaciers in Valais larger than 5.6 km$^2$ supported channels, but there is a large
variation in drainage density. This variation is in part attributed to glacier slope, which together with ice flow
velocity, governs the crevassed area of a glacier, restricting the area in which channels can form. Crevasses can
either intercept meltwater, transporting water englacially and inhibiting the formation of longer channels or instead
can route water along a trace crevasse as part of a channel depending on crevasse depth. Additionally, channel
formation is also governed by glacier hypsometry, with glaciers characterised by bottom-heavy hypsometries more
likely to produce higher drainage densities due to a larger ablation surface area. Figure 7 shows a schematic of
how the different elevation and hypsometry of glaciers is likely to influence the distribution and density of
supraglacial channels in the study area.
Whilst valley glaciers might extend down to low elevations, they often do not have the highest drainage densities
because channel formation may be restricted by small valley widths and higher rates of surface lowering (Fig. 7).
For example, the highest drainage densities occur on the Upper Theodul Glacier, which has a low relief slope, is
situated on a plateau above the valley floor, and has a wide ablation area (Fig. 4a). It is possible that because the
Upper Theodul Glacier is situated on a higher elevation plateau, air temperatures at the terminus are likely to be
cooler than at the terminus of neighbouring glaciers which extend further down valley. As a result, the rate of
surface lowering is likely lower, meaning that lower rates of incision are needed for channel formation to keep
apace with surface lowering, resulting in a higher drainage density (Rippin et al., 2015; Pitcher & Smith, 2019).
However, summer temperatures in the Alps are increasing (Sommer et al., 2020), which will result in higher rates
of surface lowering, meaning that glaciers extending to lower elevations will require increased channel incision
to counteract the higher rates of surface lowering in order for channel formation.






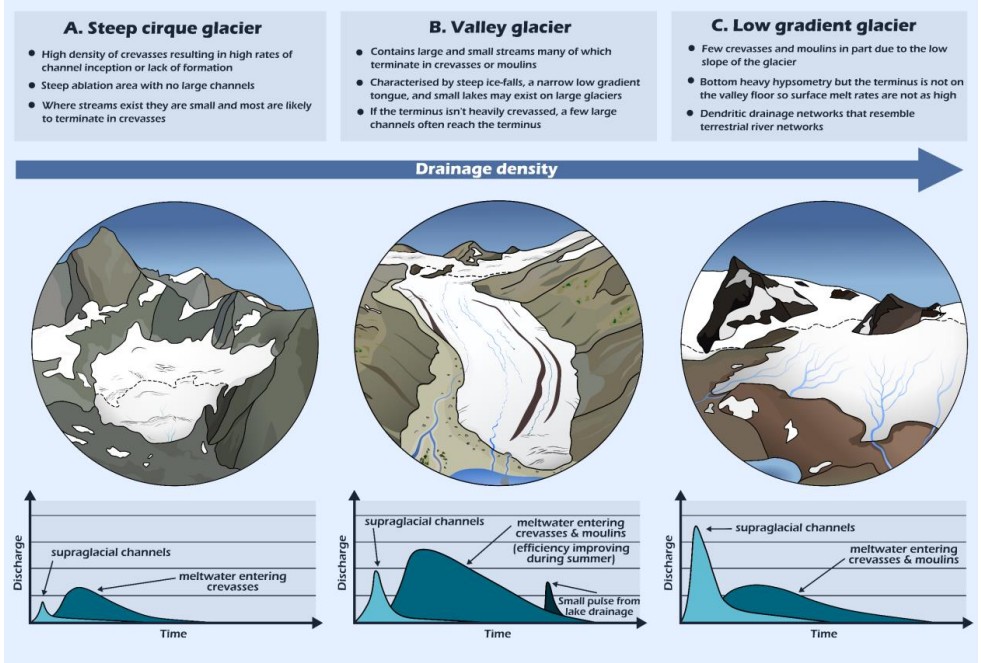


**Figure 7: A schematic depicting the range of glacier types (A-C) and their respective characteristics, increasing in drainage density from left to right. Each glacier type corresponds to a predicted hydrograph (bottom) depicting changes in proglacial channel discharge as a response to surface melt. Light blue shading shows the conceptual hydrograph if all of the meltwater were to be transported via supraglacial channels, whereas those shown in mid-blue show the hydrograph where the bulk of meltwater is transported englacially/subglacially. Dark blue in panel (b) shows a lake drainage event.**

380

The distribution/density of channels on different types of glaciers is also likely to impact on the runoff hydrograph (Fig. 7). Small cirque glaciers (Fig. 7A) are typically characterised by steep, heavily crevassed slopes, meaning meltwater is more likely to be captured by crevasses, and there is limited distance for meltwater to coalesce and form channels. Hence, run-off at cirque glaciers in response to surface melt is likely characterised by a small earlier peak from a few supraglacial channels (light blue shading in Fig. 7A) or a slightly delayed peaks as crevasses capture meltwater and route it through the en- (and -sub) glacial system (dark blue shading in Fig. 7A). By comparison, valley glaciers are frequently characterised by larger non-crevassed zones, meaning more meltwater reaches the terminus supraglacially, but channels are often intercepted by crevassed zones or ice falls. This results in an initial peak from supraglacial channels, followed by a delay in meltwater routed en- (and sub-) glacially (Fig. 7B), with a progressively reduced lag time throughout the melt season due to increased subglacial drainage network efficiency (Nienow et al., 1998). Additionally, some larger valley glaciers contain small lakes (e.g., Gorner Glacier and Aletsch Glacier), which may experience infrequent drainage events, leading to a sudden



peak in proglacial river discharge. Whilst less common in Valais, glaciers characterised by large low relief ablation
areas often contain large supraglacial drainage networks that capture the majority of surface melt due to having
smaller and fewer crevasses than most valley glaciers (Fig. 7C). They will tend to have the 'flashiest' hydrograph
because the supraglacial drainage network rapidly transfers melt off the glacier surface. Where a small number of
crevasses or moulins intercept this drainage, it leads to a more lagged hydrograph response and with a peak well
below the hydrograph from supraglacial channel drainage.
Our dataset provides new insight into meltwater transport across a large range of glaciers, allowing simple
inferences to be made about connectivity and lag times. Overall, we find that 72 % of our mapped main stem
segments route meltwater to en- or subglacial positions and 25 % run off at the glacier margin. However, this
varies between glaciers and in the case of the largest glacier, the Aletsch Glacier, no mapped channels reach the
terminus, attributed to the location of highly crevassed zones, forcing meltwater into the glacier (53 % of channels
enter moulins and 36 % enter crevasses). Unlike most glaciers in the Alps, small supraglacial lakes are also present
on the Aletsch glacier, which capture 11 % of channels. By comparison, the Upper Theodul glacier, which exhibits
the highest drainage density within our dataset, contains almost no moulins (2.8 % of channel termination) and
27 % of channels reach the terminus or periphery, compared to 0 % at the Aletsch glacier. A large number of
channels terminate in crevasses on the Upper Theodul Glacier (70 %), but crevasses tend to be small (<0.3 m
wide) and may not route the meltwater en- or sub-glacially but rather may act as part of the channel network and
are mapped as individual segments as they may not be continuous. Instead, crevasses may fill with meltwater and
be overtopped or route meltwater perpendicular to the ice flow direction (Fig. 8). We attribute the difference in
drainage pathways to the low slope of the Upper Theodul Glacier, its smaller glacier area, and the lack of high-
density crevasse fields or ice falls, meaning meltwater is more likely to be transported via channels to the terminus
(e.g., Fig. 7C). This likely results in drastically different lag times within a given day between the two glaciers,
with the Upper Theodul likely experiencing an initial peak from run-off at the terminus, whereas at the Aletsch
Glacier the delay between surface melt and proglacial discharge will be larger due to longer pathways and the
potential for supraglacial and subglacial storage (Hock et al., 1999) (e.g., Fig. 7B). Additionally, at Aletsch Glacier
the high portion of moulin and crevasse terminating channels likely contributes to the seasonal speed-up in ice
flow velocity when meltwater reaches the bed (Leinss and Bernhard, 2021). By comparison, the supraglacial and
subglacial system are anticipated to be less coupled at the Upper Theodul Glacier. Hence, the characteristics of
these glaciers (e.g., heavily crevassed areas, presence of moulins) and the channel terminal positions may help
provide simple insight into surface-to-bed connectivity but would need to be verified by fieldwork. Understanding
these variations in meltwater routing is important because increased interaction between meltwater and the glacier
bed can impact subglacial erosion rates and sediment transport out of glacial systems, which in regions such as
the Alps can directly impact agriculture and hydropower infrastructure. For example, Delaney et al. (2020) found
that increased meltwater at the bed is likely to increase sediment discharge despite a reduction in ice flow velocity
under climatic warming and resulting glacier thinning. However, freshwater flux from supraglacial channels that
terminate in lateral or proglacial positions may dilute sediment loads.



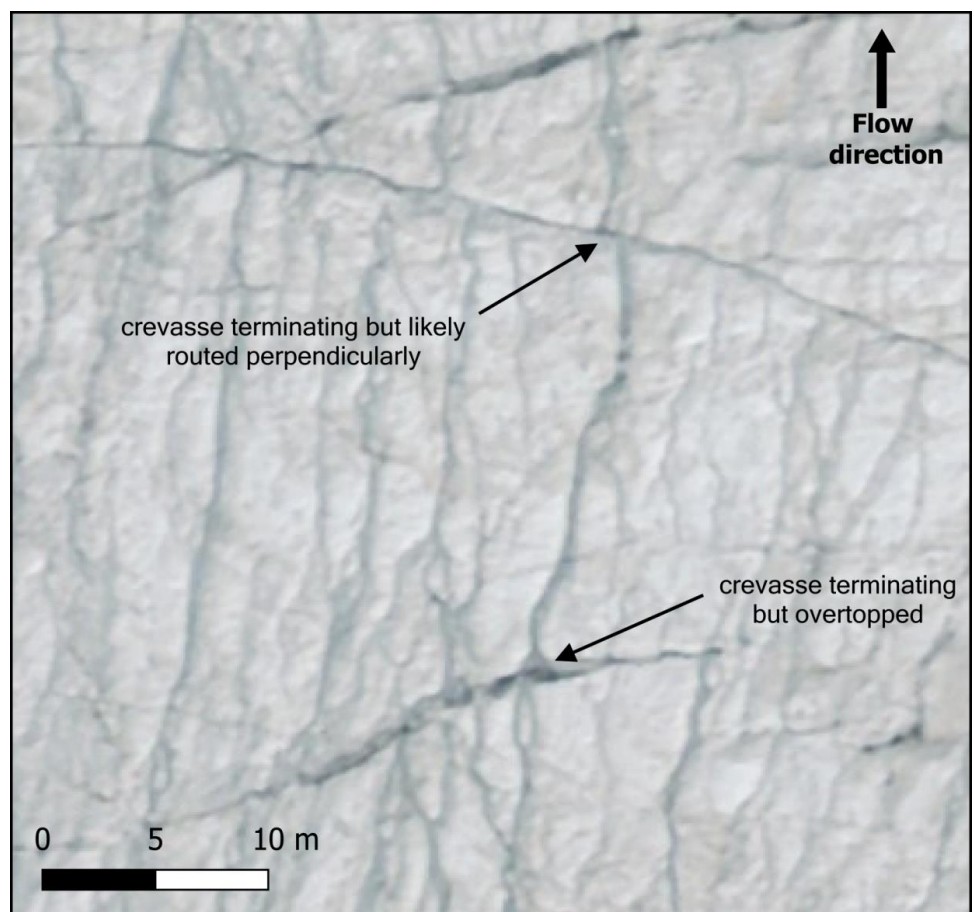


**Figure 8: Examples of channels terminating in crevasses on the Upper Theodul Glacier. Channels are shown**

**to terminate in crevasses but often continue directly down-glacier as some meltwater overtops the crevasse.**

**Imagery source: Federal Office of Topography swisstopo.**


### 5.2    Controls on channel morphology

In Valais Canton, we observe a broad range of channel characteristics (e.g., Fig. 4), with sinuosity of particular
interest due to the similarities with alluvial fluvial systems. However, the mechanisms of meandering in ice-walled
channels are less likely to result from the erosion and deposition of sediment which occurs in fluvial systems
(Knighton, 1972). Instead, in supraglacial environments, meander evolution is thought to occur by a combination
of thermal erosion and downcutting, with incision rates tending to be highest on steeper slopes (Gulley et al.,
2009). The role of glacier slope in controlling channel sinuosity is particularly evident in Valais (Fig. 5a),
demonstrating a clear upper limit where highly sinuous channels are unlikely to occur on higher relief slopes (e.g.,
over 20°). However, this figure may also reflect the preferential area for larger channels to form (i.e., flatter, less
crevassed regions) rather than the direct impact slope has on sinuosity. This may explain the difference in the



influence of slope on sinuosity between Valais and in the high Arctic, where St Germain & Moorman (2019) attribute higher channel sinuosity to higher relief slopes and increased channel discharge. As cold-based glaciers tend to be less crevassed and dynamic, the viable area for channel formation (i.e., less highly crevassed zones) is likely to be higher per glacier than in Valais. It is anticipated that the role of discharge remains similar in both environments, as our observations find that larger channels (higher discharge) tend to be more sinuous (e.g., Fig. 4d-e). The sinuous channels observed by St Germain & Moorman (2019) are typically highly incised, meaning they tend to persist inter-annually. However, in our study area, the rates of surface lowering are likely to be much higher, meaning a larger proprtion of channels might melt out/disappear if their rate of incision is less than the rate of surface lowering. The exception being large highly sinuous channels, which are likely to persist longer term due to their higher discharge and subsequent increased rates of incision.

Compared to bare-ice glaciers, supraglacial channels on debris-covered glaciers are commonly highly incised, with channel pathways influenced by local-scale glacier surface topography (e.g., medial moraines). For example, we observe some cases where less sinuous channels are funnelled through thin areas of exposed bare ice, until reaching continuous debris-cover, where large meandering planforms begin to develop (e.g., Fig. 4d). In addition, there is a general absence of small channels on debris-covered glaciers in Valais due to slow and dispersed meltwater transport through the debris matrix (Fyffe et al., 2019), with predominantly incised canyon-like channels present. Unlike debris-free glaciers, incised channels in debris-cover form exposed banks of bare ice which are too steep to harbour debris and are distinct from the thick debris covered surface. Whilst the melt rates are much lower beneath continuous debris cover (Kneib et al., 2023; Fyffe et al., 2019), once bare ice is exposed by channel incision, meltwater production is increased due to exposure of non-insulated ice (Mölg et al., 2019), with increased discharge likely furthering channel development.

Additionally, increased sediment presence within channels, which would be expected on highly debris-covered glaciers, can increase the water temperature due to inhibiting heat exchange between the meltwater and the channel wall (Isenko et al., 2005). The combination of higher water temperatures and debris-covered glaciers often means that most of their surface melt is concentrated in a few channels (i.e., higher channel discharge) and likely results in very high rates of channel incision, producing these high amplitude meanders that are not observed on bare ice glaciers in Valais. However, the role of debris in supraglacial channels is poorly understood as most supraglacial research has occurred on ice sheets, where areas of lower albedo are controlled by the presence of, dust, black carbon, algae, and cryoconite deposits (Ryan et al., 2018; Leidman et al., 2021; Khan et al., 2023).

Further insight may be gained from bedrock rivers, which contain a mixture of sediment cover and exposed bedrock. In bedrock rivers, sediment was found to increase channel erosion rates and increase sinuosity, but high sediment supply may reduce erosion due to producing sediment cover which protects the bed from erosion (Moore, 1926; Shepherd, 1972; Turowski, 2018). Ultimately, future work is needed to determine the role of debris in supraglacial systems, specifically in situ measurements of debris content in channels to determine how it affects channel properties.



### 5.3 Comparison between mountain glaciers and ice sheets

Supraglacial channels are an important component of glacier hydrology in both mountain glacier and ice sheet settings, yet some obvious differences exist between the two environments. The difference in scale means that in an ice sheet setting, particularly at less crevassed higher elevations, there is sufficient distance for meltwater to incise and channels to coalesce forming established drainage networks, which broadly follow Horton's laws i.e., mean river length increases with channel order and mean slope decreases with channel order (Yang et al., 2019). In contrast, whilst interconnected dendritic networks do occur on mountain glaciers (e.g., Fig. 5a), they may not be present on smaller glaciers because sufficient distance for channels to coalesce is not available. Additionally, mountain glacier environments differ from ice sheet settings due to small scale (e.g., surface debris cover) and large scale (e.g., increased frequency of moraines) topographic heterogeneity. The increased presence of debris on mountain glaciers results in uneven surface albedo, increased surface roughness (Rippin et al., 2015), and additional structural controls on channel formation. For example, large channels often form parallel to medial moraines due to topographic confinement and increased melt in proximity to the debris cover. Drainage network formation is also confined to an often-narrow glacier tongue, whereas channel formation on an ice sheet is often unconfined. On a large scale, meltwater routing in both mountain and ice sheet environments is largely influenced by ice elevation. However, on a smaller scale ice surface structures (e.g., trace crevasses and flow-stripes) exert a strong control meltwater routing, meaning that channels are not always oriented in the same direction (e.g., Figs. 5c & 9; Chen et al., in press). The same controls on distribution likely exist in both environments, but the addition of a wider range of variables (e.g., often higher debris cover) in mountain glacier environments makes predicting flow routing more complex.

When comparing mountain glacier systems to the SW GrIS (e.g., Smith et al., 2015; Karlstrom & Yang, 2016; Yang & Smith, 2016; Yang et al., 2016, 2021, 2022), there are notable differences in coupling between surface melt and the englacial/subglacial system. On the SW GrIS, virtually all higher-order rivers are observed to terminate in moulins (Smith et al., 2015), whereas in Valais 72 % of main stem channels terminate englacially (37 % in crevasses and 35 % in moulins). Additionally, there is a general absence of supraglacial lakes in Valais, except for small lakes on Gorner Glacier and the Aletsch Glacier, whereas they exist in abundance on the GrIS (Chu, 2014). On the GrIS, supraglacial lakes act as temporary storage, with the potential for meltwater to refreeze rather than reaching the terminus each melt season; however, hydro-fracture events can cause rapid lake drainage, which has been linked to ice speed-up events (e.g., Das et al., 2008; Morriss et al., 2013; Chudley et al., 2019). In mountain glacier environments, the general higher surface slopes and absence of temporary supraglacial storage means that proglacial discharge is more clearly linked to rates of surface melt, with very few sudden pulses of increased discharge. The input of surface meltwater to the bed has also been clearly linked to changes in ice flow velocity in mountainous regions (e.g., Willis, 1995; Jobard and Dzikowski, 2006).

### 5.4 Future evolution of supraglacial channel systems

The exact role of climatic warming on supraglacial drainage networks is unknown, but the presence of supraglacial drainage networks at higher elevations can be anticipated in the future due to rising equilibrium lines (Leeson et al., 2015). Whether discharge in current channels will increase or decrease is dependent on the rate of glacier



retreat and summer melt. It is likely that larger glaciers will see an increase in channel discharge due to an increase
in ablation area (St Germain & Moorman, 2019). However, the reduction in area for smaller glaciers may be large
enough to prevent the formation of established drainage networks, and changes in glacier slope may result in a
reconfiguration of the drainage system or reduction in drainage density (Fig. 5e). Additionally, glaciers in
mountain glacier environments are undergoing an increase in debris cover (e.g., Glasser et al., 2016; Fleischer et
al., 2021), and it is not fully understood how changes in debris cover will affect surface meltwater supply and
transport, channel morphology and surface albedo (e.g., Leidman et al., 2021). Future research would also benefit
from the growing repository of high-resolution orthophoto surveys to inform our understanding of supraglacial
hydrology outside of ice sheet settings, particularly concerning seasonal and interannual channel evolution to
better inform modelling of glacier hydrology. It is not currently known how widely applicable our research is to
regions with larger glaciers and lower rates of surface lowering (e.g., the Arctic) so future studies may benefit
from assessing their similarity.

**6     Conclusion**
This study presents the first comprehensive dataset of the distribution and characteristics of supraglacial channels
at a regional scale in a mountain glacier environment. We mapped 1890 channel segments (>0.5 m wide) on 85
glaciers found to contain channels above our mapping resolution (~0.5 m wide) in Valais Canton, Switzerland,
out of a total sample of 285 glaciers. We found large variability in channel drainage densities across glaciers, with
the highest drainage densities existing on glaciers characterised by lower relief slopes, with a large portion of their
mass at lower elevations (Fig. 7). We find that the presence of channels is primarily dictated by a sufficiently large
supply of meltwater (i.e., large enough glacier area) and uninterrupted distance for meltwater to coalesce (i.e.,
absence of crevasses and low ice surface slopes). The primary control on channel distribution is surface
topography, with the slope and size of the ablation area providing a clear limit on where channels can form and
their length (Fig. 5b). However, strong structural controls on channel distribution exist (e.g., crevasse routing and
medial moraines).
The majority of glaciers are characterised by channels that reach the glacier margin, with the percentage of
channels terminating in each position averaged by glacier revealing that on average 80 % of channels run directly
of the glacier, and 20 % terminate englacially. Overall, 48 % of glaciers contain no englacially terminating
channels, compared to only 3.5 % of glaciers where all channels terminate englacially. The variation in where
channels are located and the variation in their terminus environment (moulin/crevasse versus running off the
glacier) may result in a range of lag times in proglacial discharge in response to surface melt, with different glacier
geometries likely predictive of glacier drainage density and channel pathways (Fig. 7). This differs from typical
ice sheet settings where very few supraglacial channels reach the ice margin, with most surface melt transported
englacially through crevasses or moulins. In comparison to ice sheets, channel drainage networks in mountain
glacier environments are often less established due to glaciers having smaller drainage areas and there being less
distance for channels to coalesce, with valley glaciers often narrowing down-glacier reducing the possible size of
drainage networks. Channels in Valais are typically characterised by low sinuosity (Fig. 5a), with a general
absence of the canyon-like channels described on cold or polythermal glaciers. However, some highly sinuous





channels exist, particularly on debris-covered glaciers characterised by channels with steep ice cliffs and on low
slope termini, with sinuous channels most commonly occurring where channels are also deeply incised.  Lastly,
further in-situ measurements would be beneficial to determine whether the channels delineated as part of this
study represent the majority of meltwater transport on mountain glaciers, or whether channels below our mapping
resolution also play a key role in meltwater transport.

**7    Appendix**
**Appendix 1:** Principal Component Analysis
**Table A1: The eigenvectors for principal components 1 to 6 from a PCA analysis of glacier and channel**
**characteristics, with PC1 being the most significant. The three largest loadings for each principal**
**component are in bold.**

|  | PC1 | PC2 | PC3 | PC4 | PC5 | PC6 |
|---|---|---|---|---|---|---|
| Glacier area | **-0.439** | 0.042 | 0.034 | 0.288 | -0.187 | 0.054 |
| Glacier minimum elevation | **0.435** | -0.112 | 0.085 | -0.098 | **0.407** | -0.178 |
| Channel maximum elevation | **0.393** | 0.159 | 0.066 | **0.436** | -0.115 | 0.186 |
| Channel length | -0.070 | -0.264 | **-0.582** | 0.312 | 0.195 | **-0.199** |
| Channel minimum elevation | 0.382 | 0.196 | 0.121 | **0.423** | -0.100 | 0.196 |
| Channel sinuosity | -0.117 | -0.182 | -0.122 | -0.041 | **0.468** | **0.845** |
| Channel elevation range | 0.166 | -0.345 | **-0.542** | 0.198 | -0.173 | -0.068 |
| Channel slope | 0.320 | -0.203 | -0.066 | -0.115 | **-0.483** | **0.255** |
| Drainage density | 0.131 | **0.543** | -0.268 | -0.009 | 0.041 | 0.083 |
| Mean glacier slope | 0.119 | **-0.439** | 0.136 | -0.309 | -0.335 | 0.141 |
| Glacier mean elevation | 0.170 | **-0.376** | **0.401** | 0.280 | 0.339 | -0.188 |
| Glacier max elevation | -0.330 | -0.175 | 0.264 | **0.462** | -0.163 | 0.086 |




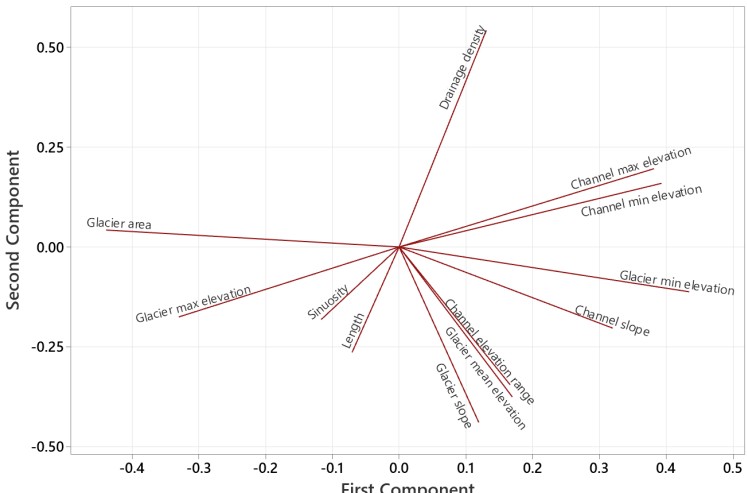

**Figure A1: loading plot for principal components 1 and 2 from a PCA analysis of glacier and channel**
**characteristics.**

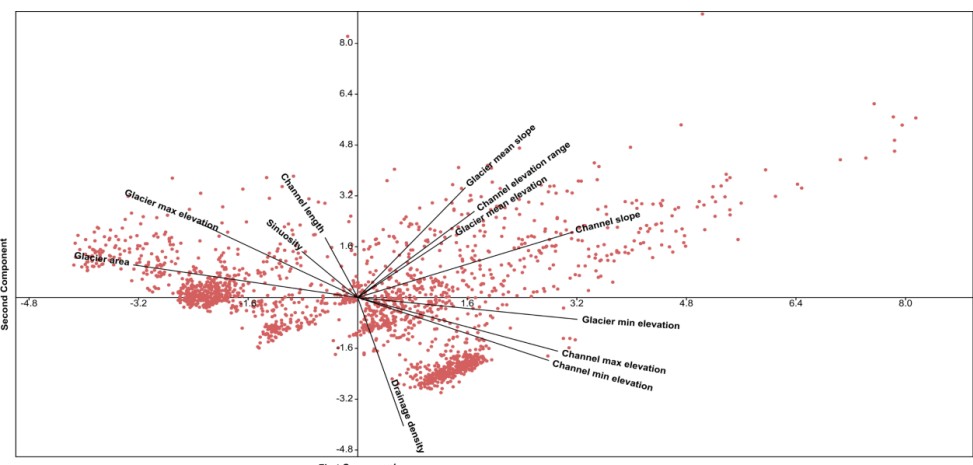

**Figure A2: A biplot from a PCA analysis showing principal components 1 and 2, which was used for cluster**
**analysis and overlayed with a loading plot (see Fig. A1).**

**Data availability**
The orthophoto and DEM data used in this study are freely available on the SwissTopo website
(https://www.swisstopo.admin.ch/). Outlines of glaciers in Valais were obtained from Glacier Monitoring in
Switzerland (GLAMOS) and are available online (https://glamos.ch/). The produced supraglacial channel data is
available       upon       request       from       the       corresponding       author       (Holly       Wytiahlowsky:
holly.e.wytiahlowsky@durham.ac.uk).



**Author contributions**

All authors contributed to the conceptualisation of this project. HW conducted the mapping, formal analysis and data visualisation under the supervision of CRS, RAH, CCC and SSRJ. HW led manuscript writing with comments and revisions provided by all authors.

**Competing interests**

One author is a member of the editorial board of The Cryosphere.

**Acknowledgements**

This work was supported by the Natural Environment Research Council via an IAPETUS2 PhD studentship held by Holly Wytiahlowsky (grant reference NE/S007431/1). We thank SwissTopo for making their orthophotos and DEM data open access.

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
