# Peer review of "Distribution and characteristics of supraglacial channels on"

_EGUsphere, 2024_

## Author Comment (AC1)

Response to Reviewer 1

Wytiahlowsky et al. present an extensive analysis of supraglacial channel characteristics for 285 glaciers within the Calais Canton, Switzerland. They manually delineated channels from high-resolution (0.15 m) orthophotos and use these delineations to perform their analysis which included investigating channel characteristics and its relationship to glacier properties. I found the manuscript to be well written and in general easy to follow, with clear and appropriate methodology to support the results and conclusions presented in the manuscript. While supraglacial channel analysis is not necessarily new, the number of channels and individual glaciers characterized in this study is impressive and allows for the understanding of supraglacial hydrology on a regional scale, which is very exciting for the field of glacial hydrology. The results of this work contribute to furthering the scientific understanding of supraglacial hydrology on alpine glaciers while also acting as a point of comparison for future studies of supraglacial channel formation in other alpine environments. My opinion is that this manuscript is suitable for publication in The Cryosphere. Below I have included some general and specific comments for the authors to review. We thank the reviewer for their constructive and encouraging comments and are delighted that they deem our manuscript suitable for publication. We address each point in turn below, with our responses indicated in blue text.

General Comments

-Figure 1: The manuscript would benefit from a overview of the glaciers within the Valais Canton. Currently, a lot of space is used to show the region that is not the study area. I would suggest to highlight the values area (as in panel a) but within a smaller area, then use a majority of the space to delineate and show all of the glaciers in the region (currently indicated by blue). A full view of all the glaciers in the area would allow the authors to then color-code them as (1) glaciers too small to be considered in this study <1km^2 (maybe white or gray to denote they are being excluded or alternatively leave them out due to the small size if they are not visible at the scale chosen), (2) glaciers without channels, and (3) glaciers with channels. Panel B does a good job in showing the large Alestsch glacier, however the debris cover is hard to see in panel C. This may be an artifact of the low-resolution of background imagery for figure 1 in the version of the manuscript submitted for review (I understand .png are often submitted early whereas .pdfs are instead used in the final version, if this is the case this last point can be ignored and I will assume the debris cover will be more apparent in the final version). Finally, if possible under the space limitations, it would be nice if some of the glaciers referenced in subsequent figures could be indicated in some way within Figure 1. This would make it easier to put specific glaciers into context when they are directly named later on in the manuscript. Amended – We have zoomed into Valais Canton on panel A and have colour coded the glaciers to show those with streams (>0.5 m), those without visible streams, those that are snow covered (omitted), and glaciers <0.1 km$^2$ (omitted). The imagery source has been updated for panels B and C which has increased the quality of the imagery.

-Figure 4. This is a great figure of the delineated channels, however, it would benefit from the inclusion of insets (or sub panels) that include the entire glacier overlaid elevation contours and annotated with the region with channels shown. By including both a glacier-wide view would help top put the streams into the glacier-wide context which is the focus of this work. Amended – We have added sub panels for each panel that show the location of the main panel on the glacier.

I am curious if glacier aspect has a relationship with channel formation. We have added in a sentence in section 4.3 to note that no relationship exists between aspect and drainage density (Kruskal-Wallis test: p = 0.61).

In referring to the two types of crevasses you see (ones that transport meltwater englacially vs. ones that redirect stream flow). The authors state that the crevasse fills with meltwater and overspills along the crevasse trace. This may be true as is seen in Greenland (e.g., Chudley et al., XXXX) but alternatively, the crevasses may have already advected shut, these old crevasses may then act as a preferential flow path as observed in Fig 8 (I see this frequently in Greenland). I would suggest elaborating on this process in the discussion, as a natural question that would arise would be why is the crevasse not hydrofracturing if it completely full of meltwater. While the answer would be the stress regime in such a case (it would suggest the crevasse is old, having advected into a compressional region). Altogether I feel this process deserves some added clarity within the discussion. Amended – We split the paragraph beginning on line 399 in the original version of the manuscript to enable us to expand on the influence of crevasses on meltwater routing.

Specific Comments

Comments regarding figures:

Fig 1: expand the description of the yellow start to include that it is a weather station. Amended – The legend has been relabelled to 'Weather station' and the station name is given in the methods.

Fig 2: Do the arrows indicate ice flow or water flow or both? Stating this in the caption would help clarify. Amended.

Fig 3: The x-axis labels are hard to see, I think all the font sizes in this figure need to be enlarged. The inclusion of the greater than/less than symbols in the x tick marks clutters the text making it even harder to read. I suggest instead including this information in the legend or using bracket notation. Amended – The font size has been increased, and we have changed the location of the x axis ticks to represent the range shown by each bar. We have clarified where we use equal to/greater than in the figure caption.

Figs 3 and 5: I am confused as to what the different colors represent, are they supposed to correspond to something? All of the different colors when one variable is being plotted is confusing. Particularly because different colors are being used in different figures. Note that this comment is only referring to Fig 3c-f, and Fig 5a,b,e-g. Amended – Plots that only show a single variable are now all the same colour.

Fig 5. Including an annotation to each of the subplots with the results from the spearman's rank analysis would be helpful here. Currently the reader has to go back and forth between figures 5 and 6 to interpret the data presented. The x-axis labels in 5d are very hard to read, I would suggest making these larger. Also, the outliers in panels c and d makes it difficult to see the differences between the box and whisker plots. Maybe consider excluding the outliers or a $\rho$ djusting your subplot size/spacing so that the distribution is more clear, particularly in 4c and 5d. Amended – Spearman's correlation values have been added to Figure 5, and the y axis limit has been modified for the box plots.

Fi 7. This is a great figure, however, most of the the text in this figure is too small to read. Also, consider either making the ela dashed line bold or a different color (or both so it is easier to see). Amended – The text size has been increased, and the ELA line thickness has been doubled.

Fig 8: I would suggest adding arrows to indicate meltwater flow direction. Amended – Arrows have been added that indicate the meltwater flow direction, and this has been added into the caption.

Fig A2: all labels are too small to see. Amended.

Manuscript comments:

L63: Here you site WorldView-3 with a ~3.7m resolution, however, this is the short-wave infrared resolution, WorldView-3 has a 31cm panchromatic and a 1.24 m multispectral. Amended.

L127-137: There is a lot of information and numbers in this paragraph that becomes hard to follow, consider removing unnecessary information or making a figure or a table to accompany this information. Amended – We have edited this paragraph to remove some of the unnecessary detail.

L141-142: Explicitly state the channel width the method was successful on and then the channel size in this study area. Amended – we have added additional clarification.

L143: define the abbreviation NDWI_ice before use. Amended.

144: state that you are referring to multispectral here, WV-2 has a 46cm panchromatic resolution. Amended.

Paragraph starting on L332: Reference specific tables and figures within the appendix rather than the appendix as a whole. Amended.

L354: change "…crevassed area of a glacier, restricting the area" to "…crevassed area of a glacier which restricts the area…". As phrased it is unclear if your last clause is referencing the preceding clause or entirety of the sentence (which would not make sense), so I suggest removing the final clause for clarity. Amended.

L356: the end of this sentence is confusing as written. I suggest rephrasing the sentence to read something like "When crevasses are open they can intercept meltwater and…, crevasses that have closed modify small-scale surface topography and meltwater can be routed along the crevasse trace…" or something like that. Amended.

L357: The term "bottom-heavy hypsometries". Amended – Reworded to "a larger portion of their mass at lower elevations".

L316: I think this should be referencing Figure 5e. Amended.

L317: This figure reference may also be incorrect. Amended – this should read Fig. 4a.

L371: change to "in order for channels to form". Amended.

L381: change "/" to "and", also remove "on" in the phrase "on the runoff hydrograph". Amended.

L390: specify lag time here, a lag between what and what? Melt and peak proglacial discharge? If so, state that here. Amended – this has been changed to reflect that it refers to the lag time between melt and peak proglacial discharge.

L393: comma between large and low. Amended.

L394: Break into two sentences, ending the first at the word "melt". Amended.

L400: Define lag times here as well. Amended.

L400: Here you refer to "main stem segments", but earlier in the text you referred to stream order. I suggest being consistent with terminology. Amended.

L451: change "/" to "or". Amended.

L465: change "additionally" to "The". Amended.

L467: do you mean on instead of "and" here? Amended.

L470-1: I am not sure if this is true, most debris-covered research has been done in the Himalaya with some in Alaska (e.g., all work by Doug Benn and many others). Amended – The wording has been edited to emphasise that whilst research on debris-covered glaicers has documented and discussed supraglacial channels, little work has looked at the role of sediment within supraglacial channels.

L496: add "on" before meltwater routing. Amended.

L509: mountain glacier environments have glacier lake outburst floods which could cause abrupt acceleration, I would mention this here even though you do not see them in your area. Alternatively, refer to your specific area rather than "mountain glacier environments" more broadly. Amended.

---

## Author Comment (AC2)

Response to Reviewer 2

This is an interesting and well put-together study. It presents some interesting insights into supraglacial drainage networks on a large number of Swiss glaciers, which must be one of the first such studies that takes advantage of modern remote sensing/photogrammetric approaches. Supraglacial drainage networks on the Greenland Ice Sheet are a popular focus of research, but such networks on valley glaciers are understudied. I thus welcome the efforts of this paper. I would also like to congratulate the authors on their attention to detail. I think I came across just one typo throughout, and the formatting, presentation, writing style and all other such aspects are top class. I very much appreciate the care taken to ensure this. It makes the job of reviewers much easier, and indeed makes the work far more accessible to a readership. We thank the reviewer for highlighting the value of our manuscript, and we are grateful for their positive comments on its presentation. We appreciate the feedback and address each point in turn below, with our responses indicated in blue text.

Whilst I see value in the paper and welcome the efforts of the authors, I do have some concerns as follows:

1) One of the key issues I have is that I am not 100% certain of the broader aims of the work. The authors very clearly state on lines 91-92 that our 'aim is to characterise the morphometry of supraglacial channels on mountain glaciers, providing insight into where and why they form'. Whilst this is clear, I would like the authors to go further, and say why this matters – why, glaciologically speaking, should we be doing this? Why is it important for us to gain these new insights? I suspect the authors can address this relatively easily, but I would like to see clear statement as to the value of this work. It's worth saying that I think a lot of text that could contribute to such a statement already exists in the Introduction, so it's just a matter of coalescing the key points. Amended – We have added an additional statement of the importance of supraglacial channels, after the aims. In the first paragraph of the introduction we have added in information about the importance of supraglacial channels that route meltwater to the bed, and the implications that has for ice motion.

2) Perhaps my main concerns are about the subjectivity of the mapping of channels, and the assumption that not seeing channels of a certain size means there are no channels. I don't believe this is correct. I believe that the approach is more about resolution rather than presence. I go into some detail in my line-by-line comments below, and indeed the authors themselves actually raise this in the very last sentence of the paper! However, I believe this deserves more attention. Please see my comments below.

Mapping accuracy:

We acknowledge that there is a subjective element to channel delineation and have now tested the accuracy of the mapping by repeat mapping of the Rhone Glacier (Fig. 2a) which contains >100 streams (>0.5 m). We find a difference in drainage density of 2.6% and a 0.21% decrease in total channel length which we deem acceptable for providing a good representation of glacier drainage density, and have added this into section 3.1 in the methods. In addition, we have added an additional figure to the appendix that shows a comparison of our repeat mapping. The primary source of error was found to be determining where to stop mapping up-channel, which we explain in the methods. However, we aimed to take a conservative approach in when to stop making to reduce our chances of overinterpreting channel pathways.

Mapping resolution:

It is outlined throughout the manuscript that we only delineate channels that are around >0.5 m wide as we cannot clearly delineate channels below this resolution. We have amended the manuscript to clearly state 'no visible channels' when referring to glaciers where channels are not identifiable below

our resolution. We have also added our justification for not mapping or quantifying small channel networks, which is simply that we cannot do this reliably enough for each glacier. We could perhaps denote the presence of channels below our mapping resolution for our study area, however it is likely that these dense networks are only visible on large glaciers where there is more surface melt. Hence, we do not do this because it raises the same issue identified by the reviewer about our mapping threshold. Unfortunately, there is no consistent way for us to map the accurate distribution of all channels on every glacier in Valais, but we believe that the most consistent way for us to do this is to choose a clear threshold that we can confidently map above, whilst acknowledging that this may create a bias towards larger glaciers (section 5.1). We too agree that these smaller channels may be important, but they are simply outside of the scope of and data products available for this current paper. Hence, we our focus is on larger channels that carry the bulk of the meltwater.

3) Another key concern is about the timing of the analysis. The imagery for the analysis was gathered in the early part of the summer. Lower altitude glaciers in early summer will have more melt and thus more channels than those at higher altitude. This could be the main or even sole driver of the differences observed in channel density. I believe it is dangerous to use a single snapshot in time, particularly when it is this early in the melt season, to assess overall drainage density. I appreciate that the authors are constrained by data availability, but this is an important point that needs careful consideration.

We agree with the reviewer that imagery obtained at the end of the melt season would be preferable, but we are limited by the fact that the orthophotos were only available in mid-July. We have since quantified snow-cover at different elevations to reinforce the point that there are still sufficiently large snow-free areas in mid-July at all elevations. We find on average that 39.6% of each glacier was snow-free, which is slightly higher (45.0%) where a glacier has a lower mean elevation (2500 – 2800 m.a.s.l), although this is not a particularly large difference compared to the snow free area (39.5%) of the highest mean elevation band (>3700). We agree that glacier mean elevation is one of the main drivers of glacier drainage density ($\rho$ = -0.66, Fig. 6). However, this relationship would be anticipated to be a strong control on drainage density regardless of when the imagery was acquired in July or late August, and our imagery does contain snow-free glacier areas at all elevations. We have now added information into the methods to provide better context on the year of imagery acquisition, and have acknowledged the impact of our imagery acquisition timings in the discussion.

I have a significant number of other comments which I relate to specific locations in the text. These are detailed below.

| Line | Issue |
| --- | --- |
| 13 | It seems odd to state that your investigations explore '<2000 supraglacial channels'. This seems rather vague to me. '<2000 could mean anything from 1 to 1999. Can you be a bit more precise? Are you looking at a few tens, a few hundreds or nearly 2000? Amended – We have altered the abstract to reflect the exact number of channels (n = 1890). |
| 13 | I must confess to being quite surprised that only 85 of your studied 285 glaciers have supraglacial channels. What time of year did you carry out your investigations, since I think this would be significant. The orthophoto images are from mid-July 2020 which is mentioned in the methods. The 85 glaciers contain channels above our threshold of 0.5 m at this specific time. However, |

| | |
|---|---|
| | we acknowledge that channel networks become more developed at the end of the melt season, but no orthophotos are available at the end of August or early September. |
| 34 | You state here that glaciers and ice caps 'are anticipated to contribute to sea level rise throughout the 21st century and beyond'. It would be worth stating that they have indeed already been contributing too. Amended. |
| 90 | With reference to my point above, here you mention that you create an inventory of 'almost 2000 supraglacial channels'. This is a much clearer indication of the numbers
involved than the <2000 referred to previously.
We have modified this to be the exact number, in line with the abstract. |
| 162 | I'm pretty astonished by the manual effort that must have gone into mapping these channels. This is an impressive feat, so very well done! However, I have a couple of concerns. Firstly, how did you assess and quantify the accuracy of your manual approach? The channels you indicate as being mapped in Figure 2 are quite clear and relatively straightforward to delineate and map I would imagine. However, I would guess that some of the images you were using were less clear and the channels less well-defined. In such circumstances I suspect there's a degree of uncertainty and
error in your mapping. How do you quantify this, and can you put some numbers on this?
We have quantified the repeatability of our mapping by repeating the mapping (using the same mapper) for the Rhone Glacier (shown in Figure 2a) which contains >100 channels (>0.5 m), some of which are more complex to delineate. We find a 2.6% difference in the drainage density for the glacier and a 0.21% decrease in total channel length, when comparing our original to our repeat mapping, which is now detailed in section 3.1. We have added a figure in the appendix (Figure A3) that shows a comparison between the original mapping in this study, and the mapping completed as part of the accuracy assessment. This error acknowledges there is some subjectivity involved in mapping the channels, but it is low enough to ensure that the drainage density for each glacier is still a reliable reflection of the drainage network, and glaciers can clearly be grouped into low, medium and high drainage densities (for channels >0.5 m). |
| Figure 2 | Following on from the point above, and particularly relating Figure 2, I am curious about how you determine what to include in your delineation. I can definitely identify several channels in part (a) that you do not choose to delineate in part (b). This causes me concern, since it suggests an (inevitable) degree of subjectivity is integral in this study, and there are clear consequences of this when it comes to considering metrics such as drainage density. Some of the channels not mapped by the authors are only just visible as they are quite fine and thus presumably smaller (in parts (c) and (d) for example, there are numerous very small channels, which would be very hard to map, but are worth being aware of). As a result, there is, I guess, a size-threshold element as to which channels are included. However, to me, there are some channels that I would consider to be of similar size (and clarity) to those which have been delineated but which the authors have chosen not to map. I think this issue is of considerable concern, and needs consideration |

| | |
|---|---|
| | and arguably, more
channels need mapping.
We agree that there is a degree of subjectivity and human error in the mapping (as quantified above). However, as mentioned in the methods we only map channels that can be clearly delineated at a 1:1000 scale, and whilst some small channels can be seen at this resolution, they cannot be clearly mapped and we focus on streams >0.5 m to prioritize accuracy and because they carry the bulk of the meltwater. We have now modified Figure 2 so that all images are shown at a 1:1000 scale so it is easier to see what is and is not visible at this resolution. We have modified panels a & b to an example that shows more small channels so it is more apparent where small channels exist but haven't been mapped. |
| 218 | This concept of the mapping resolution and the fact that you are mapping channels above this threshold, is important. However, in the imagery, it looks to me that there are hazy, grey areas which most likely indicate a dense network of small channels. Whilst you can't map these individually, they are areas of channels and so I wonder if this needs to be considered (particularly in your drainage density calculations).
We agree with the reviewer that there are dense areas of very small channels that can be seen below the imagery on many glaciers but are too small to map clearly. Whilst we could identify the number of glaciers these occur on, this would not be a fair comparison between glaciers as it would be harder to define a cut-off resolution and as the reviewer notes, we are likely to only see these dense networks on larger glaciers. Thus, we take a conservative approach by only mapping channels we can clearly identify >0.5 m to avoid additional interpretation and to keep our focus on the larger channels that carry the bulk of the meltwater. |
| 219 | As a consequence of the threshold mapping resolution, I am uneasy about this differentiation between glaciers with and without channels. More accurately, those without are simply those without channels above the threshold mapping resolution. It seems likely to me that they do have supraglacial water flowing in channels, but
these are in smaller networks that are not easily identifiable as discrete channels.
We have added clearer language, i.e., 'glaciers without large channels (> 0.5 m)' when discussing glaciers without visible channels. |
| 221 | I am concerned about the observation that all glaciers above 5.6km$^2$ have channels, and the insights being drawn about bigger glaciers having channels. Could it not be simply that bigger glaciers have bigger channels while smaller glaciers have smaller channels. As a result, we don't see the smaller channels as easily (due to the image resolution), and so are swayed into seeing these as lacking channels?
We agree that glacier area likely controls the size of the channels, which may affect drainage density because we only map large channels above our threshold (0.5 m). However, our research reveals that glacier area is not the primary control on drainage density, which is evident by the fairly weak but still significant Spearman's rank correlation ($\rho$ = -0.10) between glacier drainage density and glacier area. This means |

| | |
|---|---|
| | that whilst the size of the glacier probably does affect the amount of channels we detect, there are other factors such as slope and glacier mean elevation that are more important for determining the density of channels. |
| 231 | I've said it previously, but I am very concerned about the distinction that is being made between glaciers where channels are/aren't present. It's about resolution and not presence to my mind. If glaciers experience melt, unless all surface water immediately enters the englacial region of a glacier, there must be some surface water and to my mind, this must (at least in part) be in the form of channels. I think the distinction is between the size of the channels and not their presence. I would also argue that given your imagery was gathered in mid-July, it is highly likely that the amount of generated melt is not at its maximum, particularly on glaciers at higher elevation. The impact of altitude could be fundamental, since at this point in the melt season, glaciers at lower elevation may well have much more melt occurring than those at higher elevations. This may thus be the driver of there being more, larger, and denser channels on these lower elevation glaciers. We have strengthened the language surrounding where glaciers do not have large (0.5 m) channels and mentioned why we use the resolution that we do in the methodology but acknowledge that the channels below our mapping resolution are present and may also be important. The imagery acquisition date will affect channel development, and we find that glacier mean elevation affects drainage density ($\rho$ = -0.66) (Figure 6). However, we also find that other factors, e.g., slope, clearly affect drainage density. Hence elevation is not the sole driver of drainage density. It would be expected that elevation is an important control on channel distribution regardless of the date of imagery acquisition, as less melt will occur at higher elevations, resulting in smaller and fewer channels. |
| 255 | It is a little odd that some channels disappear 'below the mapping resolution'. I am assuming you mean that they disappear as they head downglacier, since the phraseology of the various ways channels terminate in this passage implies this. However, I would have imagined that as channels flow downslope, they get bigger (as they carry more water) rather than disappearing under the threshold resolution for mapping. Amended - The reviewer is right in that this terminology refers to channels that disappear downglacier, which seems counterintuitive as channels widen downglacier. However, there are many instances where a channel may have terminated in a crevasse or moulin, but the channel terminus location is obscured by debris/snow, or the imagery resolution is not sufficient for identifying the terminus. There are also instances where the channel disappears below the resolution down-channel, which may be because a crevasse has captured some meltwater up-channel, which we cannot pinpoint. In response to this comment, we have added a new sentence in paragraph 2 of section 3.1 |

| | |
|---|---|
| | to expand upon our terminus categories. |
| 266 | I'm a little uneasy about the statement: 'large channels often occur at the interface between debris-covered and bare ice'. Firstly, what do you mean by 'large'? Do you mean in terms of diameter or length, and what criteria do you use to designate a channel as large? Furthermore, this seems to be quite a significant statement. Like your other metrics, you should quantify this – what proportion occur at such a boundary? Does the statement really hold true, when the channels in Figures 4a, c, d and e do not seem to be controlled by debris presence, and nor do those in Figure 2.

We have removed the size descriptive element from this sentence. The statement still holds true, however, as topographic depressions between medial and lateral moraines often capture this meltwater due to streams logically flowing into lower elevation areas. The reviewer notes that this isn't visible in much of Figure 4, but this is simply because not all of these show the described environment. Figure 4b does show a channel on the interface between clean ice and debris cover, and whilst panel 4e doesn't show a channel along this interface, it does show meltwater running off into the depressions between the lateral moraine and the glacier surface, which would likely be channelized if not for the thick debris and undulating topography. |
| 278 | In drawing conclusions about the relationship between, for example, sinuosity and slope, I think you need to express some measure of statistical significance so that we know whether these relationships are real. Spearman's rank values have been added to Figure 5, and the caption now specifies that each value displayed on the plots has a significance value of at least $p = <0.05$. |
| 300-306 | I am very uneasy about the statements made here. To simply state, for example, that 'a relationship between drainage density and glacier slope exists' needs statistical support. Looking at the graphs is Figure 5 (particularly e, f and g), I do not see a strong relationship, and so to back up your statements, you need to use some statistics to prove your point.

Following suggestions from Reviewer 1, we have added in the spearman's rank values to Figure 5. Hence, when citing relationships in Figure 5, there are now statistics to support these relationships which resolves this comment. |
| 308 | I am not a statistician at all, so can't offer a lot of insight here. However, you make several statements prior to this point regarding relationships in your data, yet it is only here, towards the end of your results, that you directly address statistical relationships. I wonder if things need reordering slightly.

We have now addressed this issue by adding Spearman's rank values (all statistically significant) onto Figure 5 so there is a quantitative measure of the relationships we refer to in section 4.3. |

| | |
|---|---|
| 313-314 | I come back to my concern raised above, that the relationship between drainage density and glacier altitude. I don't think this is surprising, particularly given the timing of the imagery used to explore drainage density. I feel that this is likely to be the most significant control.

There is a relationship between glacier altitude and drainage density as noted by the reviewer (Fig. 6), but this relationship would likely exist regardless of imagery acquisition date because channels would be expected to be more prevalent at elevations characterised by higher surface melt. We are less concerned by the potential over-exaggeration of the impact of glacier elevation because Figure 6 shows that many relationships exist between our metrics and drainage density, and this is not solely dominated by elevation metrics. Hence, despite the date of imagery acquisition it was still possible to establish a range of controls on drainage density. We now acknowledge in the methods that our date of imagery acquisition means that we are unlikely to capture the peak extent of channel distribution and provide further detail on the climatic conditions that precede imagery acquisition. We also acknowledge the impact that our imagery acquisition date has on how extensive channel distribution is in section 5.1. |
| 359-360 | Whilst I rather like the schematics you provide in Figure 7, there is a degree of conjecture here, particularly in relation to the proposed hydrographs. I'd like it to be made clearer that these are not measured or calculated at all, but rather assumed. Even then, I am wary of them, since in reality, these hydrographs and their shape are

strongly influenced by time of year, air temperatures, ease and speed with which channels are formed, diurnal temperature range etc. Amended – We have added a sentence to the figure caption to clarify that these hydrographs are conceptual and do not reflect measured proglacial stream discharge. In response to the concern about the time of year/air temperatures, we now note that these will change throughout the melt season due to increased subglacial drainage network efficiency. |
| 369-371 | Interesting point, but I'd also point out that higher temperatures will mean more melt is generated and thus there's more water available to incise deeply. We discuss this in lines 517 to 519 (in the pre-print version) and we deem this to be a valid point. However, we have now added an additional sentence that follows on from lines 369-371 (pre-print version) to ensure that this point is also included here. |
| 391-393 | Need a reference here to support the statement about lake drainage. Amended – an example from Gornergletscher has been included (Huss et al., 2007). |
| 399-400 | I am very wary of the statement: 'Our dataset provides new insight into meltwater transport across a large range of glaciers, allowing simple inferences to be made about connectivity and lag times'. Your data is all about channel mapping. Water transport and lag times are assumed based on this knowledge, so I would prefer that this is toned down a little. Amended – this has been toned down. |
| 411 | One thing we can't tell is whether the channels you are mapping are currently active. So, it is possible that the crevasse identified in Figure 8 has appeared relatively recently, thus intersecting the channel shown, and halting the flow of water in this |

| | |
|---|---|
| | channel further downglacier. I don't know how likely this is, but it should at least be considered, alongside the proposed idea that the crevasse is water-filled and thus flowing water overtops it. Without fieldwork observation, we are both speculating, so due caution in the interpretation needs to be exercised.

The role of crevasses in intercepting and transporting meltwater has been expanded upon in the last paragraph of page 18 in the revised version. |
| 416-417 | Similar to the previous comments, I am wary of the statement: 'the delay between surface melt and proglacial discharge will be larger due to longer pathways and the potential for supraglacial and subglacial storage'. This is true, IF storage takes place. However, you don't know if this is the case. There could be highly efficient englacial/subglacial channels. Again, I think that it needs to be made clearer that these are suggestions rather than based on direct observations.

We have modified the language in line 416 (pre-print version) to change 'will be' to 'may be', and we believe that the other text already reflects that we do not have in situ measurements, so these statements are speculative. |
| 448 | Throughout, you make reference to 'larger' channels, but this is never quantified. What do you mean by larger? Here, for the first time (I think) you indicate that by 'larger' you mean higher discharge. Is this what you have meant throughout? This needs making clear since to me, I had assumed you meant the physical dimensions of the channels. Regardless of what 'larger' means, I do think you need to quantify the criteria somewhat.

We mean physical dimensions and we have now stated this more clearly, i.e. >0.5 m wide. We had mentioned discharge to indicate that channels with larger physical dimensions are typically accompanied by higher discharge, but this is a qualitative assumption as we did not systematically measure width because our channels were manually mapped as a centerline, and this would be too data collection intensive. |
| 480-512 | I'm not convinced of the need for this section comparing supraglacial drainage on glaciers and ice-sheets. I feel that the paper is drifting away from its focus a little by including this passage. I'm not suggesting that this section MUST be removed, but rather simply suggesting that its value to the wider paper is not terribly clear, and thus this could be considered.

We think this section has value because most of the research on supraglacial channels has been conducted in an ice sheet setting. Hence it is useful to know how comparable channels on mountain glaciers and on ice sheets are, especially if there are some important differences that advance understanding beyond the body of knowledge gleaned only from ice sheet settings. We have added an extra sentence to state this at the beginning of section 5.3. |
| 518-519 | Presumably you mean an increase in the SIZE of the ablation area, but when referring to a 'reduction in area for smaller glaciers' you are referring to the entire glacier rather than the ablation area? A bit of clarity needed here.

Amended – we have reworded line 518 (pre-print version) to instead say 'rising equilibrium lines' to avoid confusion regarding the mention of both ablation area and overall glacier |

| | |
|---|---|
| | area. |
| 545 | 'off the glacier' rather than 'of the glacier'. Amended. |
| 557-560 | This last statement is the first such mention of the existence of other channels beneath the mapping resolution. I believe this is important (see my earlier comments) and so for it to only appear as the final sentence in the conclusions is rather late. I would prefer to see this discussed earlier, and the importance considered more. We agree that it is important to acknowledge the presence of channels below our mapping resolution and have now mentioned this in the methods section. Please see our response to the second general comment on this topic for a more detailed response. |

---

## Author Comment (AC3)

This is a novel paper involving the generation of a unique and valuable data set on surface stream characteristics for 85 glaciers in Valais, Switzerland from open access 0.15 m resolution SwissTopo orthophotos and DEM. The lead author is to be congratulated for manually digitising the 1890 streams, although it is a shame this couldn't have been automated. Automation was attempted, based on previous methods applied to the Greenland Ice Sheet, but these were found not to work here. Having created the data set, the paper investigates statistical relationships (correlation, PCA) between channel variables (segment length, channel slope, sinuosity, minimum elevation, maximum elevation and elevation range) and several glacier variables (drainage density, glacier area, mean slope of the snow-free area, aspect, glacier minimum elevation, glacier mean elevation and glacier maximum elevation). DiUerences in certain channel characteristics between clean and debris covered glaciers are investigated, as are diUerences between the way the channels terminate (moulins, crevasses, running oU the terminus or edge of the glacier, etc). The results are shown in a set of nicely produced figures and tables, and examples of diUerent types of streams are shown in Fig 4 (although I'd like to have seen an example of a stream which terminates in a lake, although this type of termination only makes up 1% of the total). We thank the reviewer for their positive assessment of our paper and respond to each comment in turn below in blue text. Firstly, in response to the final sentence above, we have expanded the spatial coverage of Fig. 4e so a small supraglacial lake is visible on Gornergletscher.

I think the justification for the work is adequately provided in the Intro, although I'd add the importance of the supraglacial stream network, and particularly for whether the water enters moulins or crevasses or not, for understanding subglacial drainage evolution, water pressures, and basal motion to this. Amended – we have now added additional information and references.

The data sets and methods are described well, although I'd like to hear more about precisely what automated methods were attempted and why they failed.

The first paragraph of the methods has been reworded because it suggested that we attempted more than one method of automation. We did not attempt other methods such as flow routing because the DEM was > 3 times coarser than the orthophoto and would reduce the number of channels we could detect. The methods text now clearly conveys that we tested the $NDWI_{ice}$ method (following Yang and Smith, 2013) and describes the imagery we tested it on, prior to opting to map channels manually. The $NDWI_{ice}$ method is likely best suited to coarser imagery where the output is not complicated by water filled crevasses, ponded surface melt and surface debris.

I also note that the orthophotos were collected in mid-July 2020. Can this melt season be put into some perspective? Was it a high accumulation winter previously? What was the weather doing in the spring and early summer? So what were the snow / ice conditions in mid-July this year compared to other years? What are the implications of using orthophotos from this time? I imagine results would be very diUerent if they'd been collected later in the summer during very high melt conditions? Does the timing of the orthophotos explain why you only found 85 (out of 285) glaciers with streams on them? This seems quite a low number to me.

We have now added into the methods that the conditions that preceded the acquisition of the orthophotos in mid-July 2020 were not highly abnormal. The previous winter (DJF) precipitation total was 570 mm which is lower than the 2009/10 to 2019/20 average of 704 mm, meaning there was less winter accumulation than normal. Temperatures began to rise above freezing in May time, and May in

2020 was warmer (3.4°C) than the 2010-2020 mean temperature of 1.7°C. However, July was colder than average (4.8°C) compared to the 2010-2020 mean of 6.3°C. Hence, it is possible that the snow-cover at the time of imagery acquisition was slightly below average (2010-20).

In response to our detection of channels >0.5 m on 85 out of 285 glaciers, this more than likely represents the size distribution of glaciers in the study area, as 200 out of our 285 glaciers are smaller than 1 km$^2$ meaning that channels (if present) are likely to be harder to detect from our imagery resolution (15 cm). It may seem logical to simply omit these smaller glaciers from our dataset, but we do still find clearly mappable channels on some (33) of the glaciers <1 km$^2$. We now acknowledge in the methods that these glaciers likely have some smaller channels that are not sufficiently clear enough to map, but likely still form a key hydrological component of these glaciers.

The results are clearly presented overall, and the statistical analysis seems generally robust, and the results sensibly interpreted. The PCA doesn't add very much, and I wonder whether the authors had considered collapsing potential independent glacier variables (glacier area, mean slope of the snow-free area, aspect, glacier minimum elevation, glacier mean elevation and glacier maximum elevation) using PCA and then investigating the correlations between the PCAs and potentially dependent channel variables (segment length, sinuosity, drainage density)? This would appear justified given the significant correlations between all the variables (Fig 6), and the desire to try to explain the variability in channel characteristics.

We have explored how to reduce the dimensionality of the independent variable space following the reviewer's suggestions. We collapsed the glacier independent variables into a new PCA and explored correlations between the PCAs. This reinforced the importance of glacier area and elevation as a control on where channels exist, particularly when looking at the difference in clustering between glaciers that do and do not have channels (clustering differences are only apparent for glacier area) (see below). We also grouped points by drainage density to see if there is any clustering with the PCA that would provide insight into where the highest drainage densities occur. However, this has not provided any new insight and the original PCA already highlights the importance of area and elevation, as their presence in PC1 and PC2 explains 50% of the total variability. Instead, we strengthen the language around our PCA to state that it is evident that glacier area and elevation are the primary controls on variability within the dataset.

[Figure]

Left plot: PC1 compared against PC2. Glaciers with channels (>0.5 m) are in blue and glaciers without large channels are in orange. PC1 is primarily controlled by minimum, and mean glacier elevation but closely followed by max elevation. PC2 is primarily controlled by glacier

area. Right plot: PC1 vs PC2 with the points colored based on their drainage density.

I think the weakest part of the paper is the Discussion, which is not suUiciently focussed on the results presented, occasionally confuses results vs previous work, and appears a little long-winded and speculative in places. I think there are some interesting points to come out of the results and some legitimate comparisons that could be made with previous similar work on the Greenland Ice Sheet and other glaciers in the Arctic or which are debris covered. It's just that these need to be more streamlined and succinctly articulated. The Conclusions would then be stronger, and the paper would have more impact.

Following the suggestions in the detailed comments, we have rewritten the paragraphs highlighted by the reviewer as being weak in structure and focused on outlining our results first, before bringing in previous research. During this process we have also condensed parts of the discussion, specifically the sections on comparing our study area to ice sheets and debris cover, which has helped to streamline the discussion. We have tried to reduce speculation in places, and we have done this by toning down sections on our hypothesized hydrographs and where we discuss predicting glacier meltwater pathways based on hypsometry.

I provide more comments on all these points in the details below, together with recommendations for where grammar could be improved, clarity could be enhanced, or where I have queries.

I have one more general query which is why did the authors decide to use the English names for all the glaciers? Would it not be more appropriate to use the local Swiss names (which are in German or French)? Amended - We have modified the manuscript to use the local language for each glacier.

I think the work should be published if the authors are able to address my comments and suggestions.

Detailed line by line comments

13. I'd say '~ 2000' [as <2000 could be anything between 0 and 1999]. Amended.

12-14. Seems odd as worded. Would this be better: "Here we use high-resolution (0.15 m) orthophotos across a sample of 285 glaciers in Valais Canton, Switzerland, to identify 85 that contain supraglacial channels. For these 85 glaciers, we delineate ~ 2000 supraglacial channels (> 0.5 m wide) and investigate their distribution and characteristics." Amended – We have reworded the sentence but have shortened the suggested version to keep within the 250-word limit.

15-16. What are 'lower relief slopes'? Should this be 'lower angle slopes'? This phrase is used several times throughout the paper. Amended – we have changed this to 'lower angle' throughout the manuscript.

21. Should be "…where all channels terminate englacially…". Amended.

23-24. You could delete "the majority of channels reach the terminus supraglacially and" as you already said that on lines 18-19. Amended – We have reworded this sentence to make it less repetitive but kept the mention of where channels terminate to save the reader having to revisit information earlier in the paragraph.

43. Suggest "channels has implications…" Amended.

44-45 Suggest '…through and under glaciers, with potential to impact suspended sediment…' Amended.

48-50. The distinction between channels vs channel absence is a bit artificial here as the former is typical on an ice surface in the ablation area whereas the latter will be confined to higher elevations. Can you distinguish between importance of channels vs absence in ablation areas as this controls delivery of water to bed which influences subglacial water pressures and basal motion (e.g., Banwell, A., Hewitt, I., Willis, I. and Arnold, N., 2016. Moulin density controls drainage development beneath the Greenland ice sheet. Journal of Geophysical Research: Earth Surface, 121(12), pp.2248-2269). Channels vs absence on firn at higher elevations reflects melt rates vs infiltration capacity, and I agree this has implications for portioning of refreezing vs runoU. Amended – We have replaced these sentences with the implications of surface-to-bed routing of meltwater.

52-53. '…glacier systems…' is vague. If you elucidate importance for proglacial water quality, glacier dynamics, and mass balance above, you could refer to these things again here. But are you going to address the impacts in this paper? If not, I suggest delete this last clause. Amended – we have removed the last clause.

64-65. Suggest "…larger ice sheets also apply…" Amended.

67-8. Suggest "…about supraglacial channel distribution in mountainous environments, but previous research has helped to establish some fundaments (e.g., Knighton…" Amended.

75 'present' => 'presence' Amended.

76 'influences' [i.e. singular as refers to 'presence' ] Amended.

76-80. These two sentences are repetitive. Suggest combine to make the point about the controls of discharge and slope just once. 'increased slope' should say 'high slope'. Amended.

80. 'discharge rates' is wrong as discharge is a rate [discharge is the volumetric flow rate of a stream]. Amended – this has been removed.

86. Suggest changing 'systems' => 'glaciers'. Amended.

132 'records an historical' Amended – this section has been removed.

122-137. To what extent is all this info on climate and glacier area change relevant? Can you delete it all (or much of it)? Amended – we have removed the sentences that follow on from "the mass balance of Swiss glaciers in recent decades (Fischer et al., 2015; Davaze et al., 2020)…"

146 Need a comma here "small amounts of water, or incised channels where…" Amended.

141-147. You say you applied automated methods [i.e. plural] and refer to Yang et al 2019. Does this reference detail all the diUerent automated methods you applied? But then you only mention the first method you used and refer to Yang and Smith. This sems odd. Why did you single out just this one method to talk about if you applied others? Clarify here how many methods you applied, what they were, and what the problems with all of them were. Amended

– We acknowledge that the wording may have been confusing and have reworded this information to clarify that we only tested the NDWI$_{ice}$ method of channel detection. Other methods such as flow routing would not have been viable with the resolution of the available DEM.

148. when you say 'multispectral methods' do you mean 'automated methods' [as referred to on line 141]? Amended – Yes, this is now clarified that in the text.

149. You mention your data set here "high resolution cloud-free orthophoto imagery" but was this the data you first attempted to detect channels on automatically as mentioned above? State above what data you were working with. We have now added information about the imagery used to test the NDWI approach. This was the same orthophoto imagery (0.15 m) referenced throughout.

150-52. Suggest "hence 6% of glaciers were still snow-covered down to their termini and were omitted from further analyses as the presence or absence of channels could not be detected." Amended.

I assume by this statement that you included glaciers that were still largely snow-covered, as long as they had some exposed ice on them? Yes, we did. However, the mean percentage of snow-free glacier area was 38.9% in mid-July when we acquired our imagery. A very small number of high-elevation glaciers only had a small area of ice exposed, with a minimum snow-free area of 5%. This snow-free glacier area increases to 45.0% when glaciers have a mean elevation between 2500 and 2800 m.a.s.l. The lowest mean for an elevation band is 36.6% between 3100 and 2400 m.a.s.l, demonstrating that there is not a large difference between the elevation bands. This information has now been added into the manuscript.

150-7. You need to reorder this material. First state how many glaciers you start with, then how many you remove because they are too small, then how many of those you remove because they are still completed snow covered. Amended – this paragraph has been restructured.

167-8. How easy was it to distinguish how the streams terminated? We find that it is typically easy to identify where the streams terminate due to the imagery resolution, whereas the largest source of error is when the mapper stops mapping up-channel to avoid over interpreting. We expand upon our mapping accuracy in new sentences that have been added to the first paragraph of section 3.1.

Fig 2. The stream in C looks like it used to terminate in the moulin identified but that a new moulin on a crevasse has opened up above it. Upon further inspection of the feature that is visible upstream, it is not a moulin and the channel runs directly through it, without appearing to capture any meltwater. There are many moulins in that area, but they appear much darker in the imagery, and this feature looks to be a darker depression that contains debris. Hence, the channel terminus has been correctly identified.

256-7. You say 'When only considering terminal segments…' but you used the term 'channels terminate in a range of settings' on line 254. So what do you mean by 'terminal segments'? You mean ignoring those that join another channel or disappear below map resolution? Amended – We have clarified that we do indeed mean channels that do not join another channel or disappear below the mapping resolution.

Are there segments that disappear below map resolution but then look as though they reappear again down glacier? How common is this? What are the implications for your results?

If there are channels that disappear and reappear, we cannot confidently assume that they are part of the same channel, hence they will have been mapped separately. As we have now more clearly described in the methods, we take a conservative approach to mapping to avoid over-interpreting channels. A gap in channel mapping should not significantly affect the drainage density of the glacier, but it will affect the assigned terminus of the channel (i.e., assigned "disappears below the mapping resolution"). We do not include channels that disappear below the mapping resolution in the calculations for meltwater routing. Hence, if a channel disappears below the mapping resolution, and subsequently reappears and terminates in a crevasse, we would still be capturing the final channel terminus location, without affecting our values for where meltwater is routed. However, it is likely that we underestimate the number of englacially terminating channels where the terminus cannot be clearly identified, as we don't detect the true location where it re-emerges, but this would not be possible to identify without further information (e.g. from dye tracing).

261 'singular' => 'single' Amended.

261-2. I don't understand what you mean by 'Thus, when the percentage of channels terminating in each position are extracted as an average value from each glacier…' 'are' should read 'is' as its singular as it refers to 'the percentage'. But even with this grammatical change I don't think the statement makes sense, does it? Amended – we have reworded this sentence to clarify the meaning of the text. We are trying to convey that we calculate channel termini locations for each glacier (e.g., 30% of channels terminate englacially and 70% terminate supraglacially) and when you average these values for all glaciers you get X% terminating at a certain location. We display this value because it is likely more useful than using the raw data for all channels because a few large glaciers contain a disproportionately large portion of the channels.

274-5. You could cut words (and improve style) here and just say "Here, we investigate links between diUerent supraglacial channel characteristics. Previous studies…" [You refer to Fig 5 at the end of the relevant sentence below so no need to refer to it up front]. Amended.

279-80. Is this accurate? I think I'd remove the word 'clear' [as the boundary looks a bit fuzzy to me] and I'd say a sinuosity of 1.3 [ eyeballing the figure suggests you have 8 values > 1.2 but only 4 > 1.3 for slopes > 20 degrees]. Amended.

280-281. Do you need the word 'segment' here? You just said channel length on line 277. Amended - For the purpose of clarity and consistency we have kept 'segment' here and added 'segment' into line 277 and into the figure captions that use channel segment length as a metric.

281. You say 'which often have a lower density of crevasses' but there's no evidence for this in Fig 5b that is referred to. Is this interpretation – in which case this could be moved to the Discussion section. Or do you have evidence for this, in which case refer to it here. Amended – we have removed this.

282-283. You don't need to refer to the slope-sinuosity relationship in Fig 5a again here as you've just dealt with that. Just stick to telling us what the upper boundary for the slope-length relationship is. For example, you could say that except for one outlier, channels >500 m long are confined to slope < 20 degrees. Amended.

290 Consider "…along with…' => "…as do…" Amended.

300. What do you mean by this sentence? '…less evident' than what? '…lower number of data points' than what? Haven't you just been referring to the influence of a glacier characteristic (slope) on channel characteristics (length & sinuosity)? Amended – we have rephased the sentence to make it clear that we are referring to the number of data points in Fig. 5a-b compared to Fig. 5e-h. The reference to slope in the paragraph above refers to slope of individual channels, hence it is grouped with the paragraph on the controls on channel characteristics. This differs to the mention of slope on line 300, which refers to the overall slope of the glacier, rather than at a specific channel. This was likely confused by the mention of 'glacier slopes' when Fig. 5a was mentioned, hence we have now edited this to improve the clarity.

302-305. I think these statements about drainage density vs minimum or maximum glacier elevation and ref to Fig 5f and g are spurious. What you say is based on just 4 data points, which is not an adequate sample. I'd just point out no obvious relationships between drainage densities and either min or max elevation. Amended – we now note that there is less evidence of a relationship in Fig. 5g (p = -0.39), and tone down discussion of Fig. 5f, but suggest that a larger dataset would be needed to validate whether a relationship might exist between glacier drainage density and minimum glacier elevation.

306. Do you mean 'interception' not 'inception' here? Amended.

316. Should say 'Fig 5e' here. Amended.

340. What do you mean by 'variables such as drainage density'? This is not clear. What is drainage density an example of? Should you not just list all the variables that are not closely related to just one other variable? Note you're saying that drainage density is not obviously related to a single causal variable, which was my interpretation of the bivariate plots 5f and g above. Amended.

341. The word 'singular' is used incorrectly here. You could say "… not closely related to just one other variable. Overall, our PCA analysis reveals no single main driver of variance.." Amended.

354. I think 'restricting' should be changed to 'determining' or 'aUecting' or 'controlling' as the first part of your sentence is just establishing a general relationship not the direction of that relationship. Amended.

354-6. This sentence is not quite grammatically correct as it should be '…can either intercept…or route…". I'm not quite sure what you're saying here. Are you saying that shallow crevasses may simply route the surface water along them, contributing to the supraglacial drainage system and maintaining channel length? Whereas deep crevasses may intercept surface water and deliver it to the englacial drainage system, thereby reducing surface channel length? Amended – this has been rewritten to more clearly convey what the reviewer has suggested.

358. I think 'surface' is redundant here. Amended – this has been removed.

358. I think you should introduce Fig 7 at the start of the next paragraph and introduce it fully. Is it depicting the influence of slope more than elevation? Or at least as well as elevation? Amended – we now introduce figure 7 in the second paragraph and that paragraph now discusses both slope and elevation.

361-371. You refer to 'valley glaciers' and 'Upper Theodul Glacier' in this paragraph but you don't systematically take us through the increasing drainage densities going from left to right in your Fig 7. I think it'd be helpful to describe the drainage density component to Fig 7 more systematically and thoroughly, perhaps with more ref to your case study e.gs shown in Fig 4. Amended – we have rewritten this paragraph to go through Figure 7 in an orderly manner with reference to case studies.

360. Can you say 'Alpine settings' rather than just 'the study area' to make your conceptual model more generic? Amended.

362 'higher' => high. [you say 'small' not 'smaller so should say 'high' not 'higher']. I'd not use comparative adjectives through this paragraph unless it's obvious what your comparison is with. Amended.

370. It will not just be glaciers extending to lower elevations that will require channel incision rates to increase. It'll be the likes of Upper Theodul Glacier too won't it? Amended – this is indeed true, but what we meant to convey was that whilst warming will be universal, glaciers with larger portions of their mass at lower elevations will likely be more vulnerable to temperature increases. We have modified the text to reflect this.

385 'or a slightly delayed peak. [peak;singular] Amended.

400-1 suggest 'main stem channel segments' Amended.

407 Delete 'compared to 0 % at the Aletsch glacier' as you've just said that. Amended.

409-10. "…but rather may act as part of the channel network and are mapped as individual segments as they may not be continuous." The second part seems to contradict the first part. Surely a channel network must be continuous, doesn't it? Whether you can see it all or not is another matter.  Can you clarify the  point you're making here? This section has been rewritten in response to another reviewer's comment and hopefully helps to improve the clarity. What we are trying to convey is that there is uncertainty over whether crevasses capture or route meltwater, so we don't map meltwater along crevasses / crevasse traces unless meltwater can be clearly seen to reemerge.

411-12. What do you mean "We attribute the diUerence in drainage pathways…"? Is this the diUerence in drainage density between Aletsch and Upper Theodul Glacier'? Things are getting a bit hard to follow here. Amended – this section has been separated into a new paragraph and we have reworded this sentence to provide added clarity.

414-428. This is all rather speculative and rather long-winded. Also, it seems odd to be talking about specific glaciers again having introduced your conceptual model (Fig 7). Can't you continue to talk in the generic way wrt Fig 7, having given examples of the 3 types of glaciers depicted in the previous paragraph? Can you summarise much more succinctly and based on evidence where possible, the impacts of the diUerent surface stream densities, on hydrographs, basal motion, and proglacial hydrochemistry? Amended – this section has been rewritten more concisely within a new, smaller paragraph. It now focuses on how the glaciers described above fit into the conceptual model.

Fig 8. I'm not convinced Fig 8 adds to the paper as it's just one example of a particular

phenomena that's not especially groundbreakingly novel. Figure 8 is used to illustrate one of the uncertainties we have with identifying where meltwater is going, rather than being novel, hence we believe that keeping it is helpful.

435-453. This whole paragraph is very speculative. It contains 4 instances of the word 'likely' and 2 of the word 'may'. I don't think this deserves 18 lines of prose. Amended – we have rewritten this paragraph to more clearly convey our evidence first, followed by a more logical wording of the evidence to support our conclusions.

441. I find the term 'higher relief slopes' odd. Is this used widely in the fluvial geomorphology literature that I don't know about? Why not just say 'steeper slopes'? If you agree, can you check all instances of this term in the paper and change accordingly? Amended – we have changed the language throughout.

442. '…this figure'. What figure are you referring to here? The 20 degrees? I think this sentence needs writing more precisely. Does your PCA allow you to add evidence to the statement I think you're making here? We have amended the text to clarify that we are referring to Fig. 5a. The point is that flatter areas are more likely to contain larger channels, and larger channels are more likely to be sinuous. Given the argument set out we would be looking for a relationship between sinuosity and glacier slope (mean) in the PCA, which isn't particularly apparent. This is likely because mean, min or max slope are unlikely to provide a good enough measure of whether a large enough low angle slope is present.

454 -464. It's diUicult to work out from this paragraph, which statements are based on the new evidence presented in this paper, and which are based on previous research. Ideally these discussion paragraphs should clearly state the former and then bring in the latter to show the extent to which previous work supports or contradicts the findings from the new work presented.

For example, where is the evidence for the first sentence? The only thing I recall wrt debris-covered vs clean glacier is Fig 5c which shows no diUerence in the sinuosity between debris-covered and clean glaciers in Valais.

Amended – we have merged the paragraphs from lines 454 to 464 and lines 465 to 478 to make the paragraph more concise. We now clearly state the evidence from our research at the beginning of the paragraph and contextualize it amongst previous research later in the paragraph. The axis of Fig. 5c has been amended and the differences (even if small) between the three classes are now more obvious in the figure. The data from this figure was found to be statistically different based on an ANOVA test, which provides support for there being some difference between the classes.

465-478. As above. Potentially some interesting points in here but the discussion must be related to the results you show. Amended – we have reduced this paragraph and merged it with the previous paragraph which now more clearly follows on from results we show.

481-499. Again, I do not think enough in this paragraph stems from the work presented in the results. It's too speculative. Can you just compare your results with the results of similar work (a lot by Lawrence Smith, Kang Yang and coworkers) on the GrIS? By all means oUer a sensible reason for any diUerences, but avoid all the lengthy speculation. Amended – in response to this comment and the one below, we have merged this paragraph with the one below to provide a more concise comparison between our work and research on ice sheets.

500-512. OK This para is better and does what I suggested above. I don't think the last sentence follows on from the rest of the paragraph. Instead, I'd weave that info and the refs into a point in your introduction, justifying that the study of surface streams and where they terminate is important for subglacial water pressures, subglacial drainage evolution, and basal motion. Amended – this sentence has been removed and further information on the implications of where channels terminate has been added to the introduction.

514-529. I think this para is basically fine – it's speculative but then it must be as it's about the future.

532. Should say "…dataset on…" Amended.

536. Delete 'existing'. I'd say ''low slopes'. Amended.

537 suggest change 'mass' => 'area' as that is what you measure. Could delete 'We find that'

Amended.

539 'low ice surface slopes' YES!

543-5. This sentence is not quite grammatically correct. "…the percentage of channels…revealing that…80% of channels" do one thing and 20% do something else doesn't make sense. It is not the percentage of channels that reveals it. I think this whole sentence needs rewriting (also of =>oU). As I mentioned earlier wrt lines 261-2, I don't follow what you mean by 'averaged by glacier'. Whether you simply calculate the % of diUerent types of channel irrespective of glacier and quote those, or whether you calculate the % of diUerent types of channel for each glacier, and then average all the percentages for the diUerent channel types, and quote those doesn't matter. You get the same result. Unless I've misunderstood something. Amended – we have rewritten this paragraph and split it in two. When we refer to average by glacier, we mean that we have calculated the % of channel terminus locations per glacier and then averaged this amongst the dataset. This is to make sure that each glacier is weighted the same, because if we didn't calculate the average per glacier, the data would be over-represented by the Aletsch glacier (582/1890 channels) which contains entirely englacially terminating channels, which is not the case for most glaciers in Valais.

546-7. What exactly do you mean by "The variation in where channels are located'? Are you talking about elevation on the glacier, whether they're on clean ice or debris, close to or far from medial moraines, steep slopes or shallow slopes, or what? Amended – this was redundant, and we now just refer to where the channel terminates.

548-9. The phrase 'with diUerent glacier geometries likely predictive of glacier drainage density and channel pathways' is vague. You've not mentioned glacier geometry so far in the paper so what are you referring to? And 'pathways' has been mentioned 4 times but again not very clearly defined. You say glacier geometry is likely predictive of glacier drainage density and channel pathway and refer to Fig 7. But if I look at your evidence in Fig 5e-h it's not clear that anything shown really controls drainage density, is it? Your corelation matrix in Fig 6 shows highest correlations between glacier slope, glacier mean elevation and drainage density so glaciers with low slopes and low mean elevation have the biggest drainage densities. Anyway, I think the paper would be strengthened if you based your conclusions on the evidence that you present in the results section and avoid weak speculative statements. Amended – we have rewritten this

section to clarify what we mean by channel pathways. We have also provided an example of how drainage density and proglacial stream hydrographs can be predicted based on glacier characteristics.

---

## Referee Report (RR1)

**Distribution and characteristics of supraglacial channels on mountain glaciers in Valais, Switzerland. Holly Wytiahlowsky et al. 2nd review for TC by Ian Willis**

Thank you for the vast number of improvements to the paper following the suggestions by myself but also the other two reviewers. I noticed that many of our comments were similar.

The paper is greatly improved and is closer to being ready to publish in my view. I've been through the paper carefully and have a few remaining general comments and a few more specific detailed comments and questions.

The science is good, and the results are interesting, but my comments and suggested edits are to encourage you to work on articulating the results and implications of your work more carefully, precisely and logically than you do, with the ultimate goal that more people will read it and it will have a better impact. The Discussion was the main part of the original paper that needed improving and you've done a good job in doing so. But it remains the section that'll most benefit from more work.

General points

1. Throughout your paper, it would be clearer (and you could lose a lot of repetition), if you did not continuously tell us that your channels are 'large' or '> 0.5 m wide'. Tell us in your methods that you can only resolve channels above this size and that hereafter you refer to these simply as 'channels'. Then you can delete all subsequent references to '> 0.5 m' or 'large', as we know that when you use the term 'channels' it is an abbreviation for 'large channels > 0.5 m'.

2. You use the word 'morphology' and 'morphometry' with respect to channels throughout the paper and it's not clear what you think the difference between these two is. Can you better define these two terms when you first introduce them and then use the correct word thereafter? Or do you see them as the same thing, in which case just use one term consistently.

3. There are places in the paper where you're comparing your work to 'ice sheets' but really you're just comparing with the Greenland Ice Sheet. I think you should alter all such sections to refer to the GrIS only and not give the impression that there are vast surface drainage networks on the Antarctic Ice Sheet!

4. Similar to above, there are places in the discussion and conclusions where you refer generically to 'mountain glaciers' or 'valley glaciers'. You need to decide whether you think your results for the 85 Valais glaciers are universally applicable and can be generalised. Parts of your discussion suggest you do not think this and work is needed to apply your methods to other areas of the world. I would agree with this. In which case you should always refer to 'Valais glaciers' and not imply that your findings are necessarily applicable everywhere.

5. I would encourage you to proofread your work more than you do and have your other authors look over it, to spot and correct grammatical errors. Unfortunately, there were a lot of these, as some of the detailed line by line comments below show.

Detailed line by line comments

14. we map 1890 channels (to keep tenses consistent).

35. repetition of word 'rapidly'

48 could say ' with potential to impact the suspended…"

54-5. Better English would be "…which has implications for subglacial water pressure, the onset of subglacial channelisation and, in turn, ice motion."

72. Better to say "…to have a larger debris cover…" or "…to have a larger coverage of debris".

81/82. "…high concentrations of channels correlate with …" or "high concentrations of channels appear to be correlated with…"

91. Discharge is a rate so should say "discharge is a strong control…" [or "volume flow rates are…"]

104. '…from glaciers…" plural!

105-6. "… determining the extent to which surface hydrological characteristics (e.g., channel, channel transport pathways) are uniform between" would be less ambiguous.

120. "…comparable with Switzerland's as a whole" [Add the apostrophe - it is not comparable with the size of Switzerland!]

121. "…and varying crevasse densities…"

130. "found to contain" => 'containing'.

175. "has been found to be" => "is".

There are loads of examples of the above throughout your paper. Do a search for all the words 'found' and see if you can get rid of all the extra words.

191 remove ref to 'Canton' as you've already established that Valais is a canton.

205. delete 'width' or 'wide'.

217 'was assigned a code based…"

218 'The type of channel terminus..'

232 "…mapping had been repeated…"

233 "small enough for us to conclude that the original mapping provided a…"

234 – add "…and total channel length" to the end of the sentence.

234-5 "Both sets of mapping also clearly identified how the channels terminate. The primary source of uncertainty here, stems from knowing when to stop mapping up-channel."

239. "found to contain" => "that contained". Also, shouldn't you replace "85" with "the" as the number of glaciers with channels is reported as a result on line 280?

248 "had been used"

260. Need a colon after 'three classes'.

263-275. There is a mix of present and past tense throughout this paragraph.

279 '…glaciers that had an area…" would be less ambiguous.

280 delete 'were found to'

283-4. delete 'were found to'

302-3. "and inclusive of the higher one"

306-314. "Glaciers without large channels (>0.5 m) are more likely to terminate at higher elevations (mean minimum elevation: 2936 m) compared to glaciers with channels (mean minimum elevation: 2797 m), which are often characterised by longer valley glacier tongues. Where glaciers support channels, they are more likely to have a higher maximum elevation (mean max elevation: 3637 m) than glaciers without large channels (mean max elevation: 3555 m). Where channels >0.5 m are present, there is a mean drainage density of 2.4 km/km2 and a maximum of 15.2 km/km2. The latter was found on the Oberer Theodulgletscher, which is situated on a low slope plateau and has the lowest glacier slope angle in the dataset (13°) (Fig. 3c, Fig. 4a). To summarise, glaciers containing channels are larger, have lower mean slopes, and have a larger portion of their area at lower elevations compared to glaciers without large channels (>0.5 m)." [149 words]

could be improved and shortened to:

"Compared to glaciers without channels, glaciers with channels are more likely to have longer tongues that terminate at lower elevations (mean minimum elevation = 2797 m vs. 2936 m) and have higher maximum elevations (mean max elevation = 3637 m vs. 3555 m). Glaciers with channels have drainage densities ranging between 2.4 km/km$^2$ and 15.2 km/km$^2$. The latter was found on Oberer Theodulgletscher, which has the lowest glacier slope angle in the dataset (13°) (Fig. 3c, Fig. 4a)." [78 words].

This is just one example of where a verbose writing style could be made more succinct.

317. should say 'top right'. In the Figure, is it possible to remove the white 'gap' in the top right part of (a)?

326. Grammatical problems. This would be better:  "Few segments exceed 1,600 m, as the ablation areas of most glaciers are smaller than this."

328 ""

340-3. This sentence still needs work. It's not quite right.

345-6. Wouldn't this be better for this entire sentence: "Qualitative observations suggest that channel distribution and morphology are controlled by glacier structure and topography."

350 'low' not 'lower'

353. This title should come above the previous paragraph as that previous paragraph is exactly on this topic!

354. This sentence does not capture what this section is about. You're not only investigating links between channel characteristics but also the glacier controls on the channel

characteristics. Suggest you adjust the section heading and this sentence so that they match what is being done here.

359. I'd say 'shallow slopes' as 'lower slopes' could imply slopes at a lower elevation (i.e. opposite of upper slopes).

362. "lowest slope angles" => 'shallowest slopes"

366 I assume this should say 'ice' not 'glaciers'

372. 'higher-order channel' not '…channels'

377 'each of which is" singular!

394-5. Correct the grammar here.

399. I think Principal Component Analysis should strictly be capitalised (as you've done on lines 424-5).

404 delete "identified to be"

435. You need at least a semicolon between "variance" and "instead"

446-7. What do you mean "are likely to be" You know whether they ae or not. Do you mean "tend to be" here?

451 Comma before 'which'

454. I'd put a comma after 'meltwater and before 'which'. End the sentence after 'channels' start a new sentence "Conversely, crevasses that have closed…"

478 and Figure 7. Do you mean 'inception' or do you mean 'interception'? The latter would seem to make a lot more sense to me. "Inception" refers to the beginning or starting point of something, while "interception" means to stop or capture something that is moving or being passed. You say 'intercepted' on line 481 which seems correct.

486-7 should say "location on a high elevation plateau". You can't say 'higher' as you don't explicitly state what you're comparing it with. Be careful of your use of comparative adjectives.

493-4. I'd delete this last sentence as it confuses things. You've already described the balance of glacier surface melt vs channel melt.

574 "compared to' => "and".

578 "seasonal variation in ice velocity" [or "summer speed up in ice velocity"].

614 "observed to occur" => 'occurring'.

614 delete 'We find that'

616 Delete 'Our observations find that" [it should say "We find that" or "Our observations show that", but these and all similar phrases used throughout your paper are redundant].

613-630. This discussion on the relationships between certain channel variables on the Valais glaciers (this study), vs. those on high Arctic glaciers reported by St Germain and Moorman and those on glaciers reported by Gulley et al (2009) [location not stated] is impossible for me to follow. I've read it through several times and cannot follow the logic. I'm not convinced the statements are logically connected. I'd encourage a rewrite of this whole paragraph. There is a confusion about what the evidence is supporting the statements. Is everything just based on correlation? This is not the same as causation of course. We need better distinction between correlation and causation and the latter should be linked to physical processes. Line 614 states that slope affects sinuosity which implies causation. You say that wider and more incised channels are more sinuous but where is the evidence for this? You don't measure width or incision. You say that such channels are likely to carry a higher discharge, but what is the evidence / logic / process base of this statement? You suggest it's because St G & M "attribute" higher sinuosity to higher discharge. But on what basis did they do that? All this from lines 614-619 seems poorly reasoned.

Then you state that St G & M find a positive association between channel slope and sinuosity for glaciers in the Arctic. But this is not observed for the temperate glaciers in your study. So, state what is observed. From Fig 5 it's a moderate -ve correlation, right? OK, so we have a discrepancy which needs explaining. Then you state that steeper channel slopes increase channel incision and ref Gulley et al. You should tell us that that study was also for glaciers in the Arctic. On what basis did Gulley et al reach that conclusion? Was it also correlation analysis like you've done? Then you refer to a positive relationship in your study between incision and sinuosity. But where is the evidence for this? Incision is not one of the variables you define in your correlation matrix. So this is just anecdotal evidence, correct? I don't think you can make any meaningful statements about relationships involving incision when you don't show any evidence for incision. On line 624 you state "These channels are likely to continue to evolve inter-annually due to their incised depth." But where is the evidence or what is the process base for this? And how is a statement about what may or may not happen in the future relevant to the discussion here, which is about explaining the present day relationships between channel variables? Then you state "Given this…the relationship between slope and sinuosity represents the conditions under which large channels can form (i.e., flatter, less crevassed regions) rather than the direct impact that slope has on sinuosity." But what do you mean by this? I can't work out how you would logically deduce that from the previous statements.

651. What do you mean by 'appearance' here? Can you just loose that word as the subsequent discussion seems to be about sinuosity only.

652-654. The statement here "Channels that have a proximal debris source and those that run directly through debris cover tend to be more sinuous than channels on clean ice (Fig. 5c)" contradicts the statement in the results on lines 365-8 where you say "We find no noticeable difference in sinuosity between channels on bare ice and those on debris-covered glaciers (Fig. 5c). However, channels proximal to debris …are more likely to be highly sinuous than channels on bare ice or continuous debris cover". Which statement is correct?

654-6. I cannot reconcile this sentence with what is shown in Fig 4d. The sentence states that Fig 4d shows channels on clean glaciers but the non-sinuous channel in Fig 4d is on the same glacier as the sinuous channel. Are you talking about the two sides of the glacier tongue here?

658-9. This sentence isn't quite right. It is not the less sinuous channels that are funnelled through areas of bare ice. It is the water that would be funnelled. Or you could say the channels become straighter as they pass through the clean ice area. I don't see what your explanation for this finding is. Is it to do with the debris cover or is it just to do with the slope? If the clean ice zones of debris-covered glaciers have steeper slopes, then would this create

the straighter channels? Or is it from the paragraph 613-30 that you're ruling out a slope control on sinuosity?

659-666. This reads like a literature review of previous work. Can you better explain how this work is relevant to your results? Can the processes that are implied with reference to the previous work explain any of your findings relating to the slopes and sinuosity relationships on glaciers or parts of glaciers with different amounts of debris?

666-668. This sentence doesn't seem relevant here. It reads as though it should go in the introduction of a different paper justifying why it'd be useful to assess the effect of debris within supraglacial channels on channel morphometry (which you don't do in this paper). Or it could go in a future work section of this paper.

668-671 doesn't seem relevant at all – I'd personally delete this.

672-5. You say 'further insight' but further insight into what exactly? How is this sentence relevant to your work?

675-677. Could the sentence on lines 666-668 be merged with this and better articulated if you want to end this section with recommendations for future work that stem from the discussion about controls on channel morphometry?

714 You need to say "Previous research on surface channel morphology has…" [or some such]

714. Just say 'ice sheets' not 'ice sheet settings'.

717. Need a full stop or semi-colon after 'environments'.

718. Grammar! Should say 'which are commonly observed". Also, why not just say "observed on ice sheets"?

718-720. You say 'some glaciers' but is this all but the large glaciers? This would seem logical given what you then say. By 'less interconnected' do you mean 'less dendritic'? You don't been less anastomosing I assume, which is what 'parallel networks' implies. It's just a question of scale as to whether drainage systems can be described as dendritic?

723. What are 'flow-stripes'? These have not been mentioned previously so you can't introduce them here in the discussion.

724 I'd say "both for Valais glaciers and for ice sheets" [note you could say 'Valais glaciers" throughout most parts of your paper to remove a few words and make the sentences flow faster].

726-7 I'd say "between ice sheets and Valais glaciers, for example debris may…formation on the latter". Then delete the 'For example' on line 727. You say 'channel formation' here but do you mean that? Or do you mean channel morphology [or morphometry]?

729. You refer to Rippin et al here but surely you should refer to your work first (and then bring in others' work if it supports your argument)? Otherwise, this is just a literature review.

730 You mean 'surface channels' not 'surface melt' here. Your paper is not about melt.

731 delete 'are observed to'

735. If you want a reference for the notion that lakes temporarily store water you could refer to this: https://tc.copernicus.org/articles/8/1149/2014/tc-8-1149-2014.html

736. And if you want one for the notion that lakes may refreeze you could use this: https://www.frontiersin.org/journals/earth-science/articles/10.3389/feart.2017.00058/full

But actually, if you were to delete lines 734-740 I think your paper would be improved as it remains more focussed on your Valais glaciers.

744. What do you mean by 'will develop'? Delete?

746 'will depend on the rates of'

747 I'd say 'associated' not 'subsequent' as higher melt will cause ELA to increase but also through the albedo feedback ELA increase will cause higher melt rates.

751 should you say 'and a reduction in drainage density'? Not 'or'. Because the example you give of drainage reconfiguration would reduce drainage density.

756 need a comma after '…evolution'

757-8. Poor phrasing – you could end the sentence "…future studies should repeat our work in such regions'' [or something along those lines]. Actually, your statement isn't quite correct as you do compare your findings to those of others in Arctic settings (St G & M; Gulley et al) although that paragraph did need improving.

760. Say 'on Valais glaciers'. Your previous sentence correctly noted that you can't necessarily extrapolate from the channels you delineate in your study to all mountain glaciers.

796 'found to contain' => containing'

812. You could clarify that by terminating englacially you mean ending in a crevasse or moulin.

814 you refer to channels terminating supraglacially or englacially here. By supraglacially I assume you mean those that run off the glacier? Why not make that clear in the sentence above too? So, the sentence above would read:

"…have 80% of its channels run directly off the glacier (terminate supraglacially) while 20% would end in a crevasse or moulin (terminate englacially)." Would that make sense? It seems a little odd to refer to a channel that flows off the glacier as terminating supraglacially. Would 'proglacially' be better?

816. Refer to your Figure 7 at the end of this sentence. Delete the word 'observed' in the next.

816. Correct the English. Should say "differ from those in typical ice sheet…" [note plural therefore 'differs' not 'differ'].

818-20. Your final sentence is poorly articulated and could be stronger. This would be better, but you should decide.

"Compared to the GrIS, drainage networks on Valais glaciers are less developed due to smaller drainage areas and limited distance for channels to merge. Additionally, the downglacier narrowing of the Valais glaciers further restricts the potential size of their drainage networks."

---

## Referee Report (RR2)

**Distribution and characteristics of supraglacial channels on mountain glaciers in Valais, Switzerland**

Holly Wytiahlowsky et al.

3rd review by Ian Willis

**General Comments**

I have commented positively on this work in my previous reviews and those points still stand. The paper presents a highly original data set, much of it obtained from meticulous manual digitising of tens of orthophotos. The descriptive statistics of the channel and glacier characteristics and their relationships (analysed using correlation and PCA) are all well done. It's good to see the improvements that have been made to the manuscript since my last review. For all these reasons I do think this work deserves to be published.

In my last review, I showed how the Discussion section needed the most work and it's good to see that in the latest version of the paper many changes have been made to the Discussion. I'm sorry to say, though, that I still don't think the paper is quite ready to publish because the Discussion and Conclusions still need some work. A fundamental problem is that many of the statements made in the Discussion and Conclusion do not clearly follow on from the data presented.

I think a major problem I'm having is that the Results focus on good robust quantitative analysis of glacier and channel characteristics and their relationships (4.1, most of 4.2, 4.3 and 4.4) with a short weaker section on 'qualitative observations' towards the end of section 4.2 (6 lines of text and Fig 4) but the Discussion and Conclusions focus a lot on those 'qualitative observations' and don't draw out some of the interesting things from the quantitative work.

Given the importance of these 'qualitative observations' I think they should be defined and described better and more thoroughly. Looking at lines 295-301, the 'qualitative observations' ostensibly show examples of how 'channel distribution and morphology' are controlled by 'glacier structure and topography'. These terms are poorly defined. Channel distribution is rather a vague term and not precisely defined but I take it to mean where they are located on a glacier vs where they are not]. Channel morphology is defined on line 49 as "channel shape and structure" which is slightly confusing as when I think of channel shape I think of the shape of a channel cross section. 'Planform shape' would be a better description as this is what is meant (i.e. whether channels are straight or meandering (sinuous). Channel structure is never defined. What is this? 'Glacier structure' features twice in this paragraph and nowhere else in the paper and is never defined. It seems to refer to patterns of crevasses, which is not what I'd call 'glacier structure'. 'Glacier topography' is not defined but I think this is probably clear to most readers, i.e. the pattern of surface elevation and its derivatives (e.g. slope).

Despite it being suggested that both 'channel distribution and morphology' are controlled by both 'glacier structure and topography', we're just given one example of how 'channel distribution' is controlled by 'channel topography' (Fig 4b), which is a channel occurring 'along the interface between debris-covered and bare ice'. We're also given an example (Fig 4c) of how 'glacier structure' influences 'channel morphology' which is a straight channel following a 'trace or shallow' crevasse. Finally, we're given another example of how 'glacier topography' influences 'channel morphology' (Fig 4e), which is a sinuous channel at low elevations on a flat part of the

glacier towards it's terminus. [We're also told that such channels tend to occur on 'large glaciers', which is not defined by 'glacier structure and topography' but is anecdotally thrown in here].

I'm sorry to labour the point here but this paragraph is weak and yet a lot seems to hang on it in terms of your Discussion and Conclusions. How to improve it? First, you'd need to very precisely define the terms. Second, the reader would need to have an idea of how representative these 3 examples are of all the glaciers that you've looked at. Otherwise, how do we know that you're not just 'cherry picking'? Personally, I'd recommend removing this section as it is. Given that you have done all the robust quantitative work, I'd strongly recommend you base your Discussion and Conclusions on that evidence and perhaps show these examples in Fig 4 (and more if you like) of glaciers that show the typical behaviour that you see from the analysis of your large sample. Your quantitative work cannot tell us anything about how 'channel distribution' relates to 'glacier structure or topography' but it can tell us how 'channel length', 'drainage density', 'sinuosity' are related (or not) to the glacier variables of 'area', 'slope' and 'elevation'. Note you cannot tell us anything about how 'glacier structure' [by which I assume you mean presence and orientation of shallow crevasses] influences channel characteristics as you do not have this information about crevasses in your large data set. What you can tell us about is that 'crevasse extent' influences 'drainage density' as that's shown in Fig 5h.

You'll see in the line by line comments, when you get to the Discussion and Conclusions that I'm pointing out all the instances where I do not see evidence to support the statements you make. I would recommend a fairly root and branch edit of the Discussion and Conclusions so that your statements more firmly follow on from the robust evidence you have.

I think your correlation matrix and your PCA show some very interesting findings, only some of which you draw out. For example, your first PC shows that the greatest variability in your data set concerns glacier area, glacier elevation and channel maximum elevation. The way these variables are related (+ve / -ve), which is also shown in the correlation matrix, is interesting and could be interpreted. I think I give a possible interpretation in the line-by-line comments below. Similarly, your second PCA shows the 2$^{nd}$ greatest variability in your data set is to do with drainage density, mean glacier slope and mean glacier elevation. Again, the way the variables are related in their contribution to PC2 (and in the correlation matrix) is really interesting and supports the main conclusion that you wish to make (and which you've depicted in your conceptual model). Fig 5h also supports this. As far as I can tell you have no evidence that 'glacier area' may affect drainage density (r = -0.1) although you do have evidence that glacier area may affect sinuosity (r = +0.29). These points based on your evidence need articulating clearly.

Another thing you'll see form the line by line comments is that you need to be careful to separate out correlation (which you have evidence for) from causation (which is your interpretation). Be careful and consistent throughout your paper on this point. I think I've given examples below on how you can change things or where you should.

I hope my comments are helpful. As I've said before, they are designed to improve the paper so that it's intelligible and people will want to read it, understand it, and hopefully reference it.

**Line by line comments**

**Introduction**

45. suggest "...why surface meltwater becomes channelised on some glaciers but not others..."

55-7. correct to "...as most channels on mountain glaciers are likely much smaller than those on the GrIS and therefore fall below the resolution of even the highest-resolution freely available satellite platforms..."

58-59. Better to say: "...comparable to those on mountain glaciers. Mountain glaciers are characterised..." [because 'the latter' strictly refers to "the channels on mountain glaciers"]

70-72. "Where channels occur, they are often reactivated annually" repeats lines 67-8. And "...deeply incised channels suggested to be a product of high discharge.." is similar to lines 64-5 as high meltwater production => high discharge.

79-81. Better to say: "However, much of what we know about supraglacial channels was established from observations of a small number of individual glaciers, especially those that are cold or polythermal (e.g., Knighton, 80 1972, 1981, 1985; Gleason et al., 2016; St Germain and Moorman, 2019).

91. I'd say "glacier surface characteristics"

92. Can you briefly summarise what the 'qualitative observations" are? Although if you follow my advice in my general comments this may be removed.

**Study location**

100 Swiss canton [not capitalised here as not a specific canton, e.g. Valais Canton]

101 I'd delete the 2$^{nd}$ 'area' so "... a maximum of 77.3 km2..."

118. delete 'right'? Isn't G Aletsch just in the centre?

**Methods**

154-5. I'd say "This is because they are likely too small to form channels large enough to be detected in our imagery, and because many of the small glaciers listed in the Swiss Glacier Inventory (SGI2016) do not meet the criteria for classification as glaciers (Leigh et al., 2019).

174 "...main channels were mapped..."

178-179. confusing to have 'terminus' for channels and for glacier. Suggest use 'snout' for glacier terminus throughout paper. [Or you could use terminus for glacier and terminal for channel]. Also, later you talk about a channel running off (terminus or periphery). Would it better to refer to that here. So you could alter your list here to say: The type of terminus was assigned to each channel, which was one of: running off the glacier snout, or off the glacier side, or terminating in a moulin, crevasse, or lake, or adjoining another channel, or disappearing beyond the image resolution (i.e.., the terminus was not visible and could not be inferred confidently).

Note I've changed 'periphery' to 'side' here as 'periphery' to me would include the front (i.e. snout). If you like these suggested changes, check your entire paper and make the relevant changes.

184. "Supraglacial channels on Glacier de Moiry and (c-d) on Allalingletscher…"

186. Should assessment be plural? Also, put in brackets the initials of the 'individual'. I assume (HW).

191. Tell us approx. what the time period was. E.g. "…was conducted over an approximately 6 month period."

192. "…of each channel length and glacier's drainage density…"

193. Could delete 'here'

195-6. This is unclear to me. Do you mean you mapped the up-glacier channel limit to where you were confident the channel existed? Note past tense 'were'.

199. Helpful to reiterate "..the 85 glaciers…"

204. Regarding segment length, wasn't this derived from the orthophotos not the DEM as implied here? Regarding straight line distance, is this in 2D plan (so from the orthophotos) or in 3D (so from the orthophoto and DEM)?

209 change 'which' to 'and'

201 could delete 'record'

199-219. There's confusion I think between these paragraphs about how you calculate elevation and slope. In the first para you say the DEM was used to calculate both. In the 2$^{nd}$ you say that slope was calculated for the snow free area so calculated from the DEM. But what about elevation variables? Did you also calculate that for the snow free part?

226-27. given what you said earlier you could reorder these and say "drainage density, glacier area, aspect, minimum elevation, mean elevation, maximum elevation, and mean slope of the snow-free area."

228-233. Could be abbreviated and improved to say: "A one-way ANOVA was performed to test the significance of the relationship between the three debris-cover classes and sinuosity. In addition, a Principal Component Analysis (PCA) was conducted to examine relationships among variables and identify the main drivers of variance in the dataset, with the data normalised to enhance pattern detection."

**Results**

243 I'd change 'on' to 'of'                and say 'km2, with a maximum…"

264. It would be helpful to indicate where these glaciers in Fig 4 are on Fig 1.

286. change 'terminus' to 'snout'? See earlier comment. Change throughout if you agree.

289. change 'periphery' to 'side'? See earlier comment. Change throughout if you agree.

292. What do you mean here? what is 'the average glacier'? Also, have you defined 'terminating proglacially already? Is this the same as having water running off the glacier snout or off the glacier side?

305. 'variables to test for…"

306. 'affect' implies definite causation, but of course you're doing correlation (not causation). You could say " 'may affect' or 'are related to' here.

312-13. "The 'debris' class generally contains more sinuous channels than…" is more correct

341. You can't say 'controls' here as this implies causation which you don't know from correlation analysis. From here and for the rest of our paper you'll need to be careful with this. You'll need to be clear when you're sticking to the facts (correlation) and when you're making inferences from them about cause-effect or processes. You could say "'associations between' or 'possible controls' here.

345. 'control' No

355. What does " high channel sinuosity can in part be explained by multiple weak correlations" mean? I'd just delete this.

358. 'controlled' No

361. 'relationships' [plural]

361-2. did you use all the channel and glacier variables in your PCA? If so state that here.

362-367. Capitalize 'Principal Component' when referring to a specific PC (1, 2 , 3 etc). Or just abbreviate to PC

366. Tell us what % PC1 and PC2 explained individually (not just together).

**Discussion**

377. I assume you want to say "Previous work has suggested that the presence of visible channels is primarily controlled by…" ? Because you do not measure 'meltwater supply' in your paper. Or is this interpretation based on your results? It'd be best to start each discussion point (so 5.1, 5.2, etc) with explicit reference to your findings, then your interpretation of those findings (make it clear where results end and interpretation begins) and you could also discuss your findings / interpretation in the context of previous work.

Section 5.1

This subsection is  headed "Controls on the spatial distribution of channels" . This is a little unclear as a title, especially as you haven't really shown results on this topic. Your paper is not about why channels form in some places and not others on a glacier.

382. Your statement "glacier area controls much of the variability within the dataset (Table A1)" is correct as glacier area dominates PC1. You could refer to PC1 after Table A1 to clarify this.

383-4. Your statement "… albeit with large variation in drainage density." Is rather thrown away here. Drainage density dominates PC2. So why not explicitly say that.

384. When you say "This variation is in part attributed to glacier slope…" what exactly do you mean? I can see that PC2 is dominated by drainage density (+ve) and glacier slope (-ve). Is this what you're talking about? Refer to the evidence for your statements. Note glacier mean elevation (-ve) also contributes to PC2. Why not mention this? What this means is that a high source of variability in your data set comes from these three variables contributing to PC2. Glaciers with high drainage density tend to have low slopes and are situated at low elevations. Use your results to discuss them.

384-5. Your statement "…together with ice flow velocity, governs the crevassed area of a glacier" is either from previous work or it's your interpretation of your results. You do not analyse ice flow velocity. Nor do you show the relationship between glacier slope and crevassed area. You need to more clearly discuss your results and explain what your evidence shows and how you interpret it.

388-92 'Channel formation is also governed by glacier hypsometry…' is a little confusing as 'hypsometry' is not one of the variables you quantified and investigated. Similarly, you talk about "glaciers containing a larger proportion of their area at lower elevations', which I agree is to do with hypsometry, but again, you don't measure or report this. What I can see is that the $3^{rd}$ most important variable contributing to PC2 is glacier mean elevation, and I can see from the Table A1 and the Fig 6 correlation matrix that drainage density is inversely correlated with glacier mean elevation, so lower elevation glaciers have higher drainage densities. Refer carefully to your data and evidence and make statements that you can support. Then interpret. The last two sentences of this para don't really contribute meaningfully to the discussion.

393. The word 'hypsometry'. Again, this is not something you explicitly focussed on in our paper so I'd remove reference to it. However, here you're focussing on explaining variation in drainage density (so we're with PC2 still) and you could also draw on evidence from the correlation matrix. It is glacier slope and glacier mean elevation that correlate with drainage density and contribute to PC2.

394. "The lowest drainage densities are predicted to occur on smaller cirque glacier…" You cannot say this based on your work. First, the word 'predict' suggests you've developed a model (e.g. regression) and are using it for prediction, which is not the case. But nor did you show that low drainage densities are correlated with small glacier size. In fact, the correlation between these two variables is only -0.1 and non-significant so you need to play down the role of glacier area. Lowest drainage densities are associated with glaciers with steep slopes and high average elevations (PC2 and correlation matrix). Perhaps they're not necessarily the cirque glaciers, just steep, high elevation glaciers – they could be valley glaciers. Base your statements on the evidence. Then you could go on to infer the processes involved in explaining the correlations, i.e. your interpretation. Make it obvious to the reader in your writing when you're moving from results to interpretation.

398-400. As above, you've not shown direct evidence for this statement. You have evidence that large drainage densities are on gently sloping glaciers at low mean elevations. Why are you talking about steep slopes and crevasses here?

400-402. Again, you must use your results as the basis of your discussion. You have evidence that large glaciers extend to lower elevations and have channels that don't extend to very high elevations (PC1 and correlation matrix). I cannot see evidence for a link between glacier area and drainage density. The opposite in fact as these have a low correlation. You have evidence that glaciers with high drainage densities have low slopes and have a low mean elevation.

404-408. The two sentences here are a contradiction to your general finding based on the statistical analysis of high drainage densities for glaciers with low mean elevation. You'd be better to discuss things based on your findings!

413. Channel inception in Fig A should read interception – I think I pointed out this mistake in an earlier review.

425. As I say above, I don't think glacier size can be discussed in relation to your conceptual model that is supposed to be based on your evidence. You could talk about steep, high elevation glaciers here but not 'small'. Categorising them as 'cirque' is an interpretation I assume?

428. 'valley glaciers are larger' Again, valley glaciers is an interpretation. Your evidence doesn't allow you to equate 7B glaciers with size. Your evidence suggests these are less steep with lower mean elevations cf. glaciers in 7A.

435-436. Again, link to your evidence. Avoid talking about 'area'. Here you're talking about very shallow gradient glaciers with very low mean elevation. These have the highest drainage densities. And they have the lowest incidence of crevassing (Fig 5h).

441. 'connectivity'? What does this refer to?

442. "...based on the locations of our mapped channel termini". Up to this phrase, this sentence sounded like a summary of your previous paragraph and conceptual model shown in Fig 7. This phrase throws a spanner in the works as you have not yet discussed possible links between locations of mapped channel termini and lag times between melt and discharge. I've read on and it appears as though you're going to talk about this wrt 2 case study glaciers. I suggest rewriting this sentence to introduce the work you'll present in this paragraph.

442-3. As I mentioned before, what exactly does this mean? How do you define 'average glacier'?

444. "... the largest glacier, Grosser Aletschgletscher (type B in Fig. 7)" OK, so your largest glacier is type B not C! This adds to my advice that you should not refer to glacier area as a control on drainage density (for which you don't have evidence).

451-2. where are the 'trace crevasses' in Fig 8? They are not labelled as such.

440-456. This paragraph is a little rambling and unfocussed. State at the outset what you're aiming to achieve here. It looks like you want to show us how different types of glacier (low drainage density to high drainage density - types A to C in Fig 7) have channel segments that

terminate in different ways. Correct? One way to have done that would have been to cluster your glaciers into the three types (based on their statistical attributes - either the raw variables or the PCs) and then looked at the number of channel termination types in each of the 3 clusters. You'd hypothesize that your steep, high elevation, low drainage density glaciers (type A) would have most channel termini ending in crevasses. Type C would have most ending by flowing off the snout or side. And type B would be somewhere in between. That would have provided you with the evidence you need to support all your statements relating to crevasses affecting channel / drainage characteristics. But you're not doing that.

Instead, you're picking a single case study example of Type B and Type C and telling us about their channel termini characteristics. For completeness why not do the same for a Type A glacier? Explain at the outset that this is what you're doing in this paragraph. In fact, would it be best to do all this BEFORE you present your conceptual model because in your description of the 3 types of glaciers you mentioned crevasses without really showing us any evidence that crevasses were relevant.

460. You refer to ' meltwater overtops the crevasse' . Does this make it a trace crevasse then?

464. You refer to  Oberer Theodulgletscher and Grosser Aletschgletscher but as I said above it'd be helpful to add an example of a type A glacier wouldn't it? Note the proper nouns (names of glaciers here) do not need to be prefixed by 'the'.

473. As I said before, you don't measure 'hypsometry' but you do show the role of glacier mean elevation so I'd refer to that here.

472-3. I'd agree with the statement:  "Hence, categorising glaciers based on their slope and mean elevation is beneficial because it provides insight into the anticipated drainage density of a glacier" because this is based on your evidence from your correlation matrix and PC2.
474-5. This statement about things providing insight into "channel pathways (i.e., sub-/englacially or proglacially terminating), and whether a higher amount of surface-to-bed meltwater transfer is likely" is a bit clumsy and is less well supported by your evidence but see my suggested way forward above wrt cluster analysis.

479 "...slope affects sinuosity". Clarify you're talking about channel slope. And avoid the word 'affects' as this implies causation whereas you just show correlation.

480. As well as Fig 5a you could refer to Fig 6 as this is supported by your correlation work. It's also interesting that this -ve association between channel slope and sinuosity remains apparent in PC5.

481. After stating the correlation between channel slope and sinuosity, you could interpret it and discuss processes. It's what you'd expect isn't it? Steeper slopes → lower sinuosity. On steep gradients, water has high energy and tends to take the most direct downslope path. This reduces the development of bends, keeping channels relatively straight. Gentler slopes → higher sinuosity. On low gradients, flow velocity and stream power are lower. Water has less ability to cut straight downslope and instead meanders laterally, forming more sinuous channels.

482-3. Fig 4e doesn't provide all the evidence to support the statement. It just provides examples of two sinuous channels on clean ice. Are there other channels not shown that are straight and on dirtier ice? I'd be tempted to delete this sentence.

483-5. The statement spanning these lines could also explain the sinuosity of at least one of the channels shown in Fig 4e as that channel seems to be sourced on debris. So this all seems a little weak.

486-500. I don't really see what you're trying to explain here. Channel slope controls stream power. Surely this promotes straighter channels. So your finding of more sinuous channels on gentler slopes makes perfect physical sense to me. Like you say, discharge also controls stream power so if discharge is higher through gentler slopes, then I suppose that could override the slope control, promoting straighter channels on gentler slopes and more meandering channels on steeper slopes. But you do not find this! Nor do you have any discharge data! So why try to explain it? Are you trying to explain Fergusson's and StG and M's findings, which would seem to buck what makes more physical sense? Haven't they already done that in their papers? Is it your job to do it here? I suppose you could just briefly explain that your findings are different to those from the Arctic glacier but explain briefly why the earlier work provides evidence which is contrary to what you might expect. But be brief.

503-5. Split the sentence to be clear. So say: "Previous research on supraglacial channel morphometry has focused predominantly on the GrIS (e.g., Smith et 503 al., 2015; Karlstrom and Yang, 2016; Yang and Smith, 2016; Yang et al., 2016, 2021, 2022). We find some similarities between the drainage patterns observed on Valais glaciers, and those on the GrIS."

505-6. I don't think you should resort to just picking out this one glacier in 4a. Your work has not focussed on how dendritic drainage patterns are. Can you not compare some of the quantitative channel and glacier characteristics and relationships between Valais and the GrIS, e.g. drainage density or sinuosity and their correlations with channel / glacier attributes?

507-8. "...some glaciers in Valais display parallel, weakly interconnected channel networks, likely due to insufficient distance for meltwater to converge into a single channel" Seems a bit anecdotal. Do you have strong evidence that this is prevalent on Valais glaciers? Doesn't the GrIS also display this in places?

509 "...these networks...". Which networks are you referring to here? The parallel weakly interconnected ones? So Yang et al 2016 find these follow Horton's laws on the GrIS? Why not calculate them for your drainage networks? Then you could properly compare Valais glaciers with GrIS. So far, you've not convincingly told the reader based on evidence whether and how the supraglacial hydrology differs between Valais glaciers and the GrIS.

511. You say 'trace crevasses exhibit a strong control on meltwater routing on Valais glaciers'. I do not believe you can conclude this from the evidence you've presented. The only mention of 'trace crevasses' in your results is on p18 wrt Oberer Theodulgletscher when you say "it is not known whether meltwater enters englacially or is routed on the glacier surface through trace crevasses (e.g., Fig. 8). Fig 8 makes no mention of 'trace crevasses' explicitly.

512. I'd delete ref to Antarctic ice sheet – see the subheading title!

514. should say 'Valais glaciers and the GrIS...'

516. I'm still unsure what the "average Valais glacier" is.

520 "appears to affect..."

520-1. Where did you show debris cover affects channel distribution? And what do you mean by 'distribution'? The only place I can think of is Line 295 onwards "Qualitative observations suggest that channel distribution and morphology are controlled by glacier structure and topography. For example, channels often occur along the interface between debris-covered and bare ice (e.g., Fig. 4b), particularly adjacent to medial moraines, where channels are confined to a topographic depression, commonly occurring at the confluence between two tributaries." This is only one example you've shown and I bet you could find at least one example of a channel on the GrIS flowing adjacent to a medial moraine. I don't think you have enough evidence to say anything meaningful here about the similarities or differences between Valais glaciers and the GrIS in terms of the effects of debris on channel 'distribution'. What about morphology? Any studies of sinuosity and role of debris on GrIS?

522-4. This sentence seems misplaced as it's for a glacier in Svalbard. Suggest delete.

524. What do you mean by "scale" here? And are you comparing Valais glaciers with the GrIS?

526. At the end of this para I don't have a clear sense of the similarities and / or differences between Valais glaciers and the GrIS in terms of the role of debris on supraglacial channels.

530. 'may' => 'are likely to continue to'

534 'large enough' => 'sufficient'

537-9. Seems a shame that you can't add anything to this based on your work. What would you expect from Fig 5c?

540-47. All this is not about the future evolution of channel systems. You need a separate section, or you need to reframe this section and provide a new heading. If you're talking about future work, you should discuss the possible imitations on your work of using July imagery rather than later season imagery.

**Conclusion**

550. I'd say 'From a sample of 285 glaciers…"

551-2. Here you talk about variability in glacier drainage density. This relates to PC2. Before this you could report the PC1 finding of high glacier area variability with large glacier area correlated with low minimum glacier elevation and low maximum channel elevation. And the possible reasons / implications of this. Of course, you'd need to have this as a discussion point earlier, as I suggested you could.

553. As well as ref to Fig 7 I'd use (or add) your correlation matrix Fig 6 here.

553-554. " The presence of channels is primarily dictated by a sufficiently supply of meltwater (i.e., large enough glacier area)" Note sufficiently => sufficient. But more importantly, this can't be a direct conclusion of your work as you do not measure melt water supply in your study. In your conclusions, explain what you actually find and then explain what your interpretation is. A large glacier area is not necessarily synonymous with a large supply of meltwater as you seem to be suggesting here. I assume this conclusion relates to your first result on lines 237-8 "Glaciers with channels (n = 85) have a larger mean area than glaciers without channels (n =

200) (mean area = 5 237 km2 vs. 0.6 km2) and all glaciers larger than 5.6 km2 contain channels > 0.5 m wide (Table 1, Fig. 3a)." If so would it be useful to remind us of this or refer to the relevant Fig?

554-555. "…and an uninterrupted distance for meltwater to coalesce (i.e., absence of crevasses)" Where did you show us that the 85 glaciers had fewer crevasses on them than the 200 glaciers that did not have channels, as you are suggesting here?

555-57. I can't see evidence for this sentence. You're suggesting that there are thresholds of glacier slope and glacier area beyond which channels don't form. Where is the evidence for this? Fig 5b does not show this. You also suggest that glacier slope and glacier size provide a limit on channel length. Again, where is the evidence for this? Fig 5b shows the relationship between channel slope and channel length of your sample. There is nothing about 'limits' here.

557. "…strong structural controls on channel distribution exist." It seems really odd to be talking about channel distribution as that is not really what your paper looks at. For the glaciers with channels on them, your paper focuses mostly on the relationships between channel and glacier characteristics, not about the presence or absence of channels in particular places.

557-8. "For example, trace crevasses have been observed to act as preferential meltwater pathways, resulting in channels forming perpendicular to ice flow." Where have they been observed? I think you give just two examples in your study (Fig 4c and Fig 8) and this point is mentioned almost in passing. It is hardly a major conclusion of your work. On line 452 you state: "Observations have shown that trace crevasses may act as a preferential meltwater pathway, often resulting in channels forming perpendicular to ice flow (e.g., Chen et al., 2024). " So this is a conclusion of another paper!

559. "Channels also commonly form parallel to medial and lateral moraines due to topographic confinement." Again, is this really a major conclusion of your analysis? Where do you show this? where is the evidence that this is widespread?

560. You refer to Fig 5a here but wouldn't Fig 3f be better? What is your definition of low? Can you say that x% are < some threshold?

560-61. "….highly sinuous channels are present, particularly on moderately debris-covered ice and lower-relief glacier termini." Again, where is the evidence.? See my comments on your section 5.2.

566-575. You may wish to reword or refocus this and the entire conclusion after thinking more about the Discussion section.

---

## Referee Report (RR3)

The latest round of revisions by Wytiahlowsky and others has resulted in an improved manuscript however, several of my previous concerns were not addressed or not sufficiently addressed by the authors. I have expounded upon my previous comments to help clarify my concerns. My original comments (black, italicized), the authors response (blue), and my new comments (black) are included below. The few other concerns I have are also detailed below.

Concerns

L44-46: Here Pitcher and Smith 2019 is cited as stating that channelized meltwater flow occurs on some glaciers and not others. This isn't an accurate representation of the 2019 review paper that instead argues for a full inventory of supraglacial channels to be mapped but does not present conclusions or findings that channels are absent on some glaciers and present on others.

(1) Citations were not added to support statement that most alpine glacier channels have widths smaller than satellite imagery resolution

> *L56-57: Here the authors state that remote sensing techniques have not been applied to mountain environments because of the small channel sizes there, but this is not supported adequately, how big are the channels typically found on mountain glaciers? This should be stated and references given. Currently only satellite resolutions are given which does not mean much if the size of the streams are not given as well.*
>
> We do not provide citations or state the size range of channels on mountain glaciers simply because there is a lack of research to provide us with a good figure. The introduction notes this uncertainty as we state "…the majority of channels are **likely** to be… below the resolution of the highest resolution freely-available satellite platforms". Hence, the benefit of this study is that we provide the first large-scale characterization of channels on mountain glaciers. Previous studies that have documented channels on mountain glaciers are largely focused on a single glacier in higher latitudes (e.g., Norway), hence we cannot confidently provide a quantification of channel size in the introduction.

You do not need to provide a definitive, universal value for channel size, but you should state the range of sizes that have been observed, such as from glaciers that have been widely published on in Norway, Switzerland (Ferguson 1973), Alaska/Canada (Dozier 1974), Iceland , etc. Even though the statement is reasonable, stating something is "likely" to be true is insufficient without the necessary rationale. If you can't support statements with a citation then the logical argument needs to be presented.

(2) Measured drainage density values were not added to the Discussion text.

> *L404: state the value for drainage density here.*
> This paragraph speculates how drainage density is likely to vary between glacier types; hence, we do not provide values for drainage densities. Additionally, this sentence mentions cirque-type glaciers, for which including a value might be misleading because we have little understanding of their surface hydrology. This is because most of their channels are likely to fall below the resolution of our mapping.

I still think a range of drainage densities observed should be stated in this section. For example, what constitutes a "high drainage density" or "low drainage density" in your area? Ranges for these vague statements should be given to aid readers in comparing their observations to yours for this region of Switzerland. Values from your dataset aren't misleading if they are properly stated with the caveats you mention above.

(4)

*L484: If there are sediment laden beds there, this should be said so explicitly and cited.* This sentence refers to a study that has modelled the excavation rates of subglacial sediment from beneath a glacier, hence we do not have observations to support this from our dataset. We instead use this study to theorise the effect that channel termini locations have on the amount of subglacially derived sediment entering into proglacial rivers.

My suggestion results from a gap in the logic of the paragraph. If no previous studies have identified subglacial sediments or rational for why they would be likely (which should be included if this exists), then a qualifier such as "if subglacial till is present, then …" should be included. The phrasing implies you know there are extensive subglacial sediments.

Additional Comments

Throughout   Remove extra space between all #s and the % symbol.
L184   this is not a complete sentence
L284-  it appears that there is a space between each number and the % that should not be there, unsure if this is a weird compiling error or included in the submission but it should be fixed at some point
L377: change "from mid-July 2020" to "acquired in" and add the specific date range in July 2020.
L442-443: The sentence now reads "Overall, we find that at the average Valais glacier, 80% of channels run directly off the glacier, while the remaining 20% terminate in moulins or crevasses". Previously the text stated that 72% of the mapped highest order channels were routed englacially or sub glacially with 25% running off the glacier. This point should still be included within the manuscript as the most important channels in the context of supraglacial hydrology are the highest-order channels.

L529-530: here the authors state "**we** suggest that supraglacial drainage networks may expand to higher elevations due to rising equilibrium lines (Leeson et al., 2015)". The "we" should be changed in this sentence as the authors work does not directly assess future supraglacial drainage evolution.

---

## Author Response (AR2)

**Distribution and characteristics of supraglacial channels on mountain glaciers in Valais, Switzerland. Holly Wytiahlowsky et al. 2nd review for TC by Ian Willis**

Thank you for the vast number of improvements to the paper following the suggestions by myself but also the other two reviewers. I noticed that many of our comments were similar.

The paper is greatly improved and is closer to being ready to publish in my view. I've been through the paper carefully and have a few remaining general comments and a few more specific detailed comments and questions.

The science is good, and the results are interesting, but my comments and suggested edits are to encourage you to work on articulating the results and implications of your work more carefully, precisely and logically than you do, with the ultimate goal that more people will read it and it will have a better impact. The Discussion was the main part of the original paper that needed improving and you've done a good job in doing so. But it remains the section that'll most benefit from more work.

General points

1. Throughout your paper, it would be clearer (and you could lose a lot of repetition), if you did not continuously tell us that your channels are 'large' or '> 0.5 m wide'. Tell us in your methods that you can only resolve channels above this size and that hereafter you refer to these simply as 'channels'. Then you can delete all subsequent references to '> 0.5 m' or 'large', as we know that when you use the term 'channels' it is an abbreviation for 'large channels > 0.5 m'.

   We originally added the mention of our mapping threshold throughout, following previous comments from reviewers that noted that we do not sufficiently address the presence of channels below our threshold. To avoid repetition, we have kept the mention of our mapping threshold when it is first mentioned in each section and removed it afterwards if it is not needed.

2. You use the word 'morphology' and 'morphometry' with respect to channels throughout the paper and it's not clear what you think the difference between these two is. Can you better define these two terms when you first introduce them and then use the correct word thereafter? Or do you see them as the same thing, in which case just use one term consistently.

   Amended – We refer to morphology when visually describing channel shape (e.g., sinuous or incised), and morphometry in relation to quantifiable measurements (e.g., length, quantified sinuosity). We now define each term when they are first mentioned (paragraph 2 and 4 of the introduction) and have corrected any instances of where these terms are used incorrectly.

3. There are places in the paper where you're comparing your work to 'ice sheets' but really you're just comparing with the Greenland Ice Sheet. I think you should alter all such sections to refer to the GrIS only and not give the impression that there are vast surface drainage networks on the Antarctic Ice Sheet!

   Amended – we have gone through the manuscript and replaced 'ice sheets' with the Greenland Ice Sheet when we are not also mentioning Antarctica. Section 5.3 has now been changed to a 'Comparison between Valais Glaciers and the Greenland Ice Sheet'.

4. Similar to above, there are places in the discussion and conclusions where you refer

generically to 'mountain glaciers' or 'valley glaciers'. You need to decide whether you think your results for the 85 Valais glaciers are universally applicable and can be generalised. Parts of your discussion suggest you do not think this and work is needed to apply your methods to other areas of the world. I would agree with this. In which case you should always refer to 'Valais glaciers' and not imply that your findings are necessarily applicable everywhere.

Amended – we have gone through the discussion and conclusion and replaced 'mountain glaciers' or 'valley glaciers' with Valais glaciers where we are specifically talking about the results from our study area. In some instances, we have left 'valley glacier' in when talking about the types of glaciers in Valais, or when referring to 'mountain glaciers' more broadly.

5. I would encourage you to proofread your work more than you do and have your other authors look over it, to spot and correct grammatical errors. Unfortunately, there were a lot of these, as some of the detailed line by line comments below show.

We have now gone through and corrected any instances of grammatical errors that we could find.

Detailed line by line comments

14. we map 1890 channels (to keep tenses consistent). Amended.

35. repetition of word 'rapidly'. Amended.

48 could say ' with potential to impact the suspended…". Amended.

54-5. Better English would be "…which has implications for subglacial water pressure, the onset of subglacial channelisation and, in turn, ice motion.". Amended.

72. Better to say "…to have a larger debris cover…" or "…to have a larger coverage of debris". Amended.

81/82. "…high concentrations of channels correlate with …" or "high concentrations of channels appear to be correlated with…". Amended.

91. Discharge is a rate so should say "discharge is a strong control…" [or "volume flow rates are…"]. Amended.

104. '…from glaciers…" plural! Amended.

105-6. "… determining the extent to which surface hydrological characteristics (e.g., channel, channel transport pathways) are uniform between" would be less ambiguous. Amended.

120. "…comparable with Switzerland's as a whole" [Add the apostrophe - it is not comparable with the size of Switzerland!]. Amended.

121. "…and varying crevasse densities…" Amended.

130. "found to contain" => 'containing'. Amended.

175. "has been found to be" => "is". Amended.

There are loads of examples of the above throughout your paper. Do a search for all the words 'found' and see if you can get rid of all the extra words. Amended.

191 remove ref to 'Canton' as you've already established that Valais is a canton. Amended – we have removed 'Canton' in a few places where it is not needed but have kept 'Canton' in for the Abstract, Introduction, Study Area, and for the first mention of it in the Conclusion.

205. delete 'width' or 'wide'. Amended.

217 'was assigned a code based…". Amended.

218 'The type of channel terminus..". Amended.

232 "…mapping had been repeated…". Amended.

233 "small enough for us to conclude that the original mapping provided a…". Amended.

234 – add "…and total channel length" to the end of the sentence. Amended.

234-5 "Both sets of mapping also clearly identified how the channels terminate. The primary source of uncertainty here, stems from knowing when to stop mapping up-channel.". Amended.

239. "found to contain" => "that contained". Also, shouldn't you replace "85" with "the" as the number of glaciers with channels is reported as a result on line 280? Amended.

248 "had been used". Amended.

260. Need a colon after 'three classes'. Amended.

263-275. There is a mix of present and past tense throughout this paragraph. Amended.

279 '…glaciers that had an area…" would be less ambiguous. Amended.

280 delete 'were found to'. Amended.

302-3. "and inclusive of the higher one". Amended.

306-314. "Glaciers without large channels (>0.5 m) are more likely to terminate at higher elevations (mean minimum elevation: 2936 m) compared to glaciers with channels (mean minimum elevation: 2797 m), which are often characterised by longer valley glacier tongues. Where glaciers support channels, they are more likely to have a
higher maximum elevation (mean max elevation: 3637 m) than glaciers without large channels (mean max elevation: 3555 m). Where channels >0.5 m are present, there is a mean drainage density of 2.4 km/km2 and a maximum of 15.2 km/km2. The latter was found on the Oberer Theodulgletscher, which is situated on a low slope plateau and has the lowest glacier slope angle in the dataset (13°) (Fig. 3c, Fig. 4a). To summarise, glaciers containing channels are larger, have lower mean slopes, and have a larger portion of their area at lower elevations compared to glaciers without large channels (>0.5 m)." [149 words]

could be improved and shortened to:

"Compared to glaciers without channels, glaciers with channels are more likely to have longer tongues that terminate at lower elevations (mean minimum elevation = 2797 m vs. 2936 m) and have higher maximum elevations (mean max elevation = 3637 m vs. 3555 m). Glaciers with channels have drainage densities ranging between 2.4 km/km$^2$ and 15.2 km/km$^2$. The latter was found on Oberer Theodulgletscher, which has the lowest glacier slope angle in the dataset (13°) (Fig. 3c, Fig. 4a)." [78 words].

This is just one example of where a verbose writing style could be made more succinct.

Amended – we have taken this suggestion on board with some minor alterations to ensure that the minimum and mean drainage densities are not confused.

317. should say 'top right'. In the Figure, is it possible to remove the white 'gap' in the top right part of (a)? Amended – the white gap related to the edge of the Swiss WMTS layer. We have filled this gap with a colour that is less noticeable.

326. Grammatical problems. This would be better:  "Few segments exceed 1,600 m, as the ablation areas of most glaciers are smaller than this.". Amended.

328 "". Amended.

340-3. This sentence still needs work. It's not quite right. Amended.

345-6. Wouldn't this be better for this entire sentence: "Qualitative observations suggest that channel distribution and morphology are controlled by glacier structure and topography." Amended.

350 'low' not 'lower'. Amended.

353. This title should come above the previous paragraph as that previous paragraph is exactly on this topic! Amended by changing the following suggestion.

354. This sentence does not capture what this section is about. You're not only investigating links between channel characteristics but also the glacier controls on the channel characteristics. Suggest you adjust the section heading and this sentence so that they match what is being done here. Amended.

359. I'd say 'shallow slopes' as 'lower slopes' could imply slopes at a lower elevation (i.e. opposite of upper slopes). Amended.

362. "lowest slope angles" => 'shallowest slopes". Amended.

366 I assume this should say 'ice' not 'glaciers'. Amended.

372. 'higher-order channel' not '…channels' 377 'each of which is" singular! Amended.

394-5. Correct the grammar here. Amended.

399. I think Principal Component Analysis should strictly be capitalised (as you've done on lines 424-5). Amended.

404 delete "identified to be". Amended.

435. You need at least a semicolon between "variance" and "instead". Amended.

446-7. What do you mean "are likely to be" You know whether they ae or not. Do you mean "tend to be" here? Amended.

451 Comma before 'which' Amended.

454. I'd put a comma after 'meltwater and before 'which'. End the sentence after 'channels' start a new sentence "Conversely, crevasses that have closed…" Amended.

478 and Figure 7. Do you mean 'inception' or do you mean 'interception'? The latter would

seem to make a lot more sense to me. "Inception" refers to the beginning or starting point of something, while "interception" means to stop or capture something that is moving or being passed. You say 'intercepted' on line 481 which seems correct. Amended – we did mean interception.

486-7 should say "location on a high elevation plateau". You can't say 'higher' as you don't explicitly state what you're comparing it with. Be careful of your use of comparative adjectives. Amended.

493-4. I'd delete this last sentence as it confuses things. You've already described the balance of glacier surface melt vs channel melt. Amended.

574 "compared to' => "and". Amended.

578 "seasonal variation in ice velocity" [or "summer speed up in ice velocity"]. Amended.

614 "observed to occur" => 'occurring'. Amended.

614 delete 'We find that' Amended.

616 Delete 'Our observations find that" [it should say "We find that" or "Our observations show that", but these and all similar phrases used throughout your paper are redundant]. Amended.

613-630. This discussion on the relationships between certain channel variables on the Valais glaciers (this study), vs. those on high Arctic glaciers reported by St Germain and Moorman and those on glaciers reported by Gulley et al (2009) [location not stated] is impossible for me to follow. I've read it through several times and cannot follow the logic. I'm not convinced the statements are logically connected. I'd encourage a rewrite of this whole paragraph. There is a confusion about what the evidence is supporting the statements. Is everything just based on correlation? This is not the same as causation of course. We need better distinction between correlation and causation and the latter should be linked to physical processes. Line 614 states that slope affects sinuosity which implies causation. You say that wider and more incised channels are more sinuous but where is the evidence for this? You don't measure width or incision. You say that such channels are likely to carry a higher discharge, but what is the evidence / logic / process base of this statement? You suggest it's because St G & M "attribute" higher sinuosity to higher discharge. But on what basis did they do that? All this from lines 614-619 seems poorly reasoned.

Then you state that St G & M find a positive association between channel slope and sinuosity for glaciers in the Arctic. But this is not observed for the temperate glaciers in your study. So, state what is observed. From Fig 5 it's a moderate -ve correlation, right? OK, so we have a discrepancy which needs explaining. Then you state that steeper channel slopes increase channel incision and ref Gulley et al. You should tell us that that study was also for glaciers in the Arctic. On what basis did Gulley et al reach that conclusion? Was it also correlation analysis like you've done? Then you refer to a positive relationship in your study between incision and sinuosity. But where is the evidence for this? Incision is not one of the variables you define in your correlation matrix. So this is just anecdotal evidence, correct? I don't think you can make any meaningful statements about relationships involving incision when you don't show any evidence for incision. On line 624 you state "These channels are likely to continue to evolve inter-annually due to their incised depth." But where is the evidence or what is the process base for this? And how is a statement about what may or may not happen in the future relevant to the discussion here, which is about explaining the present day relationships between channel variables? Then you state "Given this…the relationship between slope and sinuosity represents the conditions under which large channels can form (i.e., flatter, less crevassed regions) rather than the direct impact that slope has on sinuosity." But what do you mean by this? I can't work out how you would

logically deduce that from the previous statements.

Amended – we have rewritten section 5.2. In the rewrite, we have focused on describing our data and then hypothesise reasons for the discrepancy between our data and what St Germain & Moorman (2019) find in the Arctic. We also removed the paragraph focusing on channels on debris-covered glaciers and integrated it with information from the first paragraph. This has helped to streamline the content of paragraph 2 within this section.

651. What do you mean by 'appearance' here? Can you just loose that word as the subsequent discussion seems to be about sinuosity only. Amended.

652-654. The statement here "Channels that have a proximal debris source and those that run directly through debris cover tend to be more sinuous than channels on clean ice (Fig. 5c)" contradicts the statement in the results on lines 365-8 where you say "We find no noticeable difference in sinuosity between channels on bare ice and those on debris-covered glaciers (Fig. 5c). However, channels proximal to debris …are more likely to be highly sinuous than channels on bare ice or continuous debris cover". Which statement is correct?

Amended – both are technically correct; it was just poorly worded. What we were trying to convey is that channels on debris-free ice are slightly less sinuous than those on more continuous debris cover. However, a notable increase in sinuosity is observed for channels in areas with patchy debris cover compared to channels on debris-free ice and continuous debris cover.

654-6. I cannot reconcile this sentence with what is shown in Fig 4d. The sentence states that Fig 4d shows channels on clean glaciers but the non-sinuous channel in Fig 4d is on the same glacier as the sinuous channel. Are you talking about the two sides of the glacier tongue here?

The straight channel on this glacier is still on a thin strip of exposed ice, confined by debris, rather than flowing directly through debris-cover, which is happening for the channel on the right. However, we have removed this paragraph to avoid unnecessary information and instead integrated some mention of debris cover into the paragraph above.

658-9. This sentence isn't quite right. It is not the less sinuous channels that are funnelled through areas of bare ice. It is the water that would be funnelled. Or you could say the channels become straighter as they pass through the clean ice area. I don't see what your explanation for this finding is. Is it to do with the debris cover or is it just to do with the slope? If the clean ice zones of debris-covered glaciers have steeper slopes, then would this create the straighter channels? Or is it from the paragraph 613-30 that you're ruling out a slope control on sinuosity?

Amended – we have removed this paragraph and integrated key information within the paragraph above.

659-666. This reads like a literature review of previous work. Can you better explain how this work is relevant to your results? Can the processes that are implied with reference to the previous work explain any of your findings relating to the slopes and sinuosity relationships on glaciers or parts of glaciers with different amounts of debris?

Amended – this section has been rewritten.

666-668. This sentence doesn't seem relevant here. It reads as though it should go in the introduction of a different paper justifying why it'd be useful to assess the effect of debris within supraglacial channels on channel morphometry (which you don't do in this paper). Or it could go in a future work section of this paper.

Amended – we have removed these sentences.

668-671 doesn't seem relevant at all – I'd personally delete this.

Amended – these sentences have been removed from this paragraph and are now briefly mentioned in the second paragraph of section 5.3.

672-5. You say 'further insight' but further insight into what exactly? How is this sentence relevant to your work?

Amended – we have removed these sentences.

675-677. Could the sentence on lines 666-668 be merged with this and better articulated if you want to end this section with recommendations for future work that stem from the discussion about controls on channel morphometry?

Amended – we have removed these sentences.

714 You need to say "Previous research on surface channel morphology has…" [or some such]. Amended.

714. Just say 'ice sheets' not 'ice sheet settings'. Amended.

717. Need a full stop or semi-colon after 'environments'. Amended.

718. Grammar! Should say 'which are commonly observed". Also, why not just say "observed on ice sheets"? Amended

718-720. You say 'some glaciers' but is this all but the large glaciers? This would seem logical given what you then say. By 'less interconnected' do you mean 'less dendritic'? You don't been less anastomosing I assume, which is what 'parallel networks' implies. It's just a question of scale as to whether drainage systems can be described as dendritic?

We mean that some glaciers contain large, interconnected networks that exhibit a traditional 'dendritic' pattern, whereas other glaciers contain channels that flow parallel to ice flow and do not merge into one larger channel. We hypothesize that these parallel channels form when there is not a large enough distance for these channels to coalesce or a structural influence. Hence, parallel channels may be more common on smaller glaciers, or in areas where channels don't form until close to the ice margin. We have now rewritten the sentence to provide extra clarity.

723. What are 'flow-stripes'? These have not been mentioned previously so you can't introduce them here in the discussion. Amended – this has been removed.

724 I'd say "both for Valais glaciers and for ice sheets" [note you could say 'Valais glaciers" throughout most parts of your paper to remove a few words and make the sentences flow faster]. Amended.

726-7 I'd say "between ice sheets and Valais glaciers, for example debris may…formation on the latter". Then delete the 'For example' on line 727. You say 'channel formation' here but do you mean that? Or do you mean channel morphology [or morphometry]? Amended.

729. You refer to Rippin et al here but surely you should refer to your work first (and then bring in others' work if it supports your argument)? Otherwise, this is just a literature review.

Amended – we have rewritten the first part of this paragraph, and we now introduce the Rippin paper a little later, after having discussed our work first.

730 You mean 'surface channels' not 'surface melt' here. Your paper is not about melt. Amended.

731 delete 'are observed to'. Amended.

735. If you want a reference for the notion that lakes temporarily store water you could refer to this: https://tc.copernicus.org/articles/8/1149/2014/tc-8-1149-2014.html

736. And if you want one for the notion that lakes may refreeze you could use this: https://www.frontiersin.org/journals/earth-science/articles/10.3389/feart.2017.00058/full

But actually, if you were to delete lines 734-740 I think your paper would be improved as it remains more focussed on your Valais glaciers.

Amended – we have removed lines 734 to 740.

744. What do you mean by 'will develop'? Delete? Amended.

746 'will depend on the rates of' Amended.

747 I'd say 'associated' not 'subsequent' as higher melt will cause ELA to increase but also through the albedo feedback ELA increase will cause higher melt rates. Amended.

751 should you say 'and a reduction in drainage density'? Not 'or'. Because the example you give of drainage reconfiguration would reduce drainage density. Amended.

756 need a comma after '…evolution' Amended.

757-8. Poor phrasing – you could end the sentence "…future studies should repeat our work in such regions'' [or something along those lines]. Actually, your statement isn't quite correct as you do compare your findings to those of others in Arctic settings (St G & M; Gulley et al) although that paragraph did need improving. Amended.

760. Say 'on Valais glaciers'. Your previous sentence correctly noted that you can't necessarily extrapolate from the channels you delineate in your study to all mountain glaciers. Amended.

796 'found to contain' => containing'. Amended.

812. You could clarify that by terminating englacially you mean ending in a crevasse or moulin. Amended.

814 you refer to channels terminating supraglacially or englacially here. By supraglacially I assume you mean those that run off the glacier? Why not make that clear in the sentence above too? So, the sentence above would read:

"…have 80% of its channels run directly off the glacier (terminate supraglacially) while 20% would end in a crevasse or moulin (terminate englacially)." Would that make sense? It seems a little odd to refer to a channel that flows off the glacier as terminating supraglacially. Would 'proglacially' be better? Amended.

816. Refer to your Figure 7 at the end of this sentence. Delete the word 'observed' in the next. Amended.

816. Correct the English. Should say "differ from those in typical ice sheet…" [note plural therefore 'differs' not 'differ']. Amended.

818-20. Your final sentence is poorly articulated and could be stronger. This would be better, but you should decide. Amended – we have used the suggested sentence as it more clearly conveys the information.

"Compared to the GrIS, drainage networks on Valais glaciers are less developed due to smaller drainage areas and limited distance for channels to merge. Additionally, the down-glacier narrowing of the Valais glaciers further restricts the potential size of their drainage networks."

**Reviewer 3 comments**

The previous round of revisions by Wytiahlowsky and others has resulted in a much improved manuscript. The authors addressed my comments and concerns adequately and updated figures accordingly. I therefore only have a few comments, upon incorporation and in addressing the other reviewers I would recommend the manuscript for publication in The Cryosphere.

L56-57: Here the authors state that remote sensing techniques have not been applied to mountain environments because of the small channel sizes there, but this is not supported adequately, how big are the channels typically found on mountain glaciers? This should be stated and references given. Currently only satellite resolutions are given which does not mean much if the size of the streams are not given as well.

We do not provide citations or state the size range of channels on mountain glaciers simply because there is a lack of research to provide us with a good figure. The introduction notes this uncertainty as we state "…the majority of channels are **likely** to be… below the resolution of the highest resolution freely-available satellite platforms". Hence, the benefit of this study is that we provide the first large-scale characterization of channels on mountain glaciers. Previous studies that have documented channels on mountain glaciers are largely focused on a single glacier in higher latitudes (e.g., Norway), hence we cannot confidently provide a quantification of channel size in the introduction.

L57-59: Why would the principles that govern channel formation be different on alpine glaciers than on ice sheets just because large satellite imagery studies have not been conducted? As the authors state, the foundation of glaciology and glacial hydrology was preformed on alpine glaciers, and the mechanisms for supraglacial stream formation are quite well known. Furthermore, I don't think this study is investigating channel inception, rather, channel distribution in different environments (as stated in the following paragraph). This should be revised before publication.

Amended - we agree with the reviewer that this sentence does not best reflect the contributions of this paper and have changed it to state that we do not know whether the characteristics and distribution of channels on ice sheets are comparable to mountain glaciers. We have also provided more reasons why channel characteristics and distribution might differ between ice sheets and mountain glaciers.

L392: What was the size of the ablation area for these large glaciers that had supraglacial streams? Stream presence should be a function of ablation area size, this could be a linear function but should be supported in the text.

Figure 6 shows that there is not a simple linear relationship between ablation area size and drainage density. Our calculations use the snow-free area (i.e. the ablation area at the time of image acquisition) rather than the entire glacier area throughout (see section 3.2) to reflect the area available for channel formation, which likely closely resembles the size of the ablation area. In fact, glacier mean elevation ($\rho$ = -0.66, p = ≤ 0.001) and glacier slope appear to be stronger controls on channel density ($\rho$ = -0.46, p = ≤ 0.001). Hence, a large ablation area

might not contain a high density of channels if the glacier surface is steep (increased meltwater interception from crevasses) or if much of its area is at higher elevations (less surface melt) (see sections 4.4. and 5.1).

L401-402: What was the duration of the melt season in 2020? Another reviewer commended on this but the timing of imagery acquisition within the melt season should be addressed in the manuscript.
We only have monthly temperature data so our ability to provide detailed information on the exact duration of the melt season is limited. However, the average temperature is > 0˚C from April to September in 2020 so we have added this information into the methods (section 3.1). This section now provides a better overview of the climatic conditions that affected glaciers in Valais in 2020 (snow cover, temperature, and melt season duration).

L469: The authors addressed my previous comments by adding this text explaining how crevasses can exist while water filled. Thank you.

L568: the results of this paper do not directly explore how supraglacial drainage networks will change under future warming, consider rephrasing this sentence to as to not imply this is a direct consequence of the work by using "may" or something similar. Amended – we have changed 'will' to 'may' to clarify that this statement is speculative.

Minor comments

Figure 1: Does the glacier in panel b have a name? Amended – the glacier name (Glacier de Corbassière) has been added.

Figure 2: The addition of the entire glacier outlines look great. The figure would benefit from an indication of channel flow direction, particularly in ares where the channels deviate from glacier flow direction (particularly in panel e). Amended – We assume that this refers to figure 4, and water flow direction is now indicated on the figure by small blue arrows. In one instance, the water flow direction is unclear from the DEM and flow direction raster, so we have added a question mark here to avoid overinterpretation.

Fig 8: What is the elevation and coordinates for this photo? An inset like on Figure 2 would be helpful. Amended – we have added a panel on Figure 8 that shows where the main figure is located on the Oberer Theodulgletscher to provide more context.

L403: The rational for this statement should be appended to the end of this sentence. Amended.

L404: state the value for drainage density here.
This paragraph speculates how drainage density is likely to vary between glacier types; hence, we do not provide values for drainage densities. Additionally, this sentence mentions cirque-type glaciers, for which including a value might be misleading because we have little understanding of their surface hydrology. This is because most of their channels are likely to fall below the resolution of our mapping.

L423: a citation should be added here. This sentence hypothesises the effect that climatic warming will have on channel formation/density, which has not been researched outside of the GrIS (e.g., Leeson et al. 2015). Hence, we do not include a citation because we have no definitive answer, but we do include a citation (Marston, 1983) in the previous sentence which states how channels form (i.e., incision vs surface lowering) which provides the basis for our speculation.

L436: change "is also likely to" to "will likely". Amended.

L440: I would suggested adding a citation to Mejia et al., 2022 (GRL), while this work focuses in Greenland it provides direct observations of delays/prolonged supraglacial/englacial/subglacial residence time when supraglacial streams are routed en/subglacially.

We have added a reference to this sentence that discusses the storage of meltwater within glacial systems, but have instead chosen a reference that refers to mountain glaciers (Clason et al., 2015), which is more comparable to Valais glaciers.

L441: change "glacial system" to "glacial drainage systems". Amended.

L443: comma auger supraglacially. Amended.

L454: do you mean the peak has a lower amplitude? Amended – the sentence has been made clearer to note that we mean lower amplitude.

L458: change positions to "drainage systems". Amended.

L460: location-> presence. Amended.

L484: If there are sediment laden beds there, this should be said so explicitly and cited. This sentence refers to a study that has modelled the excavation rates of subglacial sediment from beneath a glacier, hence we do not have observations to support this from our dataset. We instead use this study to theorise the effect that channel termini locations have on the amount of subglacially derived sediment entering into proglacial rivers.

L548: what is a flow stripe? A medial moraine? A flow-stripe (or longitudinal foliation) is an elongated structure on the glacier surface formed by ice flow and has different mechanisms of formation to a medial moraine. We have now removed this example as it was mentioned in relation to the Chen et al. (2024) paper but is confusing as it is not a feature we note elsewhere in the paper.

L562: consider adding citations to Andrews et al., 2018 and Mejia et al., 2021. We have removed this section following a suggestion to improve the focus of the paper.

Andrews, L. C., Hoffman, M. J., Neumann, T. A., Catania, G. A., Lüthi, M. P., Hawley, R. L., Schild, K. M., Ryser, C., & Morriss, B. F. (2018). Seasonal Evolution of the Subglacial Hydrologic System Modified by Supraglacial Lake Drainage in Western Greenland. Journal of Geophysical Research : Earth Surface, 123(6), 1479–1496. https://doi.org/10.1029/2017JF004585

Mejia, J. Z., Gulley, J. D., Trunz, C., Covington, M. D., Bartholomaus, T. C., Xie, S., & Dixon, T. H. (2021). Isolated Cavities Dominate Greenland Ice Sheet Dynamic Response to Lake Drainage. Geophysical Research Letters, 48(19). https://doi.org/10.1029/2021GL094762

Mejia, J. Z., Gulley, J., Trunz, C., Covington, M. D., Bartholomaus, T. C., Breithaupt, C. I., Xie, S., & Dixon, T. H. (2022). Moulin density controls the timing of peak pressurization within the Greenland Ice Sheet's subglacial drainage system. Geophysical Research Letters, 49, 1–13. https://doi.org/https://doi.org/10.1002/essoar.10511864.1

---

## Author Response (AR3)

**Anonymous reviewer**

The latest round of revisions by Wytiahlowsky and others has resulted in an improved manuscript however, several of my previous concerns were not addressed or not sufficiently addressed by the authors. I have expounded upon my previous comments to help clarify my concerns. My original comments (black, italicized), the authors response (blue), and my new comments (black) are included below. The few other concerns I have are also detailed below.

We appreciate the reviewer's positive assessment of the revised manuscript and thank them for their constructive feedback. In this version, we have made every effort to thoroughly address the reviewers' remaining concerns, and we hope that this is clearly reflected in the updated manuscript.

**Concerns**

L44-46: Here Pitcher and Smith 2019 is cited as stating that channelized meltwater flow occurs on some glaciers and not others. This isn't an accurate representation of the 2019 review paper that instead argues for a full inventory of supraglacial channels to be mapped but does not present conclusions or findings that channels are absent on some glaciers and present on others.

This citation was used in reference to the spatial distribution of channels not being known; however, we acknowledge that this wording may imply that the citation directly discusses why the distribution of channels may vary between glaciers. We have now reworded this sentence to clarify that this citation is used because it explicitly states that the lack of information on channel distribution is a key research gap that needs to be addressed.

(1) Citations were not added to support statement that most alpine glacier channels have widths smaller than satellite imagery resolution

> L56-57: Here the authors state that remote sensing techniques have not been applied to mountain environments because of the small channel sizes there, but this is not supported adequately, how big are the channels typically found on mountain glaciers? This should be stated and references given. Currently only satellite resolutions are given which does not mean much if the size of the streams are not given as well.

You do not need to provide a definitive, universal value for channel size, but you should state the range of sizes that have been observed, such as from glaciers that have been widely published on in Norway, Switzerland (Ferguson 1973), Alaska/Canada (Dozier 1974), Iceland , etc. Even though the statement is reasonable, stating something is "likely" to be true is insufficient without the necessary rationale. If you can't support statements with a citation then the logical argument needs to be presented.

We have modified this sentence to note that supraglacial channels on mountain glaciers tend to be less than a metre wide and have added relevant citations that include some measurements of channel width. We now cite Knighton (1972, 1981, 1985) and Ferguson (1973), whose papers include width measurements for a range of channels.

(2)  Measured drainage density values were not added to the Discussion text.

> *L404: state the value for drainage density here.*

I still think a range of drainage densities observed should be stated in this section. For example, what constitutes a "high drainage density" or "low drainage density" in your area? Ranges for these vague statements should be given to aid readers in comparing their observations to yours for this region of Switzerland. Values from your dataset aren't misleading if they are properly stated with the caveats you mention above.

While this section focuses on our conceptual schematic, we have now included some example drainage density values in the Discussion based on additional analysis carried out in response to comments from another reviewer. Specifically, we performed a cluster analysis and an additional Principal Component Analysis, which helped group our study glaciers based on their characteristics and provided further support for our conceptual schematic. Although we have added summary statistics, including drainage density values, for each glacier type in the Results section (see final paragraph of Section 4.4) and included select examples in the Discussion, we note that these values should be interpreted with caution. Drainage density can vary significantly depending on image resolution and mapping method (for example, thresholds for minimum channel width), and therefore the values are not easily comparable across studies unless identical methodologies are used.

(4)
> *L484: If there are sediment laden beds there, this should be said so explicitly and cited.*
>
> My suggestion results from a gap in the logic of the paragraph. If no previous studies have identified subglacial sediments or rational for why they would be likely (which should be included if this exists), then a qualifier such as "if subglacial till is present, then …" should be included. The phrasing implies you know there are extensive subglacial sediments.

We have now added 'If subglacial till is present…' to the beginning of this sentence to help clarify that we are proposing a hypothesis rather than discussing direct observations.

**Additional Comments**
Throughout - Remove extra space between all #s and the % symbol. The space between the numbers and the % is in accordance with the journal style guidelines, which follow the SI brochure's guidelines of 'a space separates the number and the symbol %'.

L184     this is not a complete sentence. Amended – we have rewritten the sentence to say 'Examples of supraglacial channels are shown on Glacier de Moiry (a-b) and Allalingletscher (c-d).'

L284-  it appears that there is a space between each number and the % that should not be there, unsure if this is a weird compiling error or included in the submission but it should be fixed at some point. The space between the numbers and the % is in accordance with the journal style guidelines, which follow the SI brochure's guidelines of 'a space separates the number and the symbol %'.

L377: change "from mid-July 2020" to "acquired in" and add the specific date range in July

2020.

In addressing this comment, we revisited the Swiss geoportal and noticed an error regarding the acquisition date stated in the paper. We have since discovered that the flight lines that make up the swisstopo orthophoto from 2020 were acquired on several dates in August 2020 (5th, 7th, 8th, 15th, 21st & 27th), with one small area being covered on the 4th of September. As there is overlap between all these flight lines, it is challenging to definitively determine which part of the orthophoto comes from which flight line.

Overall, this  helps to address an earlier comment regarding the use of imagery from earlier in the melt season, when channels are not fully developed (July). We have now corrected the manuscript, and any mention of mid-July has also been removed throughout.

L442-443: The sentence now reads "Overall, we find that at the average Valais glacier, 80% of channels run directly off the glacier, while the remaining 20% terminate in moulins or crevasses". Previously the text stated that 72% of the mapped highest order channels were routed englacially or sub glacially with 25% running off the glacier. This point should still be included within the manuscript as the most important channels in the context of supraglacial hydrology are the highest-order channels.

Amended – These figures have been added back into the manuscript. We chose to remove them because they are for all mapped non-tributary channels rather than per glacier, and this figure may be confusing because it predominantly contains channels from a few large glaciers that contain a very high portion of englacially terminating channels. We have now provided extra contextual information to try to avoid such confusion.

L529-530: here the authors state "**we** suggest that supraglacial drainage networks may expand to higher elevations due to rising equilibrium lines (Leeson et al., 2015)". The "we" should be changed in this sentence as the authors work does not directly assess future supraglacial drainage evolution. Amended - We have modified this sentence to state that 'previous research suggests…' instead of 'we'.

**Review by Ian Willis**

***General Comments***

I have commented positively on this work in my previous reviews and those points still stand. The paper presents a highly original data set, much of it obtained from meticulous manual digitising of tens of orthophotos. The descriptive statistics of the channel and glacier characteristics and their relationships (analysed using correlation and PCA) are all well done. It's good to see the improvements that have been made to the manuscript since my last review. For all these reasons I do think this work deserves to be published.

In my last review, I showed how the Discussion section needed the most work and it's good to see that in the latest version of the paper many changes have been made to the Discussion. I'm sorry to say, though, that I still don't think the paper is quite ready to publish because the Discussion and Conclusions still need some work. A fundamental problem is that many of the statements made in the Discussion and Conclusion do not clearly follow on from the data presented.

I think a major problem I'm having is that the Results focus on good robust quantitative

analysis of glacier and channel characteristics and their relationships (4.1, most of 4.2, 4.3 and 4.4) with a short weaker section on 'qualitative observations' towards the end of section 4.2 (6 lines of text and Fig 4) but the Discussion and Conclusions focus a lot on those 'qualitative  observations' and don't draw out some of the interesting things from the quantitative work.

Given the importance of these 'qualitative observations' I think they should be defined and described better and more thoroughly. Looking at lines 295-301, the 'qualitative observations' ostensibly show examples of how 'channel distribution and morphology' are controlled by 'glacier structure and topography'. These terms are poorly defined. Channel distribution is rather a vague term and not precisely defined but I take it to mean where they are located on a glacier vs where they are not]. Channel morphology is defined on line 49 as "channel shape and structure" which is slightly confusing as when I think of channel shape I think of the shape of a channel cross section. 'Planform shape' would be a better description as this is what is meant (i.e. whether channels are straight or meandering (sinuous). Channel structure is never defined. What is this?  'Glacier structure' features twice in this paragraph and nowhere else in the paper and is never defined. It seems to refer to patterns of crevasses, which is not what I'd call 'glacier structure'. 'Glacier topography' is not defined but I think this is probably clear to most readers, i.e. the pattern of surface elevation and its derivatives (e.g. slope).

Despite it being suggested that both 'channel distribution and morphology' are controlled by both 'glacier structure and topography', we're just given one example of how 'channel distribution' is controlled by 'channel topography' (Fig 4b), which is a channel occurring 'along the interface between debris-covered and bare ice'. We're also given an example (Fig 4c) of how 'glacier structure' influences 'channel morphology' which is a straight channel following a 'trace or shallow' crevasse. Finally, we're given another example of how 'glacier topography' influences 'channel morphology' (Fig 4e), which is a sinuous channel at low elevations on a flat part of the

glacier towards it's terminus. [We're also told that such channels tend to occur on 'large glaciers', which is not defined by 'glacier structure and topography' but is anecdotally thrown in here].

I'm sorry to labour the point here but this paragraph is weak and yet a lot seems to hang on it in terms of your Discussion and Conclusions. How to improve it? First, you'd need to very precisely define the terms. Second, the reader would need to have an idea of how representative these 3 examples are of all the glaciers that you've looked at. Otherwise, how do we know that you're not just 'cherry picking'? Personally, I'd recommend removing this section as it is. Given that you have done all the robust quantitative work, I'd strongly recommend you base your Discussion and Conclusions on that evidence and perhaps show these examples in Fig 4 (and more if you like) of glaciers that show the typical behaviour that you see from the analysis of your large sample. Your quantitative work cannot tell us anything about how 'channel distribution' relates to 'glacier structure or topography' but it can tell us how 'channel length', 'drainage density', 'sinuosity' are related (or not) to the glacier variables of 'area', 'slope' and 'elevation'. Note you cannot tell us anything about how 'glacier structure' [by which I assume you mean presence and orientation of shallow crevasses] influences channel characteristics as you do not have this information about crevasses in your large data set. What you can tell us about is that 'crevasse extent' influences 'drainage density' as that's shown in Fig 5h.

You'll see in the line by line comments, when you get to the Discussion and Conclusions that I'm pointing out all the instances where I do not see evidence to support the statements you make. I would recommend a fairly root and branch edit of the Discussion and Conclusions so that your statements more firmly follow on from the robust evidence you have.

I think your correlation matrix and your PCA show some very interesting findings, only some of which you draw out. For example, your first PC shows that the greatest variability in your data set concerns glacier area, glacier elevation and channel maximum elevation. The way these variables are related (+ve / -ve), which is also shown in the correlation matrix, is interesting and could be interpreted. I think I give a possible interpretation in the line-by-line comments below. Similarly, your second PCA shows the 2$^{nd}$ greatest variability in your data set is to do with drainage density, mean glacier slope and mean glacier elevation. Again, the way the variables are related in their contribution to PC2 (and in the correlation matrix) is really interesting and supports the main conclusion that you wish to make (and which you've depicted in your conceptual model). Fig 5h also supports this. As far as I can tell you have no evidence that 'glacier area' may affect drainage density (r = -0.1) although you do have evidence that glacier area may affect sinuosity (r = +0.29). These points based on your evidence need articulating clearly.

Another thing you'll see form the line by line comments is that you need to be careful to separate out correlation (which you have evidence for) from causation (which is your interpretation). Be careful and consistent throughout your paper on this point. I think I've given examples below on how you can change things or where you should.

I hope my comments are helpful. As I've said before, they are designed to improve the paper so that it's intelligible and people will want to read it, understand it, and hopefully reference it.

We thank the reviewer for their continued efforts to help strengthen our manuscript. Hopefully it is clear that we have made a substantial effort to address the majority of their suggestions in full. The primary instance where we do not fully address the reviewers' suggestions relates to qualitative results. This is because we believe there to be value in these observations, but we do agree that it may be more useful to tone these down and focus more on our statistical analysis, which is what we have done in this revision. The only minor comments that we have not addressed relate to choice of terminology and a suggested figure edit. We have summarised the key changes below and responded to line-by-line comments. When a common theme emerges among comments in a section, we respond at the end of that section to avoid repetition.

The primary changes made during this revision include the addition of new analysis. Notably, we now include a cluster analysis to provide further insight into how glacier properties vary within our dataset. This has helped provide support for our conceptual schematic, as each cluster is typically characterised by different drainage densities and channel termini locations. We also renamed the glacier types in our conceptual schematic to reflect the properties of the glaciers in each cluster.

We have rewritten parts of the discussion to focus more on the data presented in the results section. This included removing much of the discussion surrounding individual glaciers and replacing it with discussion of our cluster analysis, as this uses our large dataset and helps to

better substantiate our interpretations. Changes were also made to the channel morphology section to focus more on our own data rather than the research of others.

We also edited the Valais and GrIS section to provide a clearer comparison between both environments. It was suggested that it would be useful for us to provide more of a quantitative comparison; however, unfortunately, there is not much directly comparable data. For example, due to the differences in mapping resolution, it is not useful to directly compare drainage density values (e.g. studies using coarser resolution imagery/measurements will be biased towards lower drainage densities and vice versa). We would have liked to compare bifurcation ratios, but we were unable to automate the calculation of stream order due to the complexities (e.g., frequent bifurcations) of our mapped channels. Hence, this section is written in a more speculative manner to reflect the greater uncertainty surrounding our interpretations.

Lastly, the conclusion has also been rewritten to focus more on our interpretations of the data presented in the results section.

**Line by line comments**

**Introduction**

45. suggest "…why surface meltwater becomes channelised on some glaciers but not others…" Amended.

55-7. correct to "…as most channels on mountain glaciers are likely much smaller than those on the GrIS and therefore fall below the resolution of even the highest-resolution freely available satellite platforms…" Amended.

58-59. Better to say: "…comparable to those on mountain glaciers. Mountain glaciers are characterised…" [because 'the latter' strictly refers to "the channels on mountain glaciers"] Amended.

70-72. "Where channels occur, they are often reactivated annually" repeats lines 67-8. And "…deeply incised channels suggested to be a product of high discharge.." is similar to lines 64-5 as high meltwater production => high discharge. Amended – we have removed the repetition of annual reactivation and discharge.

79-81. Better to say: "However, much of what we know about supraglacial channels was established from observations of a small number of individual glaciers, especially those that are cold or polythermal (e.g., Knighton, 80 1972, 1981, 1985; Gleason et al., 2016; St Germain and Moorman, 2019). Amended.

91. I'd say "glacier surface characteristics" Amended.

92. Can you briefly summarise what the 'qualitative observations' are? Although if you follow my advice in my general comments this may be removed. Amended – we have briefly elaborated on this. We should add that we still see value in adding some discussion of qualitative observations and note that this is not uncommon in the discipline.

**Study location**

100 Swiss canton [not capitalised here as not a specific canton, e.g. Valais Canton] Amended.

101 I'd delete the 2nd 'area' so "… a maximum of 77.3 km2…" Amended.

118. delete 'right'? Isn't G Aletsch just in the centre? Amended.

**Methods**

154-5. I'd say "This is because they are likely too small to form channels large enough to be detected in our imagery, and because many of the small glaciers listed in the Swiss Glacier Inventory (SGI2016) do not meet the criteria for classification as glaciers (Leigh et al., 2019). Amended – we have followed your suggestion but kept 'unlikely' instead of 'do not' to acknowledge the uncertainty.

174 "…main channels were mapped…" Amended.

178-179. confusing to have 'terminus' for channels and for glacier. Suggest use 'snout' for glacier terminus throughout paper. [Or you could use terminus for glacier and terminal for channel]. Also, later you talk about a channel running off (terminus or periphery). Would it better to refer to that here. So you could alter your list here to say: The type of terminus was assigned to each channel, which was one of: running off the glacier snout, or off the glacier side, or terminating in a moulin, crevasse, or lake, or adjoining another channel, or disappearing beyond the image resolution (i.e.., the terminus was not visible and could not be inferred confidently).

Note I've changed 'periphery' to 'side' here as 'periphery' to me would include the front (i.e. snout). If you like these suggested changes, check your entire paper and make the relevant changes.

We retained 'terminus' because it is an internationally-recognised and widely used term in the glaciological literature. The word "snout" (meaning the projecting nose and mouth of an animal, especially a mammal) is sometimes used but may confuse non-English speakers. However, the reviewer's point is well made, and we have gone through every mention of 'terminus' in the manuscript and made sure that it is clear whether we are referring to the channel or glacier terminus.

184. "Supraglacial channels on Glacier de Moiry and (c-d) on Allalingletscher…" Amended.

186. Should assessment be plural? Also, put in brackets the initials of the 'individual'. I assume (HW). The error assessment is not plural as it was a singular assessment. To clear up any potential confusion about our mention of 'on two occasions' we have reworded the following sentence to clarify that we re-mapped Rhonegletscher once after our original Mapping. The initials of the mapper are provided in the CRediT statement.

Tell us approx. what the time period was. E.g. "…was conducted over an approximately 6 month period."Amended – We have checked the file creation and export dates and can confirm that mapping took place from the 15th of June to the 18th of December 2023 (6 months and 3 days). We have added your suggested text, as it does reflect the duration of our mapping.

191. "…of each channel length and glacier's drainage density…" The sentence states that the error margin is small enough to conclude that we provide a good representation of each glacier's drainage density. We feel it would be contradictory to add 'channel length' here, because the following sentence goes on to acknowledge that most of our uncertainty comes from channel length. While the overall difference in drainage density values between our original and repeat mapping is small, the error for individual channel length may differ. Hence, we do not add 'channel length' in this sentence because we do not have the same confidence in those values as we do with drainage density. Therefore, we consider it better to refer to them both separately, as we currently do in the manuscript.

192. Could delete 'here' Amended.

195-6. This is unclear to me. Do you mean you mapped the up-glacier channel limit to where you were confident the channel existed? Note past tense 'were'. Amended – Yes, that is what we meant. We have now edited the text.

199. Helpful to reiterate "..the 85 glaciers…" Amended.

204. Regarding segment length, wasn't this derived from the orthophotos not the DEM as implied here? Regarding straight line distance, is this in 2D plan (so from the orthophotos) or in 3D (so from the orthophoto and DEM)? Amended – We have modified the text to state that the segment length is the geodesic length and calculated using ArcGIS tools that do not use a user-inputted DEM, and straight-line distance does use a DEM because it is in 3D and uses the minimum and maximum channel elevation values from the DEM.

209 change 'which' to 'and' Amended

201 could delete 'record' Amended.

199-219. There's confusion I think between these paragraphs about how you calculate elevation and slope. In the first para you say the DEM was used to calculate both. In the 2nd you say that slope was calculated for the snow free area so calculated from the DEM. But what about elevation variables? Did you also calculate that for the snow free part? Amended – The first paragraph has been reworded so that it is clear that we extracted channel elevation values from the DEM and use these to calculate other metrics such as channel slope and straight-line distance. This should hopefully prevent any confusion as we've separately mentioned how we calculated channel and glacier slope. We have also rewritten the 2nd paragraph to more clearly state which data comes from where. Notably, it is clearer that the elevation values are from the 2015 glacier inventory data, and we do not separately recalculate elevation values for the snow-free portion. We used elevation data for the full glacier, as this allows us to characterise the glacier in its entirety, which is more difficult to do when just using the elevation of the snow-free area at a time-stamp.

226-27. given what you said earlier you could reorder these and say "drainage density, glacier area, aspect, minimum elevation, mean elevation, maximum elevation, and mean

slope of the snow-free area." Amended – we have removed the mention of 'glacier' from this list and reordered it so the snow-free area and mean slope are mentioned together.

228-233. Could be abbreviated and improved to say: "A one-way ANOVA was performed to test the significance of the relationship between the three debris-cover classes and sinuosity. In addition, a Principal Component Analysis (PCA) was conducted to examine relationships among variables and identify the main drivers of variance in the dataset, with the data normalised to enhance pattern detection." Amended.

**Results**

243 I'd change 'on' to 'of' and say 'km2, with a maximum…" Amended.

264. It would be helpful to indicate where these glaciers in Fig 4 are on Fig 1. This suggestion is well made but would overcomplicate Fig. 1 as we already colour-code glaciers in the figure and have letters showing the location of panels. Additionally, now we have shifted our focus away from a few specific glaciers.

286. change 'terminus' to 'snout'? See earlier comment. Change throughout if you agree.
289. change 'periphery' to 'side'? See earlier comment. Change throughout if you agree.
We have kept this as terminus (see our response to your comment about line 178 – 179.

292. What do you mean here? what is 'the average glacier'? Also, have you defined 'terminating proglacially already? Is this the same as having water running off the glacier snout or off the glacier side? Amended – we have provided additional clarification in this sentence. We refer to the proportion of channels that terminate englacially versus run directly off the terminus at each glacier are averaged across the dataset. This value includes channels that run off the glacier side/periphery and at the glacier terminus, which is clarified earlier in that paragraph when we refer to terminal segments.

305. 'variables to test for…" Amended.

306. 'affect' implies definite causation, but of course you're doing correlation (not causation). You could say " 'may affect' or 'are related to' here. Amended.

312-13. "The 'debris' class generally contains more sinuous channels than…" is more correct Amended.

341. You can't say 'controls' here as this implies causation which you don't know from correlation analysis. From here and for the rest of our paper you'll need to be careful with this. You'll need to be clear when you're sticking to the facts (correlation) and when you're making inferences from them about cause-effect or processes. You could say "'associations between' or 'possible controls' here. Amended.

345. 'control' No Amended.

355. What does " high channel sinuosity can in part be explained by multiple weak correlations" mean? I'd just delete this. Amended – we have made this clearer.

358. 'controlled' No Amended.

361. 'relationships' [plural] Amended.

361-2. did you use all the channel and glacier variables in your PCA? If so state that here. Amended – yes, we used the same channel and glacier variables included in our correlation matrix (Figure 6), which is the same as those listed in section 3.3. We have now also modified section 3.3 in the methods to make this clear.

362-367. Capitalize 'Principal Component' when referring to a specific PC (1, 2 , 3 etc). Or just abbreviate to PC Amended.

366. Tell us what % PC1 and PC2 explained individually (not just together). Amended – we have now given the individual percentages for PC1 & 2.

**Discussion**

**Section 5.1**

377. I assume you want to say "Previous work has suggested that the presence of visible channels is primarily controlled by…" ? Because you do not measure 'meltwater supply' in your paper. Or is this interpretation based on your results? It'd be best to start each discussion point (so 5.1, 5.2, etc) with explicit reference to your findings, then your interpretation of those findings (make it clear where results end and interpretation begins) and you could also discuss your findings / interpretation in the context of previous work. Amended: see response 2.

Section 5.1 - This subsection is headed "Controls on the spatial distribution of channels" . This is a little unclear as a title, especially as you haven't really shown results on this topic. Your paper is not about why channels form in some places and not others on a glacier. Amended: see response 1.

382. Your statement "glacier area controls much of the variability within the dataset (Table A1)" is correct as glacier area dominates PC1. You could refer to PC1 after Table A1 to clarify this. Amended – this section has been rewritten to focus on our results.

383-4. Your statement "… albeit with large variation in drainage density." Is rather thrown away here. Drainage density dominates PC2. So why not explicitly say that. Amended – see response 3.

384. When you say "This variation is in part attributed to glacier slope…" what exactly do you mean? I can see that PC2 is dominated by drainage density (+ve) and glacier slope (-ve). Is this what you're talking about? Refer to the evidence for your statements. Note glacier mean elevation (-ve) also contributes to PC2. Why not mention this? What this means is that a high source of variability in your data set comes from these three variables contributing to PC2. Glaciers with high drainage density tend to have low slopes and are situated at low elevations. Use your results to discuss them. Amended – see response 3.

384-5. Your statement "…together with ice flow velocity, governs the crevassed area of a glacier" is either from previous work or it's your interpretation of your results. You do not analyse ice flow velocity. Nor do you show the relationship between glacier slope and crevassed area. You need to more clearly discuss your results and explain what your evidence shows and how you interpret it. Amended – see response 3.

388-92 'Channel formation is also governed by glacier hypsometry…' is a little confusing as 'hypsometry' is not one of the variables you quantified and investigated. Similarly, you talk about "glaciers containing a larger proportion of their area at lower elevations', which I agree is to do with hypsometry, but again, you don't measure or report this. What I can see is that the 3$^{rd}$ most important variable contributing to PC2 is glacier mean elevation, and I can see from the Table A1 and the Fig 6 correlation matrix that drainage density is inversely correlated with glacier mean elevation, so lower elevation glaciers have higher drainage densities. Refer carefully to your data and evidence and make statements that you can support. Then interpret. The last two sentences of this para don't really contribute meaningfully to the discussion. Amended: see responses 2 to 4.

393. The word 'hypsometry'. Again, this is not something you explicitly focussed on in our paper so I'd remove reference to it. However, here you're focussing on explaining variation in drainage density (so we're with PC2 still) and you could also draw on evidence from the correlation matrix. It is glacier slope and glacier mean elevation that correlate with drainage density and contribute to PC2.

394. "The lowest drainage densities are predicted to occur on smaller cirque glacier…" You cannot say this based on your work. First, the word 'predict' suggests you've developed a model (e.g. regression) and are using it for prediction, which is not the case. But nor did you show that low drainage densities are correlated with small glacier size. In fact, the correlation between these two variables is only -0.1 and non-significant so you need to play down the role of glacier area. Lowest drainage densities are associated with glaciers with steep slopes and high average elevations (PC2 and correlation matrix). Perhaps they're not necessarily the cirque glaciers, just steep, high elevation glaciers – they could be valley glaciers. Base your statements on the evidence. Then you could go on to infer the processes involved in explaining the correlations,
i.e. your interpretation. Make it obvious to the reader in your writing when you're moving from results to interpretation. Amended: see responses 2 to 4.

398-400. As above, you've not shown direct evidence for this statement. You have evidence that large drainage densities are on gently sloping glaciers at low mean elevations. Why are you talking about steep slopes and crevasses here? Amended: see responses 2 to 4.

400-402. Again, you must use your results as the basis of your discussion. You have evidence that large glaciers extend to lower elevations and have channels that don't extend to very high elevations (PC1 and correlation matrix). I cannot see evidence for a link between glacier area and drainage density. The opposite in fact as these have a low correlation. You have evidence that glaciers with high drainage densities have low slopes and have a low mean elevation. Amended: see responses 2 to 4.

404-408. The two sentences here are a contradiction to your general finding based on the statistical analysis of high drainage densities for glaciers with low mean elevation. You'd be

better to discuss things based on your findings! Amended: see responses 2 to 4.

413. Channel inception in Fig A should read interception – I think I pointed out this mistake in an earlier review. Amended.

425. As I say above, I don't think glacier size can be discussed in relation to your conceptual model that is supposed to be based on your evidence. You could talk about steep, high elevation glaciers here but not 'small'. Categorising them as 'cirque' is an interpretation I assume? Amended – we have renamed 7A to 'Steep, high elevation glacier' and avoid assuming that all glaciers that fall into this category are cirques.

428. 'valley glaciers are larger' Again, valley glaciers is an interpretation. Your evidence doesn't allow you to equate 7B glaciers with size. Your evidence suggests these are less steep with lower mean elevations cf. glaciers in 7A. Amended: see response 4.

435-436. Again, link to your evidence. Avoid talking about 'area'. Here you're talking about very shallow gradient glaciers with very low mean elevation. These have the highest drainage densities. And they have the lowest incidence of crevassing (Fig 5h). Amended: see

441. 'connectivity'? What does this refer to? Amended – this paragraph was rewritten and no longer includes this sentence.

442. "…based on the locations of our mapped channel termini". Up to this phrase, this sentence sounded like a summary of your previous paragraph and conceptual model shown in Fig 7. This phrase throws a spanner in the works as you have not yet discussed possible links between locations of mapped channel termini and lag times between melt and discharge. I've read on and it appears as though you're going to talk about this wrt 2 case study glaciers. I suggest rewriting this sentence to introduce the work you'll present in this paragraph. Amended.

442-3. As I mentioned before, what exactly does this mean? How do you define 'average glacier'? Amended – we have reworded this to clarify that we are referring to the average proportions of channel termination locations (e.g., englacial or running directly off the terminus), calculated by averaging the values from all individual glaciers.

444. "… the largest glacier, Grosser Aletschgletscher (type B in Fig. 7)" OK, so your largest glacier is type B not C! This adds to my advice that you should not refer to glacier area as a control on drainage density (for which you don't have evidence). Amended – we no longer focus on glacier area as a control on drainage density.

451-2. where are the 'trace crevasses' in Fig 8? They are not labelled as such. Amended – this figure was removed as we have shifted the focus away from qualitative observations.

440-456. This paragraph is a little rambling and unfocussed. State at the outset what you're aiming to achieve here. It looks like you want to show us how different types of glacier (low drainage density to high drainage density - types A to C in Fig 7)  have channel segments that

terminate in different ways. Correct? One way to have done that would have been to cluster your glaciers into the three types (based on their statistical attributes - either the raw

variables or the PCs) and then looked at the number of channel termination types in each of the 3 clusters. You'd hypothesize that your steep, high elevation, low drainage density glaciers (type A) would have most channel termini ending in crevasses. Type C would have most ending by flowing off the snout or side. And type B would be somewhere in between. That would have provided you with the evidence you need to support all your statements relating to crevasses affecting channel / drainage characteristics. But you're not doing that.

Instead, you're picking a single case study example of Type B and Type C and telling us about their channel termini characteristics. For completeness why not do the same for a Type A glacier? Explain at the outset that this is what you're doing in this paragraph. In fact, would it be best to do all this BEFORE you present your conceptual model because in your description of the 3 types of glaciers you mentioned crevasses without really showing us any evidence that crevasses were relevant. Amended: see response 2.

460. You refer to ' meltwater overtops the crevasse' . Does this make it a trace crevasse then? Amended – this figure was removed as we have shifted the focus away from qualitative observations.

464. You refer to Oberer Theodulgletscher and Grosser Aletschgletscher but as I said above it'd be helpful to add an example of a type A glacier wouldn't it? Note the proper nouns (names of glaciers here) do not need to be prefixed by 'the'. Amended – we now focus on our cluster analysis rather than individual glaciers.

473. As I said before, you don't measure 'hypsometry' but you do show the role of glacier mean elevation so I'd refer to that here. Amended: see response 2.

472-3. I'd agree with the statement: "Hence, categorising glaciers based on their slope and mean elevation is beneficial because it provides insight into the anticipated drainage density of a glacier" because this is based on your evidence from your correlation matrix and PC2. Amended: see response 2.

474-5. This statement about things providing insight into "channel pathways (i.e., sub-/englacially or proglacially terminating), and whether a higher amount of surface-to-bed meltwater transfer is likely" is a bit clumsy and is less well supported by your evidence but see my suggested way forward above wrt cluster analysis..Amended: see response 2.

Response to comments on section 5.1:

We have made some key changes to this section to address all the comments above:
1. **Section name -** The name of this subsection has been changed to 'The influence of glacier characteristics on supraglacial channel networks' to better reflect the focus on the impact that glacier slope and elevation have on glacier drainage density and channel termini locations.
2. **Principal Component Analysis & cluster analysis** – Following the suggestions from the reviewer, we have now performed a cluster analysis to group glaciers based on their properties. This data is also now visualised using a PCA biplot in Figure 7. We find that the clusters presented in Figure 7 support our interpretations as they show an increase in drainage density with a decrease in slope and glacier elevation. Channel termini locations also differ between the glacier types identified in our cluster analysis, which

helps strengthen our interpretations. We have now added this analysis into our methods, results, and use it to strengthen our discussion. We thank the reviewer for this suggested analysis.

3. **Structure** – We have reconfigured the structure of section 5.1 to mention data that forms the basis of our conceptual schematic before its introduction. The section now discusses the results of our correlation matrix and initial PCA first, clearly outlining that drainage density is primarily influenced by slope and elevation. We then introduce our data on channel termini locations and how our cluster analysis can explain inter-glacier variations in meltwater pathways based on glacier slope and elevation. After this, we introduce our conceptual schematic as our interpretations are supported by the data presented in the previous two paragraphs. The final part of this section uses our conceptual schematic to speculate how different glacier types may produce different hydrographs in response to a surface melt event.

4. **Conceptual schematic –** we have changed the names of each type of glacier on the conceptual schematic to better reflect what our data shows. For example, Type A in our schematic now refers to a 'steep, high elevation glacier' rather than a 'cirque', and we have removed the mention of glacier size in order to better focus on what our data clearly shows.
We also now focus on using our data to support the interpretations made in the schematic. While we previously referred to specific glaciers for each type, we have removed this information, even though these glaciers are still classified as the same type using our cluster analysis. This is to avoid focusing on specific glaciers and instead simply use our cluster analysis to provide support for our interpretations.

**Section 5.2**

479 "…slope affects sinuosity". Clarify you're talking about channel slope. And avoid the word 'affects' as this implies causation whereas you just show correlation. Amended.

480. As well as Fig 5a you could refer to Fig 6 as this is supported by your correlation work. It's also interesting that this -ve association between channel slope and sinuosity remains apparent in PC5. Amended – we have cited Figure 6 and also Table A1 given that this provides additional evidence of this relationship.

481. After stating the correlation between channel slope and sinuosity, you could interpret it and discuss processes. It's what you'd expect isn't it? Steeper slopes → lower sinuosity. On steep gradients, water has high energy and tends to take the most direct downslope path. This reduces the development of bends, keeping channels relatively straight. Gentler slopes → higher sinuosity. On low gradients, flow velocity and stream power are lower. Water has less ability to cut straight downslope and instead meanders laterally, forming more sinuous channels. We agree. After we mention the correlation between slope and sinuosity, we now also cite fluvial literature which further supports the logic of our observations.

482-3. Fig 4e doesn't provide all the evidence to support the statement. It just provides examples of two sinuous channels on clean ice. Are there other channels not shown that are straight and on dirtier ice? I'd be tempted to delete this sentence. We have removed this sentence to avoid focusing on specific glaciers, but the examples shown in 4e are indeed the most sinuous channels in our dataset, which we include as an example.

483-5. The statement spanning these lines could also explain the sinuosity of at least one of the channels shown in Fig 4e as that channel seems to be sourced on debris. So this all seems a little weak. We now acknowledge that we find differences in channel sinuosity depending on how extensive the debris cover is surrounding the channel. We suggest this could be due to sediment transport affecting channel sinuosity, as prior research has suggested this may be the case (e.g. Rhoads and Welford, 1991; Boyd et al., 2024).

486-500. I don't really see what you're trying to explain here. Channel slope controls stream power. Surely this promotes straighter channels. So your finding of more sinuous channels on gentler slopes makes perfect physical sense to me. Like you say, discharge also controls stream power so if discharge is higher through gentler slopes, then I suppose that could override the slope control, promoting straighter channels on gentler slopes and more meandering channels on steeper slopes. But you do not find this! Nor do you have any discharge data! So why try to explain it? Are you trying to explain Fergusson's and StG and M's findings, which would seem to buck what makes more physical sense? Haven't they already done that in their papers? Is it your job to do it here? I suppose you could just briefly explain that your findings are different to those from the Arctic glacier but explain briefly why the earlier work provides evidence which is contrary to what you might expect. But be brief. Amended - We have rewritten this section to shift the focus towards our data/observations and have since added in fluvial citations that provide good support for our findings. We still briefly acknowledge the differences from St. Germain & Moorman's findings, but do not focus on explaining these differences.

**Section 5.3**

503-5. Split the sentence to be clear. So say: "Previous research on supraglacial channel morphometry has focused predominantly on the GrIS (e.g., Smith et 503 al., 2015; Karlstrom and Yang, 2016; Yang and Smith, 2016; Yang et al., 2016, 2021, 2022). We find some similarities between the drainage patterns observed on Valais glaciers, and those on the GrIS." Amended.

505-6. I don't think you should resort to just picking out this one glacier in 4a. Your work has not focussed on how dendritic drainage patterns are. Can you not compare some of the quantitative channel and glacier characteristics and relationships between Valais and the GrIS, e.g. drainage density or sinuosity and their correlations with channel / glacier attributes? Much of the research that looks at the morphometry of channels on the GrIS focuses on channel network properties (i.e., stream order and bifurcation ratios) (e.g., Yang et al., 2016). It is not possible for us to quantitatively compare these values to our dataset because we were not able to reliably automate the calculation of stream order from our channel centerlines. We also cannot directly compare our drainage density values to the GrIS due to the use of different resolution imagery, nor are we aware of any papers that have summary statistics for sinuosity, although many papers mention it in passing. Hence, any comparison between both environments will have to be more qualitative, but we still think there is value in this and have tried to refine this section to focus more on the data presented in the discussion.

507-8. "…some glaciers in Valais display parallel, weakly interconnected channel networks, likely due to insufficient distance for meltwater to converge into a single channel" Seems a bit anecdotal. Do you have strong evidence that this is prevalent on Valais glaciers? Doesn't the GrIS also display this in places? We have rewritten this paragraph to better explain that channel networks across Valais exhibit a range of drainage patterns,

i.e., some glaciers contain much more dendritic networks, while others are less interconnected. Now we more clearly outline that a range of drainage patterns are found on the GrIS. However, we suggest that controls on drainage types (i.e., dendritic vs parallel) on the GrIS may not fully explain the drainage patterns found on Valais glaciers due to the difference in area and high crevasse density. This is done in a speculative manner and does not suggest that we have more data than we do.

509 "…these networks…". Which networks are you referring to here? The parallel weakly interconnected ones? So Yang et al 2016 find these follow Horton's laws on the GrIS? Why not calculate them for your drainage networks? Then you could properly compare Valais glaciers with GrIS. So far, you've not convincingly told the reader based on evidence whether and how the supraglacial hydrology differs between Valais glaciers and the GrIS. We previously intended to calculate stream order and bifurcation ratios for our dataset, but this has not been possible to automate because the polyline network often bifurcates and re-merges, which doesn't align with the simple, branching structure that stream ordering tools are designed for. Small overlaps between segments and issues during raster conversion also broke flow continuity, leading to unreliable results.

511. You say 'trace crevasses exhibit a strong control on meltwater routing on Valais glaciers'. I do not believe you can conclude this from the evidence you've presented. The only mention of 'trace crevasses' in your results is on p18 wrt Oberer Theodulgletscher when you say "it is not known whether meltwater enters englacially or is routed on the glacier surface through trace crevasses (e.g., Fig. 8). Fig 8 makes no mention of 'trace crevasses' explicitly. Amended – we have removed mention of trace crevasses.

512. I'd delete ref to Antarctic ice sheet – see the subheading title! Amended.

514. should say 'Valais glaciers and the GrIS…' Amended.

516. I'm still unsure what the "average Valais glacier" is. Amended - This was removed when this paragraph was rewritten. Any mention of the "average Valais glacier" earlier in the text has been rewritten to provide further clarification on what we were referring to.

520 "appears to affect…" The language in this paragraph has been modified to acknowledge uncertainty.

520-1. Where did you show debris cover affects channel distribution? And what do you mean by 'distribution'? The only place I can think of is Line 295 onwards "Qualitative observations suggest that channel distribution and morphology are controlled by glacier structure and topography. For example, channels often occur along the interface between debris-covered and bare ice (e.g., Fig. 4b), particularly adjacent to medial moraines, where channels are confined to a topographic depression, commonly occurring at the confluence between two tributaries." This is only one example you've shown and I bet you could find at least one example of a channel on the GrIS flowing adjacent to a medial moraine. I don't think you have enough evidence to say anything meaningful here about the similarities or differences between Valais glaciers and the GrIS in terms of the effects of debris on channel 'distribution'. What about morphology? Any studies of sinuosity and role of debris on GrIS? We have rewritten this paragraph to acknowledge more uncertainty with regard to these interpretations. The mention of channel distribution in the first sentence of this paragraph is in reference to the discussion of how moraines may affect the transport of meltwater and affect where larger channels are observed. We also refer to

the distribution of areas with high channel density when discussing roughness later in the paragraph. Unfortunately, we are not aware of any studies on the GrIS that could meaningfully strengthen this section, hence the more speculative nature of our discussion.

522-4. This sentence seems misplaced as it's for a glacier in Svalbard. Suggest delete. We have kept the reference in but rewritten this section. We have kept this citation as it is useful to acknowledge that debris may affect channel density, regardless of where that research was conducted.

524. What do you mean by "scale" here? And are you comparing Valais glaciers with the GrIS? Amended - We mean the extent of debris cover and have since rewritten this paragraph to clarify its meaning.

526. At the end of this para I don't have a clear sense of the similarities and / or differences between Valais glaciers and the GrIS in terms of the role of debris on supraglacial channels. This paragraph has been rewritten to more clearly emphasise the similarities between the data we have which is comparable with the GrIS, which has hopefully strengthened it. However, as few studies have investigated the impact of debris on channel distribution and morphology, this section has to remain fairly speculative. For example, we acknowledge that the extent of debris coverage varies between the two environments and then focus on discussing how debris may affect channel distribution and morphology when it is abundant.

**Section 5.4**

530. 'may' => 'are likely to continue to' Amended

534 'large enough' => 'sufficient' Amended

537-9. Seems a shame that you can't add anything to this based on your work. What would you expect from Fig 5c? Figure 5c is discussed on lines 343 to 348 and 627 to 631. We could further speculate about the relationship between debris and sinuosity, but given that most suggestions have favoured removing speculation, we do not deem this to be a fitting addition.

540-47. All this is not about the future evolution of channel systems. You need a separate section, or you need to reframe this section and provide a new heading. If you're talking about future work, you should discuss the possible imitations on your work of using July imagery rather than later season imagery. Amended - We have split this paragraph to keep these lines separate and have changed the subheading to acknowledge that this section also discusses research gaps.

**Conclusion**

550. I'd say 'From a sample of 285 glaciers…" Amended.

551-2. Here you talk about variability in glacier drainage density. This relates to PC2. Before this you could report the PC1 finding of high glacier area variability with large glacier area correlated with low minimum glacier elevation and low maximum channel elevation. And the possible reasons / implications of this. Of course, you'd need to have this as a discussion

point earlier, as I suggested you could. This section has been rewritten in line with the reviewer's suggestions below. We do still mention glacier area briefly, but we focus more heavily on what the clear controls on glacier drainage density are (i.e., glacier elevation and slope).

553. As well as ref to Fig 7 I'd use (or add) your correlation matrix Fig 6 here. The rewrite of this section doesn't include references, as these are not commonplace to include in the conclusion. We do, however, provide references to figures in our results and discussion, which now better support the statements in the conclusion.

553-554. " The presence of channels is primarily dictated by a sufficiently supply of meltwater (i.e., large enough glacier area)" Note sufficiently => sufficient. But more importantly, this can't be a direct conclusion of your work as you do not measure melt water supply in your study. In your conclusions, explain what you actually find and then explain what your interpretation is. A large glacier area is not necessarily synonymous with a large supply of meltwater as you seem to be suggesting here. I assume this conclusion relates to your first result on lines 237-8 "Glaciers with channels (n = 85) have a larger mean area than glaciers without channels (n = 200) (mean area = 5 237 km2 vs. 0.6 km2) and all glaciers larger than 5.6 km2 contain channels
> 0.5 m wide (Table 1, Fig. 3a)." If so would it be useful to remind us of this or refer to the relevant Fig? Amended – the mention of meltwater supply has been removed.

554-555. "…and an uninterrupted distance for meltwater to coalesce (i.e., absence of crevasses)" Where did you show us that the 85 glaciers had fewer crevasses on them than the 200 glaciers that did not have channels, as you are suggesting here? This point refers to our finding of higher drainage densities on less crevassed glaciers (Fig. 5h) and on lower relief slopes (Fig. 6), which are assumed to be less crevassed. We see how our original wording may have been confusing and now just mention the controls on drainage density rather than presence.

555-57. I can't see evidence for this sentence. You're suggesting that there are thresholds of glacier slope and glacier area beyond which channels don't form. Where is the evidence for this? Fig 5b does not show this. You also suggest that glacier slope and glacier size provide a limit on channel length. Again, where is the evidence for this? Fig 5b shows the relationship between channel slope and channel length of your sample. There is nothing about 'limits' here. This point is no longer a primary focus in the revised section. However, we note that Fig. 5b shows an absence of long channels on steep slopes and no channels above a certain slope, which we interpret as a constraint likely related to crevasse density. While it may not be the only control, glacier area also places an upper bound on channel length. The revised text no longer implies the existence of hard thresholds.

557. "…strong structural controls on channel distribution exist." It seems really odd to be talking about channel distribution as that is not really what your paper looks at. For the glaciers with channels on them, your paper focuses mostly on the relationships between channel and glacier characteristics, not about the presence or absence of channels in particular places. Amended – the conclusion now focuses more on the relationships between channel and glacier characteristics.

557-8. "For example, trace crevasses have been observed to act as preferential meltwater pathways, resulting in channels forming perpendicular to ice flow."  Where have they been observed? I think you give just two examples in your study (Fig 4c and Fig 8) and this point

is mentioned almost in passing. It is hardly a major conclusion of your work. On line 452 you state: "Observations have shown that trace crevasses may act as a preferential meltwater pathway, often resulting in channels forming perpendicular to ice flow (e.g., Chen et al., 2024). " So this is a conclusion of another paper! This is something we commonly observed whilst mapping, which is why it was initially mentioned. In this revision, we have removed mention of it in an effort to shift the focus more towards our quantitative results.

559. "Channels also commonly form parallel to medial and lateral moraines due to topographic confinement." Again, is this really a major conclusion of your analysis? Where do you show this? where is the evidence that this is widespread? As mentioned above, this statement is based on our observations during the mapping process. We have since toned down the mention of qualitative observations per the reviewer's suggestion.

560. You refer to Fig 5a here but wouldn't Fig 3f be better? What is your definition of low? Can you say that x% are < some threshold? Amended – we now just mention that there is a negative association between sinuosity and slope.

560-61. "….highly sinuous channels are present, particularly on moderately debris-covered ice and lower-relief glacier termini." Again, where is the evidence.? See my comments on your section 5.2. Amended – we've removed mention of glacier termini here.

566-575. You may wish to reword or refocus this and the entire conclusion after thinking more about the Discussion section. Amended – this section has been revised.

Response to comments on the conclusion:
We have rewritten the conclusion section to better summarise our revised discussion. Hopefully, it is evident that the conclusion is now based on our interpretation of the data presented in the results section, and focuses less on qualitative observations. In this new conclusion, we do not include references as it is not commonplace, but references are present when the points in the conclusion are discussed in the previous section.